# Instruction-Tuned Video-Audio Models Elucidate Functional Specialization in the Brain

## Abstract

Recent voxel-wise multimodal brain encoding studies have shown that multimodal large language models (MLLMs) exhibit a higher degree of brain alignment compared to unimodal models in both unimodal and multimodal stimulus settings. More recently, instruction-tuned multimodal models have shown to generate task-specific representations that align strongly with brain activity. However, prior work evaluating the brain alignment of MLLMs has primarily focused on unimodal settings or relied on non-instruction-tuned multimodal models for multimodal stimuli. We still lack a clear understanding of whether instruction tuning drives IT-MLLMs to organize their representations around functional task demands or if they simply reflect surface semantics. To address this gap, we investigated brain alignment, that is, measuring the degree of predictivity of neural activity recorded while participants were watching naturalistic movies (video along with audio) with representations derived from MLLMs. We utilized instruction-specific embeddings from six video and two audio instruction-tuned MLLMs. Experiments on 13 video task-specific instructions show that instruction-tuned video MLLMs significantly outperform in-context learning multimodal models (by ∼9%), non-instruction-tuned multimodal models (by ∼15%) and unimodal models (by ∼20%). Our evaluation of MLLMs for both video and audio tasks using language-guided instructions shows clear disentanglement in task-specific representations from MLLMs, leading to precise differentiation of multimodal functional processing in the brain. We also find that MLLM layers align hierarchically with the brain, with early sensory areas showing strong alignment with early layers, while higher-level visual and language regions align more with middle to late layers. These findings provide clear evidence for the role of task-specific instructions in improving the alignment between brain activity and MLLMs, and open new avenues for mapping joint information processing in both systems. Future work should test causal understanding via model-driven experiments and the development of task-conditioned naturalistic stimuli for in-lab and in-silico settings, enabling tighter links between controlled and naturalistic paradigms.

## 1 Introduction

The alignment between internal representations of multimodal Transformer models and cortical activation patterns obtained from naturalistic stimuli has emerged as a key focus in the study of brain-model correspondence. Recent research has demonstrated that multimodal models in brain encoding can be broadly categorized into two settings (see Appendix A Table 4): (i) multimodal models evaluated with unimodal stimuli (Doerig et al., 2022; Wang et al., 2023; Oota et al., 2022b; Popham et al., 2021; Tang et al., 2024; Oota et al., 2025a; Srijith et al., 2025), and (ii) multimodal models evaluated with multimodal stimuli (Nakagi et al., 2024; Subramaniam et al., 2024; Dong & Toneva, 2023a; Oota et al., 2025b; Sartzetaki et al., 2025). In the former setting, brain recordings are obtained from unimodal image stimuli, but representations from multimodal models, which integrate modalities such as vision and language, achieve a higher degree of brain alignment compared to vision-only models (Doerig et al., 2022; Wang et al., 2023; Oota et al., 2022b; Popham et al., 2021). This observation holds true to the new class of instruction-tuned multimodal large language models (MLLMs), especially when prompted with natural instructions (Oota et al., 2025a). In the latter setting, where brain recordings are obtained from multimodal stimuli (e.g., watching movies with both visual and

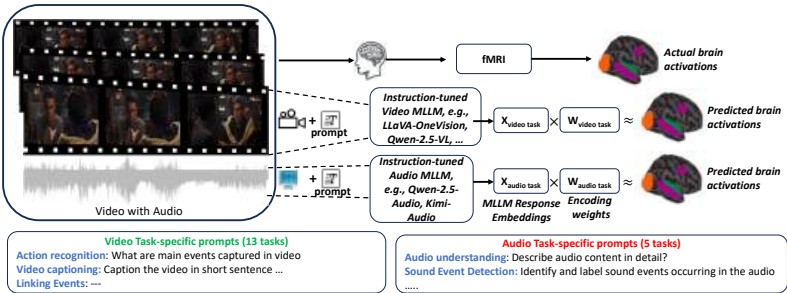

Figure 1: Leveraging instruction-tuned multimodal video and audio models for brain encoding with a diverse set of instructions. For the given movie clip, we can obtain different multimodal representations using instructions that ask the model to (i) generate the caption of the video, (ii) identify whether temporal events are present, (iii) determine the primary colors dominant in the video, etc. Using instruction-specific representations (X), we estimate the alignment using a simple linear function $f$ (ridge regression), which maps MLLM representations to brain recordings. Here, W denotes voxelwise encoding model weights.

auditory stimuli), studies show that multimodal models exhibit higher degree of brain alignment over unimodal models (Dong & Toneva, 2023a; Oota et al., 2025b). While prior studies have examined brain alignment with instruction-tuned MLLMs (IT-MLLMs), they have largely been limited to unimodal stimuli, or have used non-instruction-tuned models in the context of multimodal stimuli. In this work, we bridge this gap by systematically investigating IT-MLLMs in the presence of rich multimodal stimuli. Specifically, we assess how well representations elicited through naturalistic, task-specific instructions involving both video and audio align with brain activity across the cortical hierarchy, from early sensory regions to higher-order cognitive areas.

Several unimodal studies report that task-specific fine-tuned Transformer models better align with brain activity during text (Oota et al., 2022a; Aw & Toneva, 2023; Sun & Moens, 2023; Oota et al., 2024b), speech (Oota et al., 2023; Tuckute et al., 2023; Oota et al., 2024a), and vision (Wang et al., 2019; Conwell et al., 2022) processing, outperforming pretrained models in brain predictivity. However, these models are task-specific, limiting generalization, requiring separate data and training per task. Instruction-tuning (Xu et al., 2023; Dai et al., 2023; Liu et al., 2024) offers a scalable alternative, fine-tuning a single LLM across diverse NLP tasks and surpassing task-specific models (Taori et al., 2023; Touvron et al., 2023; Jiang et al., 2023; Abdin et al., 2024; Dubey et al., 2024), while showing stronger brain alignment (Sun et al., 2023; Sun & Moens, 2023; Loong Aw et al., 2024) (see Appendix B for more.) Building on this, recent work aligns IT-MLLMs with brain data for text (Benara et al., 2024) and images (Oota et al., 2025a), though limited to unimodal stimuli. Motivated by advances in multimodal MLLMs for video and audio tasks, we ask: *(a) how do IT-MLLMs elicit task-specific representations using language-guided instructions and whether these representations go beyond surface semantics?* (b) *what is the nature of the representational changes across layers before and after instruction tuning?* (c) *finally, do instruction-tuned video/audio MLLMs prompted with natural language yield better brain alignment than their pretrained in-context learning and non-instruction-tuned counterparts, while also distinguishing task- specific representations?* To our knowledge, this is the first study to use such MLLMs to model fMRI responses across video and audio tasks (workflow in Fig. 1).

Using brain recordings from participants watching several popular movies with audio (Boyle et al., 2020), we investigate the brain alignment of IT-MLLMs. Specifically, we evaluate six video IT-MLLMs, two audio IT-MLLMs, two pretrained video MLLMs with in-context learning, two non-instruction-tuned multimodal models (video+audio), two unimodal models for video and one unimodal model for audio. These models are probed with 13 video task-specific instructions, and 5 audio task-specific instructions. Overall, this study addresses the following research questions: (1) How do different task-specific instructions influence the degree of brain alignment in instruction-tuned video and audio MLLMs? (2) Do instruction-tuned video MLLMs exhibit better brain alignment than their audio counterparts when exposed to multimodal stimuli? (3) Do IT-MLLMs produce functionally distinct representations that map onto different brain regions, offering a data-driven alternative to traditional experimental stimuli? (4) How do task instructions related to semantic categories (e.g., narrative understanding, spatial reasoning) explain differential activation across language, auditory, and visual brain regions?

To further quantify how IT-MLLMs capture shared and distinct neural processes across tasks, we use a variance partitioning approach. This analysis reveals the unique and overlapping contributions of individual task-specific representations to brain responses, enhancing our understanding of the brain's functional organization in processing multimodal information.

Our analysis of IT-MLLMs and brain alignment with multimodal stimuli reveals several key conclusions: (i) Video-based IT-MLLMs show significantly higher brain alignment than audio-based IT-MLLMs, pretrained in-context learning MLLMs, non-instruction-tuned multimodal models, as well as unimodal video and audio models. This holds across the whole brain, as well as within language, visual and auditory regions. (ii) On the other hand, Audio MLLMs outperform both non-instruction-tuned multimodal and unimodal models only in the auditory cortex (AC) and middle frontal gyrus (MFG) language regions, while exhibiting comparable performance in other language-related areas. (iii) Surprisingly, both video and audio MLLMs generate task-specific representations based on task-instructions and effectively differentiate functional processing across brain regions. For example, audio understanding and captioning tasks show stronger alignment with language areas, while sound event detection aligns with the auditory cortex and temporal lobe. *Further, probing IT-MLLMs reveals that instruction-tuning organizes representations primarily by functional task demands and less by semantic representations.* (iv) Grouping 13 video tasks into 5 semantic categories reveals strong alignment of MLLM representations with brain sub-regions in line with the existing literature. Tasks involving language and narrative understanding exhibit stronger alignment in language-related sub-regions such as angular gyrus and lateral temporal regions, consistent with prior findings on event structure representation in naturalistic stimuli (Baldassano et al., 2017). Similarly, spatial understanding tasks engage regions of the dorsal visual pathway, particularly the intraparietal sulcus and surrounding parietal cortex. *Thus, the representations extracted from task-selective subspaces carved out by IT-MLLMs enable better characterization of multimodal information processing across cortical hierarchies.* Overall, our analysis reveals that IT-MLLMs capture both hierarchical and task-specific brain representations, making them powerful tools for studying functional specialization and bridging cognitive modeling with neuroscience. Our code is part of the supplementary material.

## 2 DATASET AND MODELS

### 2.1 BRAIN IMAGING DATASET

We experiment with Movie10 (Boyle et al., 2020), a multimodal naturalistic fMRI dataset, obtained from the Courtois NeuroMod databank. This dataset was collected while four human subjects (s1, s2, s3, s5; data for s4 and s6 is not public) passively watched four different movies: *The Bourne supremacy (∼100 mins)*, *The wolf of wall street (∼170 mins)*, *Hidden figures (∼120 mins)* and *Life (∼50 mins)*. Among these, *Hidden figures* and *Life* are repeated twice, with the repeats used for testing and the remaining movies for training. We use *Life* movie for testing where we average the two repetitions to reduce noise. This is among the largest publicly available multimodal fMRI datasets by samples per participant, with 4024 TRs (Time Repetitions) of *The Bourne supremacy* and 6993 TRs of *The wolf of wall street* for training and 2013 TRs of *Life* for test. Train and test sets are totally disjoint. The fMRI data is collected every 1.49 seconds (= 1 TR).

The dataset is already preprocessed and projected onto the surface space ("fsaverage6"). We use the multimodal parcellation of the human cerebral cortex based on the Glasser Atlas (which consists of 180 regions of interest in each hemisphere) to report the ROI (region of interest) analysis for the brain maps (Glasser et al., 2016). This includes four visual processing regions (early visual cortex (EVC), object-related areas (LOC), face-related areas (OFA) and scene-related areas (PPA)), one early auditory area (AC), and eight language-relevant regions, encompassing broader language regions: angular gyrus (AG), anterior temporal lobe (ATL), posterior temporal lobe (PTL), inferior frontal gyrus (IFG), inferior frontal gyrus orbital (IFGOrb), middle frontal gyrus (MFG), posterior cingulate cortex (PCC) and dorsal medium prefrontal cortex (dmPFC), based on the Fedorenko lab's language parcels (Milton et al., 2021; Desai et al., 2023). We show the flatmap with these labeled ROIs in Appendix Fig. 6 and list the detailed sub-ROIs of these ROIs in Appendix C.

**Estimating cross-subject prediction accuracy.** To account for the intrinsic noise in biological measurements, we adapt Schrimpf et al. (2021)'s method to estimate the cross-subject prediction accuracy for a model's performance for the Movie10 fMRI dataset. Each subject $s \in ([1,4])$ is chosen as the prediction target and the other three are used to predict this target. We use a voxel-

Table 1: Pretrained MLLMs for video, audio vs. multimodal, unimodal models (IT: Instruction-tuned) (IC: In-context learning).

| Model Name | IT | #Layers | Modality |
|---|---|---|---|
| InstructBLIPVideo | ✓ | 33 | Video+Text |
| Video-LLaVA | ✓ | 33 | Video+Text |
| LLaVa-NeXT-Video | ✓ | 33 | Video+Text |
| Qwen-2.5-VL | ✓ | 29 | Video+Text |
| Videochat-R1 | ✓ | 29 | Video+Text |
| LLaVA-OneVision | ✓ | 28 | Video+Text |
| Qwen-2.5-Audio | ✓ | 29 | Audio+Text |
| Kimi-Audio | ✓ | 29 | Audio+Text |
| Qwen-2.5-Omni (IC) | ✗ | 29 | Video+Audio+Text |
| InternVL (IC) | ✗ | 29 | Video+Text |
| VILA | ✗ | 29 | Video+Audio |
| TVLT | ✗ | 12 | Video+Audio |
| VideoMAE | ✗ | 24 | Video |
| TimeSFormer | ✗ | 12 | Video |
| AST | ✗ | 24 | Audio |

Table 2: Instructions for various multi-modal audio tasks.

| Task | Description |
|---|---|
| Audio Understanding | Can you describe the audio content in detail? |
| Audio Comprehension | What are people doing in the audio? |
| Audio Captioning | Caption the audio in a short sentence. |
| Sound Event Detection | Identify and label the sound events occurring in the audio. |
| Speaker Identification | Who is speaking in the audio? |

wise encoding model (see Section 3) to predict one participant's response from others. The detailed approach is described in Appendix D. Note that the estimated cross-subject prediction accuracy is based on the assumption of a perfect model, which might differ from real-world scenarios, yet offers valuable insights into model's performance. We present the cross-subject prediction accuracy across voxels for the Movie10 fMRI dataset for each of the four participants in Appendix D. The plots show that across all participants higher activity is observed in the language and visual regions with a max correlation up to 0.4 implying that data has low noise and low cross-subject variability.

## 2.2 INSTRUCTION-TUNED MULTIMODAL MODELS FOR VIDEO AND AUDIO

To investigate whether IT-MLLMs models, when prompted using natural language-guided instructions, align with the way humans process multimodal information in the brain, we consider six popular modern instruction-tuned video MLLMs (InstructBLIPVideo (Dai et al., 2023), Video-LLaVA (Lin et al., 2024), LLaVA-Next-Video (Zhang et al., 2024), Qwen-2.5-VL (Wang et al., 2024), Videochat-R1 (Li et al., 2025), LLaVA-OneVision (Li et al., 2025)) and two instruction-tuned audio MLLMs (Qwen-2.5-Audio (Chu et al., 2024), Kimi-Audio (Kimi Team, 2024)). We also experiment with two pretrained video MLLMs with in-context learning (Qwen-2.5-Omni (Xu et al., 2025) and InternVL (Chen et al., 2024)), two non-instruction-tuned multimodal (VILA (Lin et al., 2023) and TVLT (Tang et al., 2022)), two video unimodal models (VideoMAE (Tong et al., 2022) and TimeSFormer (Bertasius et al., 2021)), and one audio unimodal (AST (Baade et al., 2022)) model. Details for these models are reported in Table 1.

## 2.3 NATURAL LANGUAGE INSTRUCTIONS AND FEATURE EXTRACTION FROM IT-MLLMS

**Video-specific tasks.** To ensure the diversity of task-specific instructions while considering videos as input, we consider 13 instructions, as shown in Table 3, and extract the language-guided representations from multimodal instruction-tuned video models. This set of 13 tasks are inspired from VideoInstruct100K dataset (Maaz et al., 2024). We borrowed those tasks, which are generally applicable to any video regardless of the contents in the image frames. We provide a sample of generated outputs for all the six video MLLMs in Tables 5, 6, 7, 8, 9 and 10 in Appendix G.

To extract instruction-specific representations from multimodal instruction-tuned video models for the brain encoding task, we input a video and task instruction to obtain the embeddings for the language-guided instruction. For in-context learning models, a video is paired with a natural language prompt without instruction tuning. For TVLT and VILA, we input video and audio. For TimesFormer and VideoMAE we input video only. We perform zero-shot inference on these models. For all multimodal instruction-tuned video models, we use the pretrained Transformer weights, which generate hidden state representations at each layer. We then average these hidden state representations at layer level of output generated tokens to obtain final embedding at each layer for each video with respect to task instruction.

**Audio-specific tasks.** Similar to video tasks, we consider five natural instructions while considering audio as input, as shown in Table 2, and extract the language-guided representations from multimodal instruction-tuned audio model. We provide a sample of generated outputs for one of the instruction-tuned audio models across the five tasks in Tables 11 and 12 in Appendix G.

Table 3: Instructions for various multimodal video tasks.

| Task | Description |
|------|-------------|
| Action Recognition | What are the main events captured in the video? |
| Video Understanding | Can you describe the video content in detail? |
| Visual Question Answering | How many people are in the video, and what are they doing? |
| Video Captioning | Caption the video in a short sentence. |
| Object and Scene Recognition | What are the main objects and people visible in the video? Describe each one briefly. |
| Commonsense Reasoning | Why did the character take this action? What could have motivated them to do this? |
| Spatial Understanding | Where is this video taken from? What place/landmark is shown in the video? |
| Temporal Ordering | Step-by-step describe the activity shown in the video. |
| Video reasoning | What is unusual about this video? |
| Narrative Understanding | Summarize the main storyline of the movie. What is the central conflict, and how is it resolved? |
| Emotion and Sentiment Analysis | What emotions do the characters express during the video? How does the video make you feel overall? |
| Global Appearance | Describe changes in characters' appearances throughout the video, including any noticeable outfit changes. |
| Linking Events | Explain how an early event in the video influences later developments. |

Similar to instruction-tuned video models, to extract instruction-specific representations from the multimodal instruction-tuned audio model for the brain encoding task, we input a audio and task instruction to obtain the embeddings for language-guided instruction. For AST we input audio only. We follow similar feature extraction method as video-tasks to extract audio task representations.

## 3 METHODOLOGY

**Voxel-wise encoding model.** We train banded ridge regression based voxel-wise encoding models (la Tour et al., 2022) to predict the fMRI brain activity associated with the stimulus representations obtained from 13 task-specific instructions from multimodal instruction-tuned video models. Banded ridge regression optimizes a different regularization hyperparameter per feature space, and decomposes the explained variance over feature spaces. This decomposition helps in identifying which task-specific instruction contributes most to the explainable variance in different brain regions. Overall, banded ridge regression helps to accurately identify the contribution of each task-specific instruction, leading to better prediction accuracy and better interpretability. We employ z-score thresholding separately for both input stimulus representations and brain recordings for training and test datasets. For each subject, we account for the delay in the hemodynamic response by modeling hemodynamic response function using a finite response filter (FIR) per voxel with 5 temporal delays (TRs) corresponding to ∼7.5 seconds (Huth et al., 2022). Formally, at each time step $t$, we encode the stimuli as $X_t \in \mathbb{R}^D$ and brain region voxels $Y_t \in \mathbb{R}^V$, where $D$ denotes the dimension of the concatenation of delayed 5 TRs, and $V$ denotes the number of voxels. Overall, with $N$ such TRs, we obtain $N$ training examples. Detailed hyper-parameter settings are in Appendix E.

**Evaluation metrics.** We evaluate our models using Pearson Correlation (PC), which is a standard metric for evaluating brain alignment (Jain & Huth, 2018; Schrimpf et al., 2021; Goldstein et al., 2022). Let TR be #time repetitions in the test set. Let $Y = \{Y_i\}_{i=1}^{TR}$ and $\hat{Y} = \{\hat{Y}_i\}_{i=1}^{TR}$ denote actual and predicted value vectors for a single voxel. Thus, $Y$ and $\hat{Y} \in \mathbb{R}^{TR}$. We use PC to compute the correlation function, $corr(Y, \hat{Y})$. The final measure of a model's performance is obtained by calculating Pearson's correlation between the model's predictions and neural recordings. To quantify the model predictions, the resulting model prediction correlations are divided by the estimated cross-subject prediction accuracy; and averaged across voxels, regions, and participants, resulting in a standardized measure of performance referred to as normalized brain alignment. For calculating *normalized alignment*, we select the voxels with cross-subject prediction accuracy $\geq 0.05$.

## 4 RESULTS

### 4.1 INSTRUCTION-TUNED VIDEO MLLMS REPRESENTATIONS ALIGN WELL WITH BRAIN ACTIVITY ACROSS WHOLE BRAIN, LANGUAGE, VISUAL AND AUDITORY REGIONS

First, we examine the brain alignment by measuring the degree of brain predictivity using representations extracted from instruction-tuned video MLLMs, focusing on whole brain, language, visual and auditory regions. For each instruction-tuned MLLM, we calculate the average normalized brain alignment across 13 tasks, multiple subjects, and best MLLM layer, using the Movie10 fMRI dataset. Similarly, for instruction-tuned Audio MLLMs, we calculate the average normalized brain alignment across five tasks, multiple subjects, and best MLLM layer. Additionally, we report the brain alignment performance of in-context learning video MLLMs, non-instruction-tuned

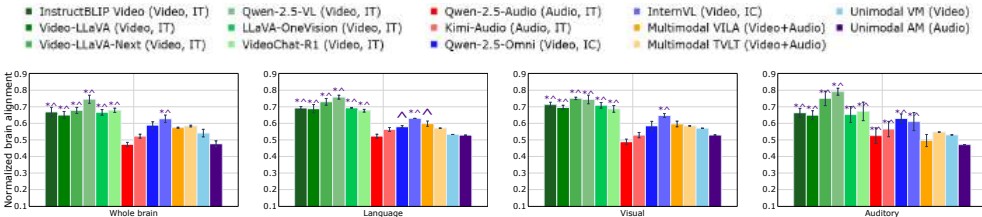

Figure 2: Average normalized brain alignment of instruction-tuned video MLLMs vs instruction-tuned audio MLLMs vs in-context learning video MLLMs vs multimodal and unimodal models across whole brain, language, visual and auditory regions. Error bars indicate the standard error of the mean across participants. ∗ implies that instruction-tuned MLLM embeddings are significantly better than multimodal models and ∧ means that instruction-tuned MLLM embeddings are significantly better unimodal models with p≤ 0.05. IT: Instruction-tuned, IC: In-context learning

multimodal models, unimodal video models, and unimodal audio model (AST). We treat the non-instruction-tuned multimodal models and unimodal models (audio and video) as the baselines when comparing against the IT-MLLMs.

**Whole brain analysis.** Fig. 2 (a) shows the results for whole brain analysis. We make the following observations: (i) At the whole-brain level, the Wilcoxon signed-rank test reveals that the differences in brain alignment between instruction-tuned video MLLMs and in-context learning models, the non-instruction-tuned multimodal and unimodal models are statistically significant. In particular, all instruction-tuned video MLLMs achieve over ∼9% improvement in brain alignment compared to in-context learning models, and ∼15% improvement compared to other baselines. This contrasts with prior findings on instruction-tuned image-based MLLMs, which demonstrated comparable performance to multimodal models when evaluated on unimodal image stimuli (Oota et al., 2025a), suggesting that instruction-tuned video MLLMs are more effective at capturing brain-relevant representations. (ii) Instruction-tuned audio MLLM embeddings show less alignment compared to non instruction-tuned multimodal and unimodal video models. These findings imply that instruction-tuned video MLLM models capture brain-relevant representations and contain additional information beyond the in-context learning, non-instruction-tuned multimodal and unimodal models.

**Language, visual and auditory region analysis.** We also present the average normalized brain alignment across language, visual and auditory regions in Fig. 2 (b, c & d). The results from Wilcoxon signed-rank test is consistent with whole-brain performance both in the language and visual regions i.e instruction-tuned video MLLMs embeddings exhibit significantly higher alignment in both language and visual regions compared to in-context learning video MLLMs, non-instruction-tuned multimodal, unimodal video, and audio models. On the other hand, instruction-tuned audio MLLM embeddings show significant alignment primarily in the auditory cortex and the middle frontal gyrus; when compared to non-instruction-tuned multimodal and unimodal models. Results for detailed language, visual and auditory sub-regions are shown in Fig. 8 and 9 in Appendix H.

These results suggest that instruction-tuned video MLLMs more effectively capture brain-relevant multimodal representations, particularly when processing naturalistic multimodal stimuli.

Additionally, we present contrast of brainmaps to display the average normalized brain alignment across voxels. Figs. 10 and 11 in Appendix I compare instruction-tuned video MLLMs with in-context learning video MLLMs (InternVL and Qwen-2.5-Omni, respectively). Figs. 12, 13, 14, 15, and 16 in Appendix J compare instruction-tuned video MLLMs with the non-instruction-tuned multimodal VILA and TVLT. The results show that instruction-tuned video MLLMs consistently achieve significantly higher alignment across all brain voxels. However, Figs. 17 & 18 in Appendix J reveal clear differences between audio MLLMs and multimodal models: the prediction performance of audio MLLMs lacks brain-relevant semantic information compared to multimodal models.

## 4.2 VIDEO AND AUDIO IT-MLLMS SUCCESSFULLY DIFFERENTIATE TASK-SPECIFIC INSTRUCTIONS

To investigate which instructions are more effective in predicting brain activity and whether IT-MLLMs differentiate task-specific representations and provide clear separation in brain regions, we analyze the voxels as follows. For each voxel, we select the instruction that results in the highest normalized brain alignment and apply the instruction-specific color code to the voxel.

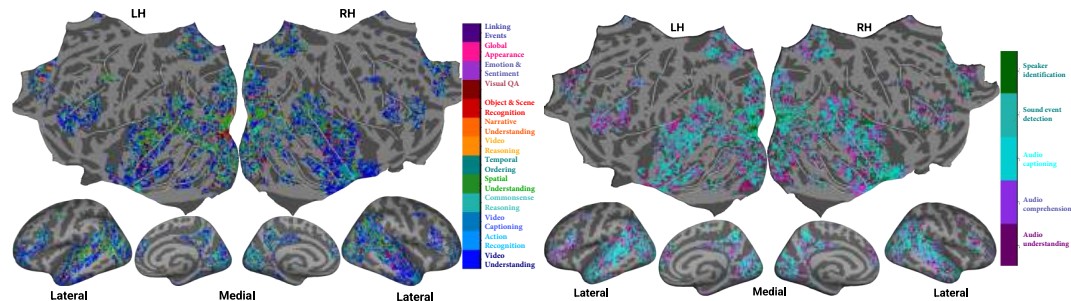

Figure 3: Each voxel is color-coded with the instruction that led to the highest normalized brain alignment. The color bar highlights color codes for each instruction. The voxels are projected onto the flattened cortical surface of the 'fsaverage' subject. (Left): video MLLM (Qwen-2.5-VL). (Right): audio MLLM (Qwen-2.5-Audio).

**Instruction-tuned video MLLMs.** Fig. 3 (left) shows brain maps for Qwen-2.5-VL for video tasks for average normalized brain predictivity across subjects where the voxel color codes are projected onto the flattened cortical surface of the 'fsaverage' subject. The color-scheme corresponding to each instruction is also reported. We make the following observations: (i) Video understanding exhibits the strongest alignment across the whole brain. (ii) Tasks such as spatial understanding, narrative understanding, and visual question answering show higher alignment in language-related regions, including the angular gyrus, posterior temporal lobe, and visual regions. (iii) Higher-order language regions in the frontal cortex are predominantly identified by the video understanding task, with a smaller proportion of voxels also activated by video reasoning and temporal ordering tasks.

These findings suggest that instruction-tuned video MLLMs not only capture modality-specific representations (e.g., visual, linguistic), but also encode task-specific instructions involving semantic integration and event structure (like video understanding). This highlights that these models can encode complex neural patterns. We observe similar performance gains in other instruction-tuned video MLLMs, flatmaps showing task-specific encoding performance for average of subjects are shown in Figs. 19 and 20 in Appendix K.

**Instruction-tuned audio MLLMs.** Fig. 3 (right) shows brainmap for Instruction-tuned audio MLLM (Qwen-2.5-Audio) where the predictions are average across subjects. The voxel color codes are projected onto the flattened cortical surface of the 'fsaverage' subject. There is a clear distinction between different audio tasks. Audio captioning and sound detection are aligned with the auditory cortex (AC), while audio understanding activates higher-level regions like the inferior temporal (IT) cortex and inferior frontal gyrus (IFG). In contrast, speaker identification shows very sparse and scattered alignment, with some unexpected activation in the primary visual cortex (V1), suggesting it does not strongly reflect brain-relevant semantic processing. Fig. 21 in Appendix K shows similar brainmap for Kimi-Audio.

**IT-MLLMs capture layer-wise cortical hierarchy.** Inspired from previous literature (Namburi et al., 2023; Mitchell et al., 2022) which shows that Transformers process information differently across layers, we examine whether IT-MLLMs reflect the brain's hierarchy of information processing across layers by analyzing the voxels as follows. For each voxel, we select the layer that results in the highest normalized brain alignment and apply a color code for the 29/33 layers for each MLLM. Fig. 4 presents brain maps for the Qwen-2.5-VL & Qwen-2.5-Audio, where the voxels with their corresponding color codes are projected onto the flattened cortical surface of the 'fsaverage' subject. We make the following observations: (i) Early sensory areas-including early visual regions and early auditory cortex-are best aligned with the lower layers of the model, suggesting that shallow model representations capture low-level sensory features. (ii) High-level visual areas such as the lateral occipital complex (LOC) and parahippocampal place area (PPA), as well as language-related regions like the superior temporal sulcus and angular gyrus, show stronger alignment with the middle to deeper layers of the model. This reflects the model's progression toward more abstract and semantically rich representations. (iii) Notably, language-related areas such as the inferior frontal gyrus (IFG), anterior temporal lobe (ATL), and angular gyrus show strongest alignment with the deepest layers of the model. These results indicate that IT-MLLMs naturally develop a layered structure that maps well onto the brain's own representational hierarchy. Similar brain maps for the remaining models are provided in Fig. 22 in Appendix L.

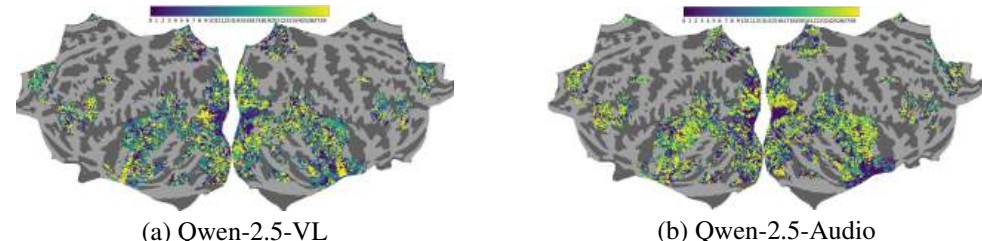

(a) Qwen-2.5-VL          (b) Qwen-2.5-Audio

Figure 4: (a) Qwen-2.5-VL and (b) Qwen-2.5-Audio (layer-wise alignment): Each voxel is color coded with the MLLM layer number (out of 29) that led to the highest normalized brain alignment. The color bar highlights color codes for each layer. The voxels are projected onto the flattened cortical surface of average across subjects on 'fsaverage' surface.

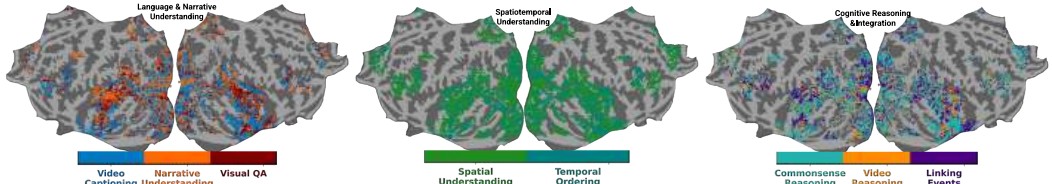

Figure 5: Semantic Task Group Analysis: Each voxel is color coded with the task instruction that led to the highest normalized brain alignment. The color bar highlights color codes for each instruction. The voxels are projected onto the flattened cortical surface averaged across all subjects for video MLLM (Qwen-2.5-VL). While this plot shows brain maps for 3 groups, brain maps for remaining 2 task groups are in Fig. 23 in Appendix M.

### 4.3 REPRESENTATIONS FROM INSTRUCTION-TUNED VIDEO MLLMS FOR SEMANTIC TASK GROUPS REVEAL DISTINCT COGNITIVE AND NEURAL PROFILES

To further examine how instruction-tuned video MLLMs generate task-specific representations and reveal functional specialization in the brain, we group the 13 video tasks into 5 cognitively grounded categories: Perceptual visual processing, Cognitive reasoning and integration, Spatiotemporal understanding, Language and narrative understanding, and Social and affective understanding. Fig. 5 illustrates that this grouping captures meaningful distinctions.

Tasks in the **Language and narrative understanding** group show broader and denser cortical engagement, particularly across the temporal and parietal cortices, compared to visual and frontal regions. In particular, we observe strong activity in the bilateral temporal lobes for narrative understanding, as well as in the angular gyrus, posterior superior temporal sulcus (pSTS), and posterior cingulate cortex (PCC) regions known to support multimodal integration, which is critical for narrative comprehension. This is aligned with previous work (Mar, 2011; Baldassano et al., 2017).

**Spatiotemporal understanding.** Temporal ordering elicits more widespread activation in the angular gyrus and posterior temporal lobe, whereas spatial understanding shows stronger engagement in the dorsal parietal cortex (part of the dorsal visual pathway) and anterior temporal lobe (Zacks et al., 2007; Baldassano et al., 2017). Additionally, we observe that early visual areas are more active during the spatial understanding task, whereas early auditory cortex shows higher activity in the temporal ordering task, likely due to its role in processing sound-based events (Belin et al., 2000). However, the brain does not strictly separate spatial and temporal processing. These representations often co-exist, particularly in narrative and event-based cognition.

**Cognitive Reasoning.** Commonsense reasoning elicits widespread activation in the temporal cortex, angular gyrus, and higher-order visual regions, reflecting its reliance on semantic processing and world knowledge. In contrast, video reasoning shows strong alignment with early visual areas (V1, V2, V3), indicating a greater dependence on visual perception and motion processing. Linking events tasks activate the early auditory cortex and show more distributed engagement of anterior temporal lobe (involved in word-level semantics), inferior frontal gyrus, and angular gyrus, highlighting the integration of temporal, linguistic, and episodic information necessary for narrative comprehension. These results show that different forms of higher-order reasoning highlights the brain's flexible organization for supporting diverse reasoning demands across modalities and timescales.

Similarly, we observe task-specific differences in brain regions for perceptual visual processing, and affective social processing (Appendix M). These patterns underscore the ability of IT-MLLMs to modulate their representations based on distinct cognitive demands reflected in the brain.

### 4.4 PARTITIONING EXPLAINED SHARED AND UNIQUE VARIANCE BETWEEN TASK-SPECIFIC INSTRUCTIONS

While the previous analysis reveals that task-specific instructions from MLLMs modulate their representations based on distinct cognitive demands, we further examine the representations of task-specific instructions to measure the overlap in brain variance explained by MLLMs. To accomplish this we use variance partitioning approach discussed in Appendix N.

Fig. 24 presents Venn diagrams for the whole brain, language and visual regions, depicting shared and unique variance across these regions between narrative understanding and other task instructions. Similarly, we show analysis for all pairs from the 13 tasks in Table 13 in Appendix N. Across nearly all task pairs, the whole brain region consistently exhibits the highest shared variance. Tasks that are conceptually or functionally related exhibit high shared variance in all regions, indicating similar cognitive processing demands. Higher-level semantic and reasoning tasks (e.g., Narrative Understanding, Commonsense Reasoning, Temporal Ordering) show increased unique variance in the language network, indicating language-specific processing distinct from visual features. High visual load tasks (e.g., Action Recognition, Object and Scene Recognition, Global Appearance) contribute more uniquely in visual cortex, especially when paired with non-visual tasks.

## 5 DISCUSSION AND CONCLUSION

Using instruction-tuned representations from both video and audio MLLMs for various task-specific instructions, we evaluated how well these representations predict fMRI brain activity when participants viewed naturalistic movies (video included with audio). Additionally, we compared different video and audio MLLMs' representations, assessing their alignment with each instruction across whole brain, language, visual and auditory regions. We show that instruction-tuned video MLLMs exhibit significantly better brain alignment than audio MLLMs, vision-only, audio-only, and non-instruction-tuned multimodal models.

Our study on IT-MLLMs and their alignment with multimodal stimuli yields several key findings: (1) Although instruction-tuned video MLLMs demonstrate strong brain alignment across the whole brain (including language, visual, and auditory regions) audio MLLMs show effective alignment primarily in auditory and language-related areas such as the middle frontal gyrus (MFG). This highlights the potential of instruction-tuned audio MLLMs to capture different features relevant to auditory processing, providing information on the function of the auditory cortex similar to those observed in previous studies (Oota et al., 2024a; 2025b). However, their performance remains comparable to non-instruction-tuned multimodal models, indicating that further improvements are needed for instruction-tuned audio MLLMs to fully capture brain-relevant representations – an effort that aligns with recent work on inducing brain-relevant biases in model design (Moussa et al., 2025; Vattikonda et al., 2025). (2) The surprising effectiveness of task-specific instructions in predicting multimodal brain activity across different regions points out that both video and audio MLLMs generate distinct task-specific representations. These representations enable the models to effectively differentiate functional processing across brain regions, unlike prior work by Oota et al. (2025a), which did not observe such differentiation when using unimodal stimuli (e.g., static images). Specifically, certain audio instructions, such as audio captioning and audio understanding, show stronger alignment with language-related regions, while tasks such as sound event detection better align with the auditory cortex and temporal lobe. These findings imply that IT-MLLMs offer a powerful framework for designing controlled stimuli by a systematic manipulation of task goals through instructions, allowing researchers to isolate and examine task-specific brain responses using the same input. (3) By grouping task-specific instructions into functional categories, we find that narrative understanding consistently engages the bilateral temporal lobes, angular gyrus, and posterior cingulate cortex which are regions known for multimodal integration. Temporal ordering tasks elicit stronger responses in the angular gyrus and posterior temporal lobe, while spatial understanding activates the dorsal parietal cortex. These findings highlight the potential of instruction-tuned video MLLMs as powerful tools for probing functional specialization in the brain, offering a structured and interpretable framework for mapping high-level cognitive processes to specific neural substrates. (4) The observed correspondence between IT-MLLM layers and the brain's functional hierarchy suggests

that these models inherently develop structured, brain-like representations, ranging from early sensory information processing in shallow layers to abstract semantic processing in deeper layers. This layered alignment not only enhances their interpretability but also highlights their potential as tools for investigating how the brain encodes and organizes complex, task-driven information.

Our findings also clearly show that despite the growing popularity of instruction-tuned video and audio MLLMs in handling generic task instructions, we are still far from fully interpreting how language-based instructions guide information flow through model layers and how fine-grained details are processed across layers to achieve brain-like representations. Future work should focus on leveraging the alignment strengths of these models using more fine-grained instruction-driven prompts, similar to controlled stimulus paradigms in neuroscience, to deepen our understanding of functional specialization in the brain. Lastly, we discuss limitations of our work in Appendix O.

## REPRODUCIBILITY STATEMENT

Both the naturalistic stimuli (movies) and the fMRI recordings used in this study are publicly available, with preprocessing steps and experimental settings described in Section 2.1 and further detailed in Appendix C. Task-specific instruction representations from instruction-tuned video and audio MLLMs, as well as in-context learning video MLLMs, are described in Section 2.3. Implementation details of voxelwise brain encoding models and evaluation metrics are provided in Section 3, with hyperparameters listed in Appendix E. To facilitate reproducibility, we release anonymized source code for all models (instruction-tuned, in-context learning, multimodal, unimodal), brain encoding, and evaluation in the supplementary zip file.

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

# Overview of Appendix Sections

## A  OVERVIEW OF MULTIMODAL MODEL EVALUATION SETTINGS IN BRAIN ENCODING STUDIES

Table 4: Overview of multimodal model evaluation settings in brain encoding studies.

| Study | Model Type | Stimulus Modality | Brain Data | Dataset | Instruction-Tuned |
|---|---|---|---|---|---|
| Doerig et al. (2022) | Vision-Language (CLIP) | Unimodal (Images) | fMRI | NSD | ✗ |
| Wang et al. (2023) | Vision-Language (CLIP) | Unimodal (Images) | fMRI | NSD | ✗ |
| Oota et al. (2022b) | Vision-Language (CLIP, VisualBERT, LXMERT) | Unimodal (Images) | fMRI | BOLD5000 | ✗ |
| Popham et al. (2021) | Vision-Only CNNs vs. Vision-Language | Unimodal (Silent Videos) | fMRI | Gallant lab short video clips | ✗ |
| Tang et al. (2022) | non-instruction-tuned multimodal model (BridgeTower) | Unimodal (Silent Videos), Unimodal (listening stories) | fMRI | Gallant lab short video clips | ✗ |
| Oota et al. (2025a) | Instruction-tuned Image+Text MLLMs | Unimodal (Images) | fMRI | NSD | ✓ |
| Sartzetaki et al. (2025) | Image Recognition models, Action recognition models | Unimodal (Visual) | fMRI | Bold Moments Dataset | ✗ |
| Nakagi et al. (2024) | Language models (BERT, GPT-2, Lllama2, OPT) | Multimodal (Videos with audio) | fMRI | 8.3 hours of video dataset | ✗ |
| Subramaniam et al. (2024) | non-instruction-tuned multimodal models (SLIP-CLIP, SimCLR, BLIP, BEIT) | Image frame-text pairs (Movies) | SEEG | AMMT | ✗ |
| Dong & Toneva (2023a) | non-instruction-tuned multimodal models (Merloreserve) | Multimodal (Movies: Videos with audio) | fMRI | Neuromod Friends dataset | ✗ |
| Oota et al. (2025b) | non-instruction-tuned multimodal models (TVLT and ImageBind) | Multimodal (Movies: Videos with audio) | fMRI | Neuromod Movie10 | ✗ |
| Our study | instruction-tuned video and audio MLLMs, in-context learning video and audio MLLMs | Multimodal (Movies: Videos with audio) | fMRI | Neuromod Movie10 | ✓ |

## B  RELATED WORK

**Brain encoding using multimodal models.** Our work is closely related to that of Conwell et al. (2022); Wang et al. (2023); Doerig et al. (2022); Tang et al. (2024); Nakagi et al. (2024); Dong & Toneva (2023b); Oota et al. (2025b), who proposed using multimodal model representations to study the contribution of brain alignment in unimodal and multimodal stimuli. The majority of brain encoding studies in using multimodal models focused on a single modality of input – vision alone (Conwell et al., 2022; Wang et al., 2023; Doerig et al., 2022; Wang et al., 2023; Tang et al., 2024; Nakagi et al., 2024). Recently, Dong & Toneva (2023b); Oota et al. (2022b) interpreted the effectiveness of multimodal Transformer language models in multimodal naturalistic stimuli. However, these studies focus on pretrained multimodal models which are not generic to tasks and lack the investigation of recent instruction-tuned models.

**Task-based brain alignment.** Our work is also closely related to that of Wang et al. (2019); Oota et al. (2022a); Aw & Toneva (2023); Sun et al. (2023) and Loong Aw et al. (2024), who propose using task-specific model representations to study the contribution of individual tasks to brain alignment. Wang et al. (2019) investigated 21 computer vision tasks to explore which vision tasks are more aligned with the brain while subjects engaged in viewing passive images. Similarly, Oota et al. (2022a) and Sun et al. (2023) explored 10 GLUE NLP tasks to study which NLP tasks are more brain-aligned during reading and listening to stories. More recent work by Loong Aw et al. (2024) uses instruction-tuned LLMs to investigate the effect of natural language instruction model representations on brain alignment across layers for language comprehension. Further, Oota et al. (2025a) use IT-MLLMs (image+text), using natural language instructions across diverse vision tasks to analyze their alignment with brain activity across layers during visual processing. However, these

studies primarily focused on unimodal stimuli and thus do not fully capture the capabilities of multimodal instruction-tuned models under multimodal conditions. We complement these works by examining the impact of a wide range of IT-MLLMs—spanning video and audio-based models with text-based prompts—on their alignment with brain activity from multimodal stimuli.

## C    DETAILED SUB-ROIS OF LANGUAGE, VISUAL AND AUDITORY REGIONS

The data covers seven brain regions of interest (ROIs) in the human brain with the following subdivisions: (i) early visual (EV: V1, V2, V3, V3B, and V4); (ii) object-related areas (LO1 and LO2); (iii) face-related areas (OFA), (iv) scene-related areas (PPA), (v) middle temporal (MT: MT, MST, LO3, FST and V3CD), (vi) late language regions, encompassing broader language regions: angular gyrus (AG: PFm, PGs, PGi, TPOJ2, TPOJ3), lateral temporal cortex (LTC: STSda, STSva, STGa, TE1a, TE2a, TGv, TGd, A5, STSdp, STSvp, PSL, STV, TPOJ1), inferior frontal gyrus (IFG: 44, 45, IFJa, IFSp) and middle frontal gyrus (MFG: 55b) (Baker et al., 2018; Milton et al., 2021; Desai et al., 2023).

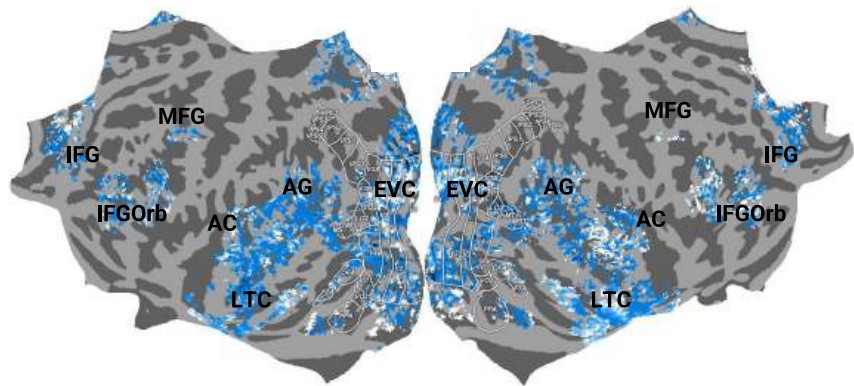

Figure 6: Flattened cortical surfaces for language-, visual- and auditory-selective regions displayed on the 'fsaverage' surface, used as the mask for all participants.

## D    CROSS-SUBJECT PREDICTION ACCURACY

We follow the method introduced by Schrimpf et al. (2021) to estimate how well brain activity in one individual can be predicted from others, using the Movie10 fMRI dataset. Starting with data from $n$ participants (e.g., $n = 4$), for each subject $s \in ([1,4])$ is chosen as the prediction target and the other three are used to predict this target, we use a voxel-wise encoding model (see Sec. 3) to predict one participant's response from others. For every combination, one participant was randomly chosen as the target, and the model was trained to predict their brain responses using data from the remaining $s-1$ participants. This gave us an average prediction score (correlation) for each voxel at each participant. To extrapolate to infinitely many humans and thus to obtain the highest possible (most conservative) estimate, as suggested by Schrimpf et al. (2021), we fit the equation $v = v_0 \times \left(1 - e^{-\frac{x}{\tau_0}}\right)$ where $x$ is each subsample's number of participants, $v$ is each subsample's correlation score and $v_0$ and $\tau_0$ are the fitted parameters. This fitting was performed for each sensor independently with 100 bootstraps each to estimate the variance where each bootstrap draws $x$ and $v$ with replacement. The final ceiling value was the median of the per-voxel ceilings $v_0$.

Fig. 7 shows the estimated cross-subject prediction accuracy for all four participants for the naturalistic movie watching. Pearson correlation scores for each voxel in each subject are projected onto the subject's flattened cortical surface. The plots show that across all subjects higher activity is observed in the language and visual regions with a max correlation up to 0.4 implying that data has low noise and low cross-subject variability.

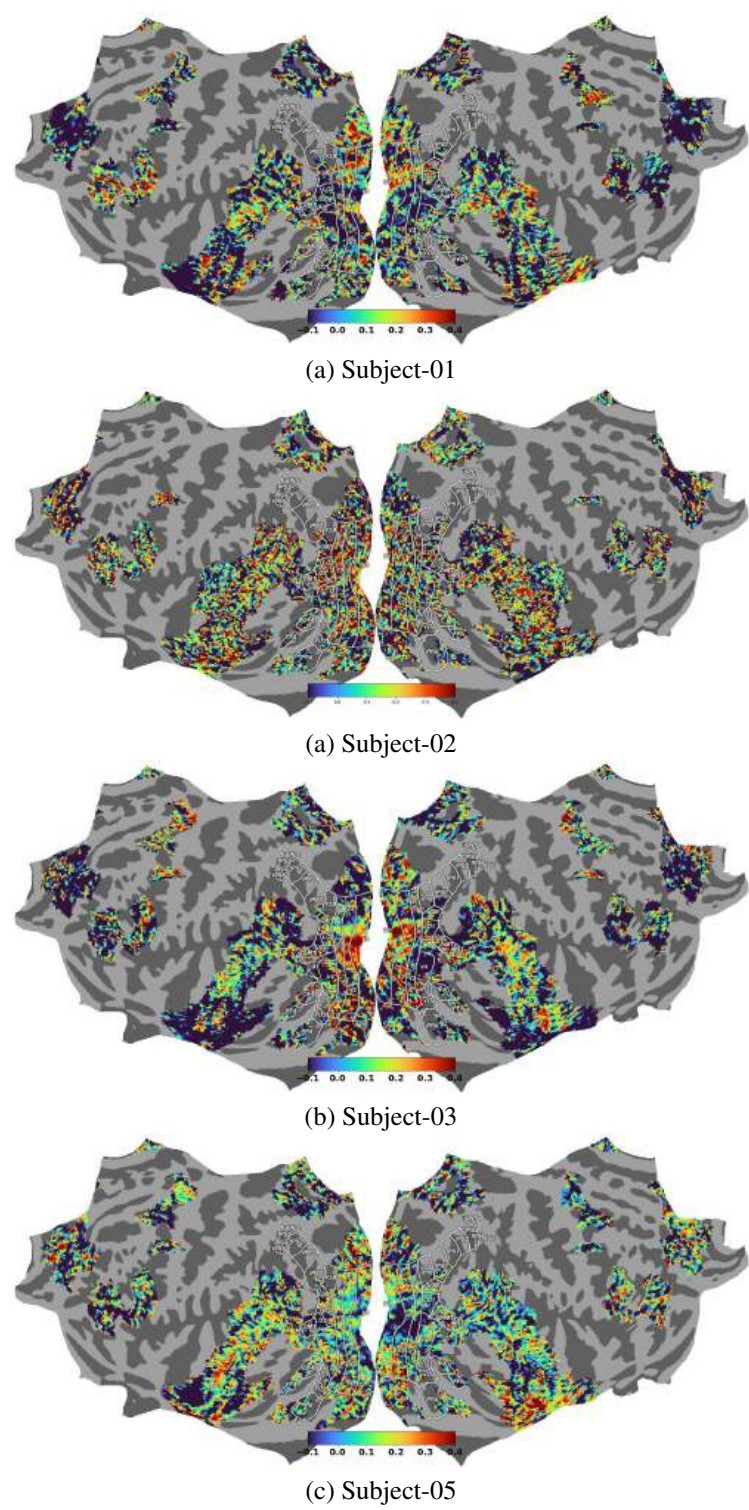

(a) Subject-01

(a) Subject-02

(b) Subject-03

(c) Subject-05

Figure 7: Estimated cross-subject prediction accuracy for all four participants for the naturalistic movie watching. Pearson correlation scores for each voxel in each subject are projected onto the subject's flattened cortical surface.

# E    IMPLEMENTATION DETAILS FOR REPRODUCIBILITY

All feature extraction experiments were conducted on a machine equipped with an NVIDIA A100 GPU with 80 GB of GPU RAM, partitioned into two devices of 40 GB each. The voxelwise encoding models were trained on NVIDIA GeForce RTX 3050 GPU with 4GB of GPU RAM. We used banded ridge-regression with the following parameters: MSE loss function; L2-decay ($\lambda$) varied from $10^{-1}$ to $10^3$; the best $\lambda$ was chosen by tuning on validation data that comprised a randomly chosen 10% subset from the train set used only for hyper-parameter tuning.

# F    STATISTICAL SIGNIFICANCE

To determine if normalized predictivity scores are significantly higher than chance, we run a permutation test using blocks of 10 contiguous fMRI TRs (considering the slowness of hemodynamic response) rather than individual TRs. By permuting predictions 5000 times, we create an empirical distribution for chance performance, from which we estimate p-value of the actual performance. The choice of these specific permutation test configurations is based on established methodologies in previous research (Deniz et al., 2019; Reddy & Wehbe, 2021; Oota et al., 2024a). To estimate the statistical significance of performance differences, such as between the model's predictions and chance or residual predictions and chance, we utilized the Wilcoxon signed-rank test (Conover, 1999), applying it to the mean normalized predictivity for the participants. Finally, the Benjamini-Hochberg False Discovery Rate (FDR) correction for multiple comparisons (Benjamini & Hochberg, 1995) is used for all the tests (appropriate because fMRI data is considered to have positive dependence (Genovese, 2000)).

# G    MODEL GENERATED OUTPUTS ACROSS INSTRUCTIONS

Tables 5, 6, 7, 8, 9 and 10 show model generated outputs for a sample video from the Movie10 dataset using InstructBLIPVideo, Qwen-2.5-VL, Video-LLaVA, LLaVa-NeXT-Video, LLaVA-OneVision and VideoChat-R1 models, respectively. Similarly, Tables 11 and 12 show model generated outputs for a sample video from the Movie10 dataset using Qwen-2.5 Audio and Kimi-Audio models.

Table 5: Outputs from InstructBLIPVideo for a sample clip from the Movie10 dataset.

| Frame 1 | Frame 3 | Frame 8 | Frame 13 | Frame 16 |

| Instruction | Model Output |
|---|---|
| Action Recognition | The main event captured in the video is a man sitting at a bar with a glass of beer, engaging in conversation with another man. The man is wearing glasses and is sitting in a dining area, surrounded by other people, possibly in a bar or restaurant. The scene also features a man wearing a towel, which is wrapped around his neck. |
| Video Understanding | In the scene, a man is sitting in a bar with a beer in his hand, engrossed in a conversation with another man. The two men are sitting at a table, with one of them holding a glass of wine. There are multiple bottles of beer visible in the background, placed on a table and around the bar area. A few other people can also be seen in the scene, possibly patrons or other bar visitors, scattered around the bar and interacting with the two main characters. The scene is set in a dimly lit environment with some lighting, providing a relaxed atmosphere for the men's conversation. |
| Visual Question Answering | There are two people in the scene, one man and one woman. They are having a conversation at a bar, sitting at a table, and drinking. |
| Video Captioning | A man with glasses sitting at a table talking with another man. |
| Object and Scene Recognition | In the scene, there is a man wearing glasses sitting in a bar, talking to another person, who is also sitting in a chair next to him. The man is holding a glass of beer, while the other person is holding a cup. |
| Commonsense Reasoning | The man is sitting in a bar with a glass of wine, engaging in conversation with another man, possibly discussing something. This action could have motivated the man to take this step for various reasons. It could be that he is seeking to socialize, share his thoughts or ideas, or simply engage in casual conversation with a friend or acquaintance. |
| Spatial Understanding | Restaurant |
| Temporal Ordering | The scene depicted in the video shows a man sitting at a bar, engaging in a conversation with another man. The man is wearing glasses and is holding a glass in his hand, which suggests that he is drinking or holding a drink. They appear to be having a casual conversation, possibly discussing something of interest or sharing a story. |
| Video reasoning | In this video, there is an unusual scene where a man with glasses is having a conversation with another man in a bar, and they are both holding glasses. This is an unusual scene because it is not a typical social setting for two men to be having a conversation in a bar. They are sitting in a dark, dimly lit environment, which adds to the unusual nature of their interaction. |
| Narrative Understanding | The central conflict in the movie is the man sitting in a bar talking to another man, who is wearing glasses. The man in the bar is having a conversation with another man, possibly discussing something of interest. It is not specified what the topic of their conversation is, but it is likely related to the movie's storyline. |
| Emotion and Sentiment Analysis | The video shows a man sitting at a bar with a friend, talking and having a conversation, which suggests a social and relaxed atmosphere. The man is wearing glasses, and the scene is set in a dimly lit bar, which creates a more intimate and cozy atmosphere. |
| Global Appearance | Man with glasses and a towel |
| Linking Events | The scene depicted in the image shows a man wearing glasses sitting at a bar, talking to another person, possibly a friend, while holding a glass of beer. This early event sets the stage for the later development of a story or a situation that may unfold in the video. |

Table 6: Outputs from Qwen-2.5-VL for a sample video from the Movie10 dataset.

| Frame 1 | Frame 3 | Frame 8 | Frame 13 | Frame 16 |

| Instruction | Model Output |
| --- | --- |
| Action Recognition | The video appears to be a still from a movie or a scene from a film. It shows two individuals engaged in a conversation in what looks like a dimly lit bar or restaurant setting. The person on the left is wearing a suit and tie, holding a drink, and seems to be listening intently. The person on the right is gesturing with their hand, possibly emphasizing a point during the conversation. The atmosphere suggests a serious or intense discussion. |
| Video Understanding | The video depicts a scene from a movie or TV show, featuring two characters engaged in a conversation. The character on the left is dressed in a formal gray suit with a patterned tie and is holding a cigarette in his right hand. He appears to be listening intently to the other character, who is partially visible from behind. The setting seems to be an indoor environment, possibly a bar or a similar establishment, as suggested by the background elements such as a mirror and what looks like a bar counter. |
| Visual Question Answering | There are two people in the scene. One person is sitting at a table, wearing a suit with a patterned tie, and appears to be engaged in a conversation or interaction with another person who is partially visible from behind. The second person is facing away from the camera, and their attire includes a plaid shirt. The setting seems to be indoors, possibly in a bar or restaurant, given the background elements like the glassware and the dim lighting. |
| Video Captioning | A man in a suit is engaged in a conversation with another person in a dimly lit bar setting. |
| Object and Scene Recognition | In the video, there are two main individuals and several objects: **1. The Man in the Suit:** He is wearing a brown suit with a white shirt and a red paisley tie. He has dark hair styled neatly and is holding a glass of what appears to be a drink in his left hand. His expression seems serious or contemplative as he looks at the other person. **2. The Other Person:** Only the back of this individual's head is visible, suggesting they are facing away. |
| Commonsense Reasoning | In the video, the character on the left appears to be in a conversation with another person whose back is turned to the camera. The character's hand is extended towards the other person, possibly gesturing or pointing at something. This action could be motivated by several reasons: **1. Clarification or Explanation:** The character might be trying to explain something important or clarify a point in their conversation. **2. Emphasis:** The gesture could be used to emphasize a particular point they are making. |
| Spatial Understanding | The video appears to be from the movie "The Wolf of Wall Street", directed by Martin Scorsese and released in 2013. The setting looks like a bar or a lounge, which is consistent with the film's depiction of the characters' activities in the financial world. However, there are no specific landmarks visible in the image that would definitively identify the exact location within the movie. The scene is set in New York City, as the film is based on the real-life events. |
| Temporal Ordering | The image depicts a scene from a movie where two characters are engaged in a conversation. Here is a step-by-step description of the activity shown: **1. Setting:** The setting appears to be an indoor environment, possibly a bar or a restaurant, given the presence of a glass and what looks like a menu board in the background. **2. Characters:** - The character on the left is wearing a suit with a patterned tie and is holding a glass in his hand. |
| Video reasoning | The video appears to be from a scene in a movie or TV show, featuring two characters engaged in a conversation. The setting looks like a bar or a similar social environment, with dim lighting and a relaxed atmosphere. What might be considered unusual is the character on the right, who seems to be gesturing with their hand while holding a drink, which could indicate they are explaining something or making a point during the conversation. However, without additional context, it's difficult to determine if there is anything particularly. |
| Narrative Understanding | The movie depicted in the video is "The Wolf of Wall Street," directed by Martin Scorsese and starring Leonardo DiCaprio as Jordan Belfort. The film is based on the memoir of the same name by Belfort himself. **Main Storyline:** The story follows Jordan Belfort, a young stockbroker who rises to become one of the most successful and corrupt figures in the financial world during the late 1980s and early 1990s. |
| Emotion and Sentiment Analysis | The video depicts two individuals in what appears to be a dimly lit, possibly upscale setting, such as a bar or restaurant. The person on the left is dressed in a suit and tie, suggesting a formal or professional demeanor, while the person on the right is wearing a plaid shirt, which might indicate a more casual or relaxed attitude. |
| Global Appearance | In the video provided, there is no indication of significant changes in the characters' appearances or outfit changes. The man on the left appears to be wearing a brown suit with a patterned tie and a white shirt. He has dark hair styled neatly. The person on the right is seen from behind, wearing a plaid shirt with a blue and red pattern. There are no visible changes in their attire or appearance within this single frame. |
| Linking Events | In the video provided, we see a scene from the movie "The Wolf of Wall Street," where Leonardo DiCaprio's character, Jordan Belfort, is engaged in a conversation with another character. The setting appears to be a dimly lit bar or lounge, which is typical for scenes involving discussions about business deals or personal matters. An early event that could influence later developments in this scene might involve a previous interaction between Jordan and his current interlocutor. |

Table 7: Outputs from Video-LLaVA for a sample clip from the Movie10 dataset.

| Frame 1 | Frame 3 | Frame 8 | Frame 13 | Frame 16 |
|---|---|---|---|---|

| Instruction | Model Output |
|---|---|
| Action Recognition | The main events in the video include a man sitting at a bar, holding a beer, and talking to another man. The man holding the beer is wearing a plaid shirt and glasses. |
| Video Understanding | The video features a man sitting at a bar with a drink in his hand. He appears to be enjoying the drink and is engaged in conversation with someone else at the bar. The man is wearing glasses and a plaid shirt, giving him a casual and relaxed demeanor. The bar is well-lit, with a variety of bottles and glasses on display, indicating a lively and bustling atmosphere. In the background, there are other patrons. |
| Visual Question Answering | There are three people in the scene. Two of them are sitting down, while the third man is standing up and holding a beer glass. |
| Video Captioning | A man in a blue and plaid shirt drinking beer at a bar. |
| Object and Scene Recognition | In the scene, the main objects visible are a man sitting at a bar with a beverage in a glass and another man sitting next to him. There are also two other men sitting at the bar, and a woman standing nearby. The bar is well-stocked with various bottles and cups. The man sitting at the bar is wearing glasses, a blue plaid shirt, and a beige blanket. |
| Commonsense Reasoning | The character in the image is holding a beer glass and appears to be in a bar or a similar setting. The motivation for the character to take this action could be due to various reasons, such as relaxation, socialization, or simply enjoying a beverage during leisure time. The character may have chosen to have a beer to unwind after a long day, to connect with others in a social setting, or simply to indulge in a favorite beverage. |
| Spatial Understanding | The video is taken from a bar, as indicated by the presence of multiple bottles on the bar and a man holding a beer glass. The video also shows the man standing in front of a bar, suggesting the location of the video is a bar or similar establishment. |
| Temporal Ordering | The video depicts a man sitting at a bar, holding a beer in his hand. He is wearing a plaid shirt and glasses. The scene is set in a dark room, and there are other people in the background. The man is seen drinking from the beer glass, which is placed on the bar counter. The camera captures the man's facial expressions and body language, highlighting his relaxed and social atmosphere. |
| Video reasoning | The unusual aspect of this video is that it features a man in a plaid shirt and glasses, who is sitting in a bar and drinking a beer. The man is also wearing a beige scarf, which adds a distinctive touch to his appearance. However, the most striking feature of this video is that the man is not only drinking a beer, but he is also holding a cell phone. |
| Narrative Understanding | The central conflict in the movie is a man's desire for a drink at a bar, but he is unable to get one due to a bartender's refusal. The man's frustration and disappointment lead to him making a scene and causing a commotion at the bar. The conflict is ultimately resolved when the man is given a drink, and he leaves the bar with a smile. |
| Emotion and Sentiment Analysis | In the video, the man in the blue shirt and the woman are seen laughing and smiling while enjoying drinks and conversation. The man in the plaid shirt is seen drinking beer and looking down, while the man in the blue shirt is holding a beer glass. The video conveys a sense of relaxation, camaraderie, and enjoyment among the characters. |
| Global Appearance | In the video, the main character, a man with a beard, is seen wearing a plaid shirt and a scarf. He is also seen wearing a different outfit in one of the scenes, which is not specified in the given information. The other man in the scene is wearing a dark shirt and glasses. There are also two other men in the background, one wearing a dark jacket and the other wearing a dark shirt. |
| Linking Events | The early event in the video is the man in the blue and white checkered shirt sitting at the bar with a beer in his hand. This scene introduces the setting and the man's presence at the bar, setting the stage for later developments. The man's interaction with other patrons, the woman, and the man in glasses adds depth and complexity to the narrative. |

Table 8: Outputs from LLaVA-NeXT-Video for a sample clip from the Movie10 dataset.

| Frame 1 | Frame 3 | Frame 8 | Frame 13 | Frame 16 |

| Instruction | Model Output |
| --- | --- |
| Action Recognition | The main event captured in the video is a conversation between two men at a bar, where one of them is wearing a red and white checkered shirt and glasses, while the other man is wearing a dark suit and tie. They are discussing a topic, possibly related to a movie, as suggested by the context. There are also other patrons in the background, and a bartender is visible in the foreground. |
| Video Understanding | The video features a scene from the movie The Hangover featuring actors Zach Galifianakis and Bradley Cooper. In the scene, Zach Galifianakis, portraying a character named Chip Baskin, is sitting at a bar with a group of people. He is seen holding a beer and appears to be in a state of disbelief as he looks on at the camera. |
| Visual Question Answering | There are several people in the scene. The man with glasses is sitting at a bar, holding a drink, and appears to be in a conversation with someone. Another man is standing at the bar, and there is a woman sitting nearby. There is a bartender standing at the bar, and a person in the background who is partially visible, but it's unclear what their role is in the scene. |
| Video Captioning | Two men are seen having a conversation in a bar, with the man in the plaid shirt appearing confused or disbelieving the man in the suit. |
| Object and Scene Recognition | In the scene, we can see a man in a suit sitting at a table with a woman and a bartender. There's also a man in a suit standing at a bar, a woman sitting at a table with a man, a woman standing at a bar, a man standing at a bar, and a man sitting at a table with a woman. |
| Commonsense Reasoning | In the video you provided, the character is seen holding up a beer bottle and making a face, which could be interpreted as a playful or humorous gesture, possibly to amuse others or to express a lighthearted or jovial moment. This action might be motivated by a desire to entertain, bond with others, or to simply enjoy a casual and relaxed moment. |
| Spatial Understanding | The video appears to have been taken from a scene in a movie, specifically "Goodfellas" featuring Robert De Niro. The scene is set in a bar, and the background includes a sign that says "The Goodfella's." This is a reference to the character Robert De Niro's character, who is a bartender in the movie. The specific setting and the sign are consistent with the movie's depiction of the bar. |
| Temporal Ordering | The video depicts two men sitting at a bar, engaged in a conversation. One man is wearing a suit, while the other is dressed in a plaid shirt. They are both holding drinks and appear to be having a casual conversation. The setting suggests they might be at a social event, such as a bar or a party, where people gather to relax and enjoy drinks. |
| Video reasoning | In the video, there is a scene where a man is seen sitting at a bar with another man. The man in glasses appears to be engaged in a conversation with someone off-camera, but the camera is focused on the other man, who is not the focus of the scene. The man with glasses is holding a beer and appears to be in the middle of a discussion, but his face is not shown in the shot. |
| Narrative Understanding | The movie "The Big Short" is a biographical comedy-drama film about the life of stockbroker Michael Burry, who predicts the financial crisis of 20008. The central conflict of the story is the housing bubble and the subprime mortgage market, which Burry sees as unsustainable and warns his colleagues about, but they dismiss his ideas as unrealistic. |
| Emotion and Sentiment Analysis | The characters in the video express a range of emotions, from surprise and shock to amusement and even a bit of confusion. The video itself is quite humorous, with the man in a suit and tie appearing to be in a state of disbelief or shock as the other man, who is wearing a plaid shirt and a hoodie, is seen drinking from his beer bottle. |
| Global Appearance | In the video, we can see a man with a beard who is wearing a suit at the beginning, but he is later seen in a more casual outfit with a plaid shirt. There are also other people in the scene, including a man with glasses and a red shirt, a man with a beard and a red shirt, and a woman in a red dress. |
| Linking Events | In the scene you've described, a man is shown sitting in a bar with a drink in front of him, looking at a TV. The TV shows a news segment about a man who has died, which is later revealed to be the man himself. This early event in the video influences the later developments because it sets up the narrative of the story, introducing the main character and establishing his presence in the scene. |

Table 9: Outputs from LLaVA-OneVision Video for a sample clip from the Movie10 dataset.

| Frame 1 | Frame 3 | Frame 8 | Frame 13 | Frame 16 |

| Instruction | Model Output |
|---|---|
| Action Recognition | The video captures a conversation between two individuals at a bar, with one person gesturing and speaking while the other listens. |
| Video Understanding | The video takes place in a lively bar setting, creating a warm and inviting atmosphere. The background features a well-lit bar counter adorned with various bottles and glasses, and several people can be seen in the background, adding to the bustling ambiance. In the foreground, two individuals are seated at a table. One person is dressed in a gray suit with a red and white patterned tie, while the other is wearing a blue and red plaid shirt with a yellow jacket draped over their shoulders. |
| Visual Question Answering | There are two people in the scene, engaged in a conversation. |
| Video Captioning | A man with curly hair and glasses sits at a table in a bar, holding a glass of beer, and engages in a conversation. |
| Object and Scene Recognition | The main objects and people visible in the scene are two men sitting at a table in a dimly lit, bustling bar or restaurant. One man is dressed in a gray suit and tie, and the other is wearing a plaid shirt with a beige sweater over it, holding a drink. The background features a bar with various bottles and glasses, and other patrons are visible in the distance. |
| Commonsense Reasoning | The character took this action to emphasize their point in the conversation, which could have been motivated by the seriousness of the discussion or the need to convey a specific message. |
| Spatial Understanding | The video is taken from a movie scene, and it shows a bar. |
| Temporal Ordering | The video takes place in a dimly lit bar with a lively background filled with people and bottles on shelves. Two individuals are seated at a table, engaged in a conversation. The person on the left is dressed in a gray suit with a red and white patterned tie and is gesturing with their hand, while the person on the right, wearing a blue and red plaid shirt with a yellow sweater draped over their shoulders and holding a glass of beer, listens attentively. |
| Video reasoning | The unusual aspect is that the scene remains static with minimal changes in the individuals' expressions and the background, suggesting a continuous conversation or interaction. |
| Narrative Understanding | The central conflict involves two characters in a bar, with the character in the plaid shirt and yellow sweater being advised or persuaded by the other character. The conflict is resolved through a conversation, leading to a change in the character's demeanor and posture. |
| Emotion and Sentiment Analysis | The characters express a range of emotions, including surprise, concern, and contemplation. The video creates a sense of intrigue and emotional depth, drawing the viewer into the characters' conversation. |
| Global Appearance | The character on the right initially has curly hair and is wearing a blue plaid shirt with a beige sweater, but later, the character's hair appears straight and they are dressed in a dark suit with a red tie. |
| Linking Events | The man in the plaid shirt and glasses is seen holding a glass of beer and engaged in a conversation with the man in the gray suit, which sets the stage for the unfolding dialogue and interaction. |

Table 10: Outputs from VideoChat-R1 for a sample clip from the Movie10 dataset.

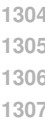 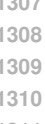

| Frame 1 | Frame 3 | Frame 8 | Frame 13 | Frame 16 |
|---|---|---|---|---|

| Instruction | Model Output |
|---|---|
| Action Recognition | The video captures a scene from what appears to be a social setting, possibly a bar or restaurant. Two individuals are engaged in conversation. One person is wearing a plaid shirt and has a blanket draped over their shoulders, suggesting they might be cold or seeking comfort. The other individual is dressed in a suit and tie, indicating a more formal attire. The background shows other patrons and bar stools, contributing to the ambiance of a casual gathering. |
| Video Understanding | The video depicts a scene from what appears to be a movie or TV show, set in a dimly lit bar or lounge. Two individuals are seated at a table, engaged in conversation. The person on the left is wearing a dark suit and tie, while the individual on the right is dressed in a plaid shirt with a yellow scarf draped over their shoulders. Both appear to be holding drinks, suggesting they might be enjoying a casual evening out. |
| Visual Question Answering | There are two people in the scene. They appear to be engaged in a conversation at a bar or restaurant. One person is wearing a plaid shirt and has a blanket draped over their shoulders, while the other person is dressed in a suit. The setting suggests a casual social interaction. |
| Video Captioning | Two individuals engage in conversation at a bar, surrounded by dim lighting and patrons. |
| Object and Scene Recognition | The video shows two individuals seated at a bar, engaged in conversation. The person on the left is wearing a dark suit and appears to be listening attentively. The person on the right is wearing a plaid shirt with a yellowish blanket draped over their shoulders and is holding a drink. The background features a bar setting with various bottles and glasses, indicating a social or casual environment. The lighting suggests an indoor setting, likely during the evening. |
| Commonsense Reasoning | In the video, one character appears to be engaged in a conversation with another character at what looks like a bar or a similar social setting. The character on the right seems to be leaning forward slightly, which might indicate that they are listening intently or trying to emphasize a point during their conversation. The motivation for this action could be: 1. **Engagement**: The character might be trying to show interest or engagement in the conversation by leaning closer. |
| Spatial Understanding | The video appears to be from a movie or TV show set in a bar or restaurant. The setting includes a bar counter with bottles and glasses, suggesting it could be a scene from a film or series that takes place in a social or dining establishment. However, without additional context or recognizable landmarks, it's difficult to pinpoint the exact location or landmark shown in the video. It could be from a variety of films or shows that feature such settings. |
| Temporal Ordering | The video depicts two individuals seated at a bar, engaged in conversation. The person on the left is wearing a suit and appears to be listening attentively. The individual on the right is dressed casually in a plaid shirt with a blanket draped over their shoulders, suggesting they might be cold or seeking comfort. They both have drinks in front of them, indicating they are likely enjoying a social setting. The background shows other patrons and bar stools, contributing to the ambiance of a lively bar environment. |
| Video reasoning | The video appears to be from a scene in a movie or TV show, likely set in a bar or similar social setting. The unusual aspect here is the presence of a large, animated hand with a glove that resembles a cartoonish character, which seems out of place in a real-world setting like a bar. This suggests it might be a special effect or a prop used for comedic or dramatic purposes within the context of the film. |
| Narrative Understanding | The video appears to be from a scene in the movie "The Hangover," where the characters are at a bar, seemingly engaged in a conversation. The central conflict in this movie revolves around a group of friends who go on a bachelor party to Las Vegas but end up getting lost and having various mishaps. |
| Emotion and Sentiment Analysis | The video depicts two individuals engaged in what appears to be an intense conversation at a bar. The person on the right seems to be speaking with a sense of urgency or concern, as indicated by their open mouth and expressive eyes. The other individual is listening attentively, suggesting a moment of serious discussion. Overall, the scene conveys a sense of tension and engagement. The setting, with its dim lighting and crowded bar, adds to the atmosphere of a private yet public interaction. |
| Global Appearance | The video shows two characters sitting at a bar. The character on the left is wearing a dark suit and tie, while the character on the right is wearing a plaid shirt with a yellow blanket draped over their shoulders. There are no significant changes in their appearances or outfits throughout the video; they remain in these respective outfits for the duration of the frames provided. |
| Linking Events | In the video, two individuals are seated at a bar, engaged in conversation. The setting suggests a casual and possibly intimate interaction, which could imply that they are discussing something personal or significant. This early event of them sitting together and talking could influence later developments by establishing a foundation for their relationship or the context of their conversation. For instance, if this is part of a larger narrative, the initial interaction might set up themes such as friendship, trust, or conflict that will unfold in subsequent scenes. |

Table 11: Outputs from Qwen-2.5 Audio for a sample audio from the Movie10 dataset (Wolf of wallstreet).

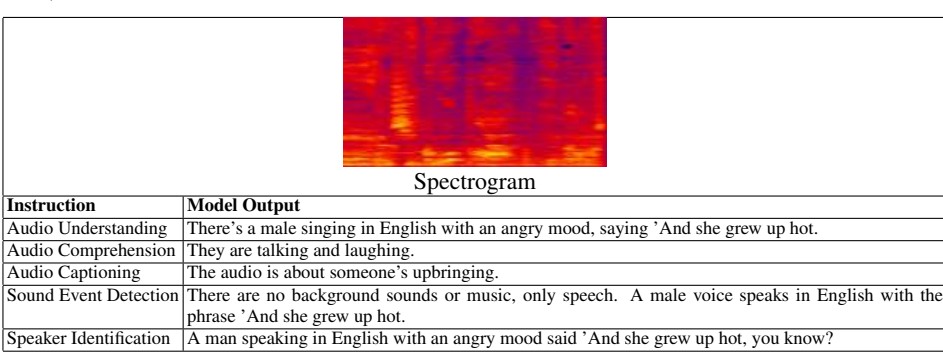

Spectrogram

| Instruction | Model Output |
| --- | --- |
| Audio Understanding | There's a male singing in English with an angry mood, saying 'And she grew up hot. |
| Audio Comprehension | They are talking and laughing. |
| Audio Captioning | The audio is about someone's upbringing. |
| Sound Event Detection | There are no background sounds or music, only speech. A male voice speaks in English with the phrase 'And she grew up hot. |
| Speaker Identification | A man speaking in English with an angry mood said 'And she grew up hot, you know? |

Table 12: Outputs from Kimi-Audio for a sample audio from the Movie10 dataset (Wolf of wallstreet)

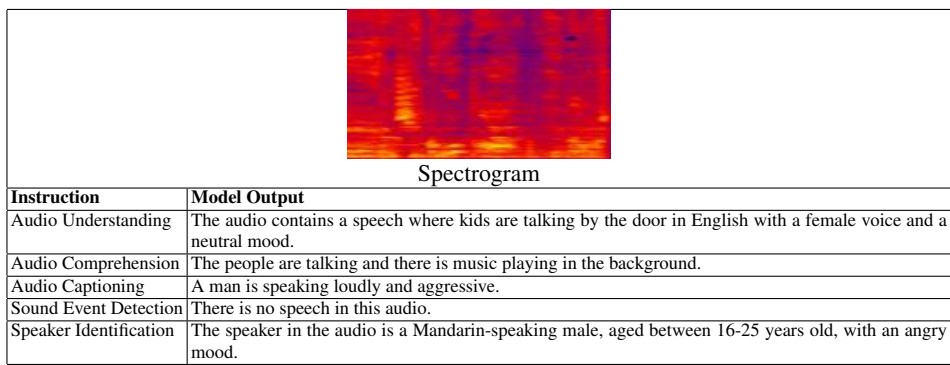

Spectrogram

| Instruction | Model Output |
| --- | --- |
| Audio Understanding | The audio contains a speech where kids are talking by the door in English with a female voice and a neutral mood. |
| Audio Comprehension | The people are talking and there is music playing in the background. |
| Audio Captioning | A man is speaking loudly and aggressive. |
| Sound Event Detection | There is no speech in this audio. |
| Speaker Identification | The speaker in the audio is a Mandarin-speaking male, aged between 16-25 years old, with an angry mood. |

# H EFFECTIVENESS OF INSTRUCTION-TUNED VIDEO MLLMs VS AUDIO MLLMs VS MULTIMODAL VS UNIMODAL REPRESENTATIONS FOR VARIOUS BRAIN REGIONS

Fig. 8 show average normalized brain alignment of instruction-tuned video MLLMs vs instruction-tuned audio MLLMs vs multimodal and unimodal models across several ROIs (AG, ATL, PTL, IFG, MFG, IFGOrb, PCC and dmPFC) of language region. Fig. 9 show the same for visual, auditory and motor regions.

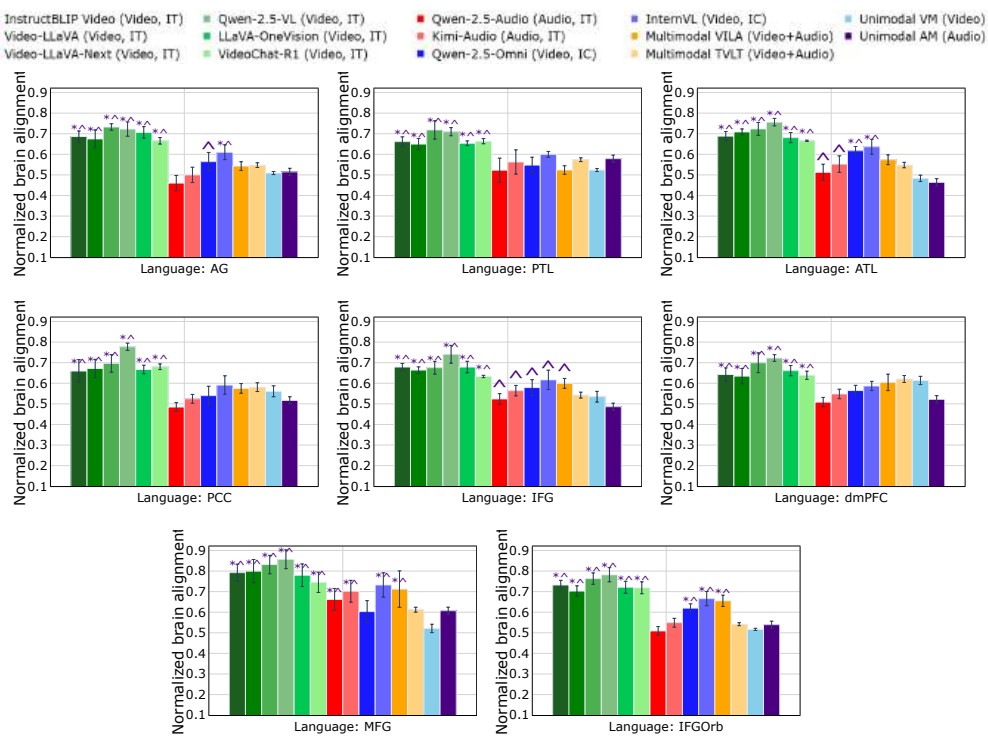

Figure 8: Average normalized brain alignment of instruction-tuned video MLLMs vs instruction-tuned audio MLLMs vs multimodal and unimodal models across several ROIs (AG, ATL, PTL, IFG, MFG, IFGOrb, PCC and dmPFC) of language region. Error bars indicate the standard error of the mean across participants. ∗ implies that instruction-tuned MLLM embeddings are significantly better than multimodal models and ∧ means that instruction-tuned MLLM embeddings are significantly better unimodal models with p≤ 0.05.

# I CONTRASTING INSTRUCTION-TUNED VIDEO MLLMs WITH IN-CONTEXT LEARNING VIDEO MLLMs

We present contrast of brainmaps to display the average normalized brain alignment across voxels for the instruction-tuned video MLLMs versus the in-context learning video MLLMs in Figures 10, and 11. The results show that instruction-tuned video MLLMs consistently achieve significantly higher alignment across all brain voxels.

# J CONTRASTING INSTRUCTION-TUNED VIDEO MLLMs WITH NON-INSTRUCTION-TUNED MULTIMODAL

We present contrast of brainmaps to display the average normalized brain alignment across voxels for the instruction-tuned video MLLMs versus the non-instruction-tuned multimodal models VILA and TVLT in Figures 12, 13, 14, 15, and 16. The results show that instruction-tuned video MLLMs consistently achieve significantly higher alignment across all brain voxels. However, Figures 17 and 18 reveal clear differences between audio MLLMs and multimodal models: the prediction per-

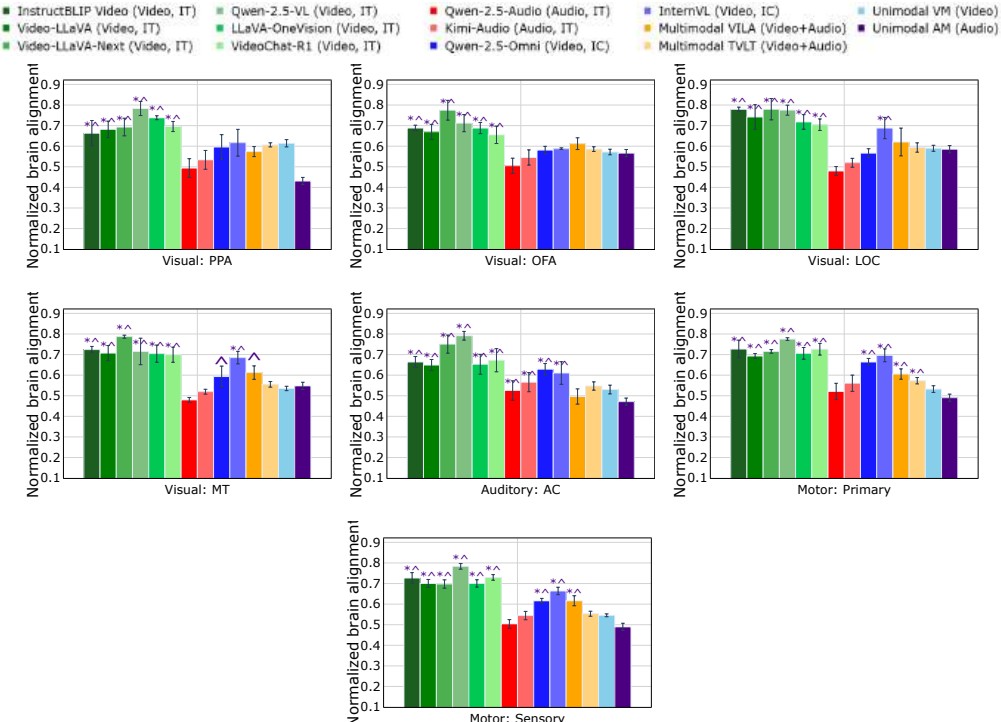

Figure 9: Average normalized brain alignment of instruction-tuned video MLLMs vs instruction-tuned audio MLLMs vs multimodal and unimodal models across several ROIs of visual cortex (PPA, OFA, LOC, MT), Auditory cortex (AC), and Motor Area (PMA and SMA). Error bars indicate the standard error of the mean across participants. ∗ implies that instruction-tuned MLLM embeddings are significantly better than multimodal models and ∧ means that instruction-tuned MLLM embeddings are significantly better unimodal models with p≤ 0.05.

formance of audio MLLMs lacks brain-relevant semantic information compared to multimodal models.

## K  BRAIN MAPS FOR TASK-SPECIFIC INSTRUCTIONS

Figures 19 and 20 show brain maps for InstructBLIPVideo, Video-LLaVA, LLaVA-NeXT-Video, LLaVA-OneVision and VideoChat-R1 for video tasks for average normalized brain predictivity across subjects where the voxel color codes are projected onto the flattened cortical surface of the 'fsaverage' subject. The color-scheme corresponding to each instruction is also reported. We make the following observations: (i) Video understanding exhibits the strongest alignment across the whole brain. (ii) Tasks such as spatial understanding, narrative understanding, and visual question answering show higher alignment in language-related regions, including the angular gyrus, posterior temporal lobe, and visual regions. (iii) Higher-order language regions in the frontal cortex are predominantly identified by the video understanding task, with a smaller proportion of voxels also activated by video reasoning and temporal ordering tasks.

Fig. 21 shows brainmap for audio instruction-tuned MLLM (Kimi-Audio) where the predictions are average across subjects. Here, the voxel color codes are projected onto the flattened cortical surface of the 'fsaverage' subject. The figure shows a clear distinction between different audio tasks.

## L  BRAIN MAPS SHOWING LAYER-WISE DETAILS FOR VIDEO INSTRUCTION-BASED MLLMS

To examine whether IT-MLLMs reflect the brain's hierarchy of information processing across layers, we analyze the voxels as follows. For each voxel, we select the layer that results in the highest normalized brain alignment and apply a color code for the 29/33 layers across the various MLLMs.

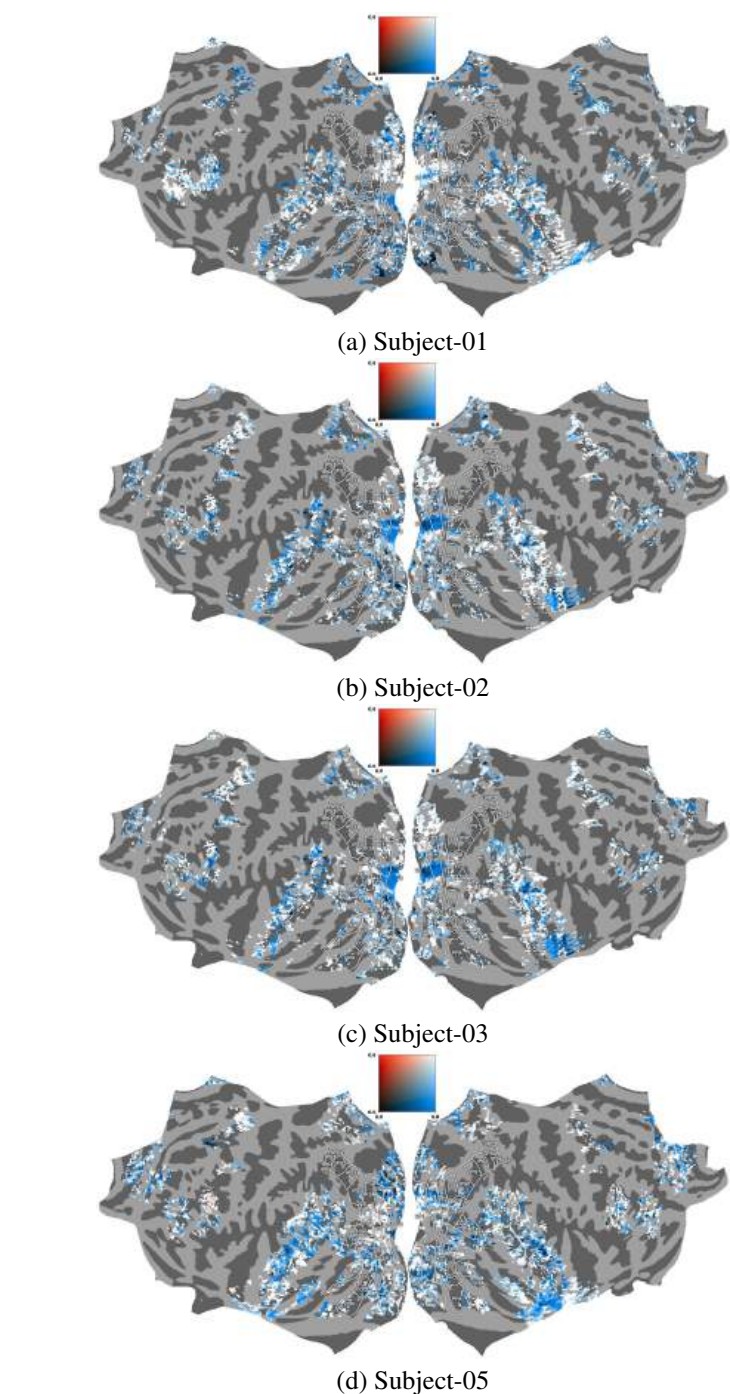

(a) Subject-01

(b) Subject-02

(c) Subject-03

(d) Subject-05

Figure 10: Qwen-2.5-VL vs.InternVL: Contrast of estimated cross-subject prediction accuracy for all participants for the naturalistic movie watching. Pearson correlation scores for each voxel in each subject are projected onto the subject's flattened cortical surface. Blue and Red voxels depict higher prediction accuracy estimates during instruction-tuned video MLLM and in-context learning video MLLM (InternVL), respectively. Voxels that have similar cross-subject prediction accuracy appear white. Here, middle frontal gyrus (MFG), inferior frontal gyrus (IFG), inferior frontal gyrus orbital (IFGOrb), angular gyrus (AG), and lateral temporal cortex (LTC) are late language regions, EVC denotes early visual cortex and AC denotes auditory cortex.

Fig. 22 presents brain maps for four video MLLMs, where the voxels with their corresponding color codes are projected onto the flattened cortical surface of the 'fsaverage' subject.

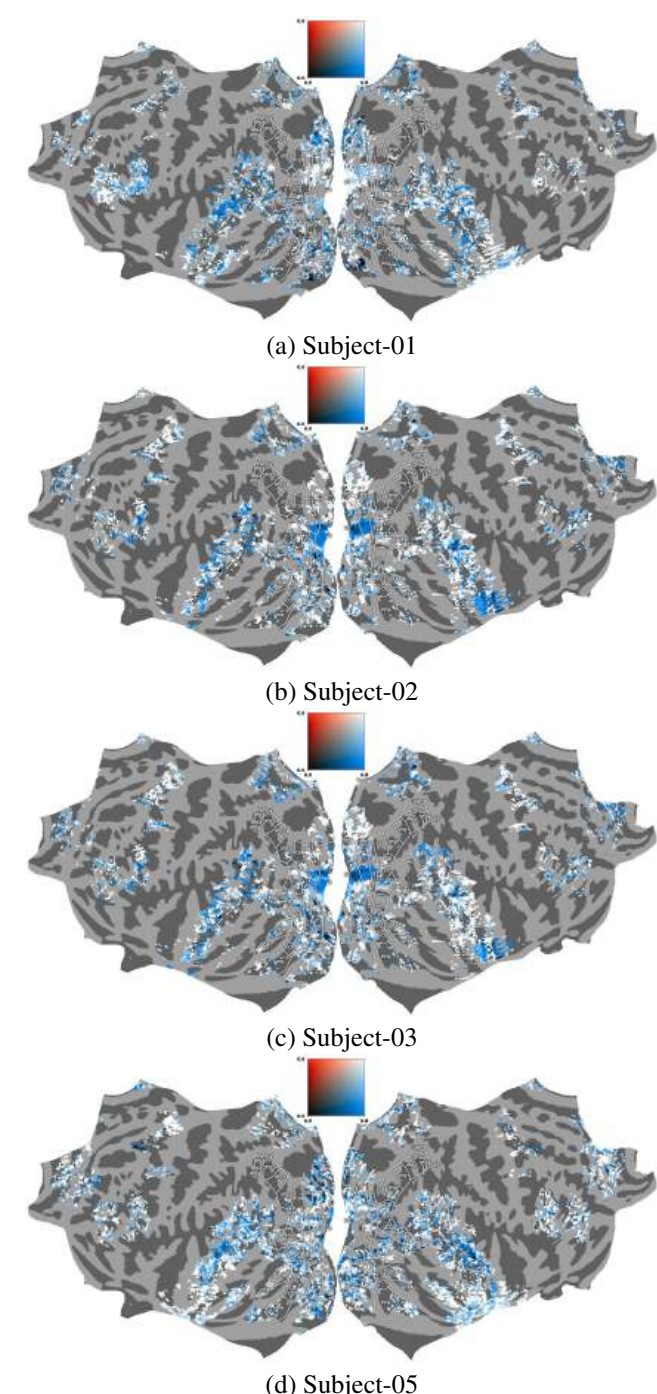

(a) Subject-01

(b) Subject-02

(c) Subject-03

(d) Subject-05

Figure 11: Qwen-2.5-VL vs.Qwen-2.5-Omni: Contrast of estimated cross-subject prediction accuracy for all participants for the naturalistic movie watching. Pearson correlation scores for each voxel in each subject are projected onto the subject's flattened cortical surface. Blue and Red voxels depict higher prediction accuracy estimates during instruction-tuned video MLLM and in-context learning video MLLM (Qwen-2.5-Omni), respectively. Voxels that have similar cross-subject prediction accuracy appear white. Here, middle frontal gyrus (MFG), inferior frontal gyrus (IFG), inferior frontal gyrus orbital (IFGOrb), angular gyrus (AG), and lateral temporal cortex (LTC) are late language regions, EVC denotes early visual cortex and AC denotes auditory cortex.

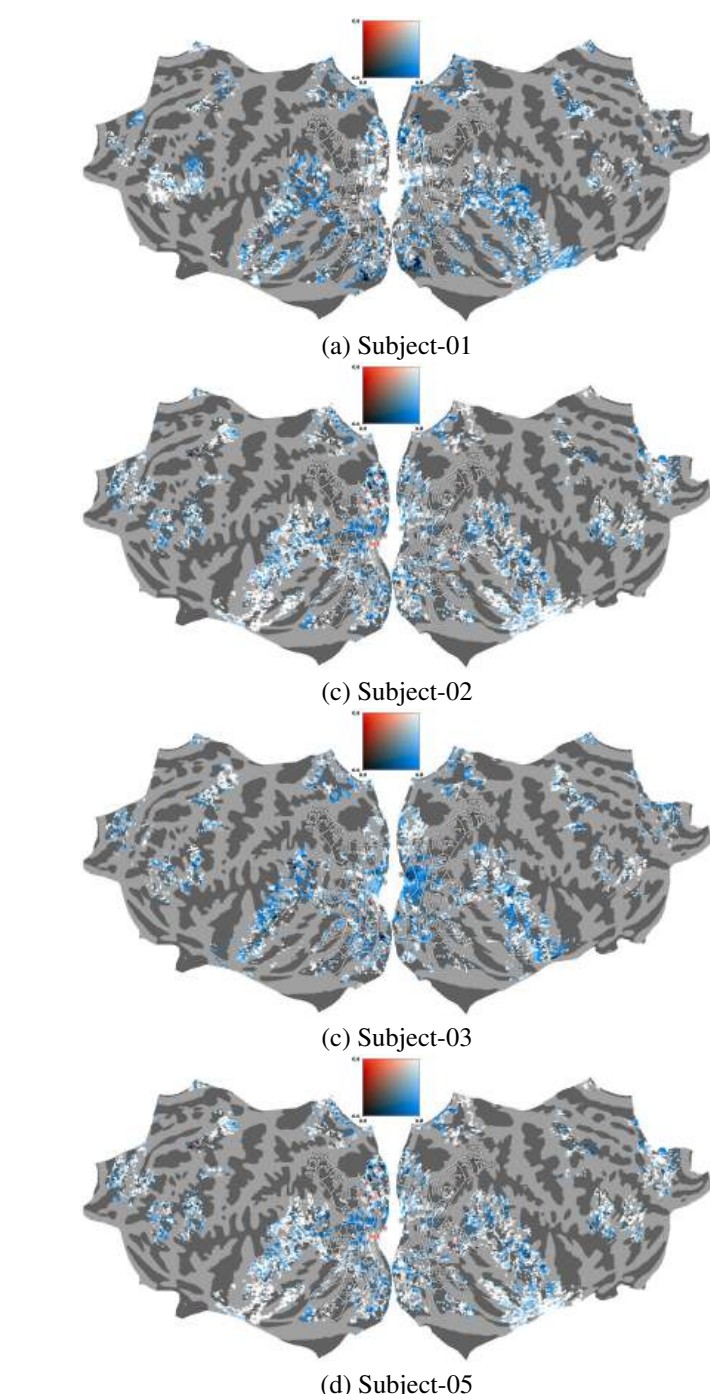

(a) Subject-01

(c) Subject-02

(c) Subject-03

(d) Subject-05

Figure 12: Qwen-2.5-VL vs. VILA: Contrast of estimated cross-subject prediction accuracy for all participants for the naturalistic movie watching. Pearson correlation scores for each voxel in each subject are projected onto the subject's flattened cortical surface. Blue and Red voxels depict higher prediction accuracy estimates during instruction-tuned video MLLM and multimodal VILA, respectively. Voxels that have similar cross-subject prediction accuracy appear white. Here, middle frontal gyrus (MFG), inferior frontal gyrus (IFG), inferior frontal gyrus orbital (IFGOrb), angular gyrus (AG), and lateral temporal cortex (LTC) are late language regions, EVC denotes early visual cortex and AC denotes auditory cortex.

## M    DETAILS OF SEMANTIC TASK GROUP ANALYSIS

To further examine how instruction-tuned video MLLMs generate task-specific representations and reveal functional specialization in the brain, we group the 13 video tasks into five cognitively

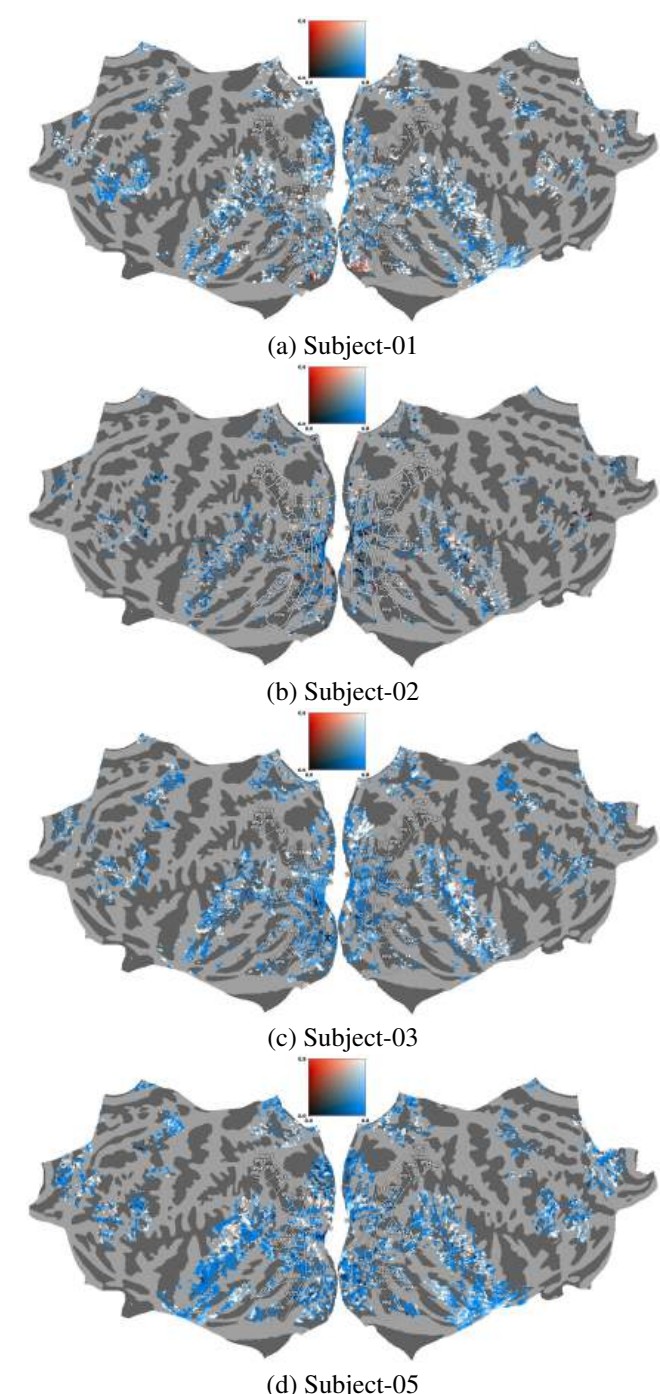

(a) Subject-01

(b) Subject-02

(c) Subject-03

(d) Subject-05

Figure 13: Qwen-2.5-VL vs. TVLT: Contrast of estimated cross-subject prediction accuracy for all participants for the naturalistic movie watching. Pearson correlation scores for each voxel in each subject are projected onto the subject's flattened cortical surface. Blue and Red voxels depict higher prediction accuracy estimates during instruction-tuned video MLLM and multimodal TVLT, respectively. Voxels that have similar cross-subject prediction accuracy appear white. Here, middle frontal gyrus (MFG), inferior frontal gyrus (IFG), inferior frontal gyrus orbital (IFGOrb), angular gyrus (AG), and lateral temporal cortex (LTC) are late language regions, EVC denotes early visual cortex and AC denotes auditory cortex.

grounded categories: Perceptual visual processing, Cognitive reasoning and integration, Spatiotemporal understanding, High-level language and narrative understanding, and Social and affective un-

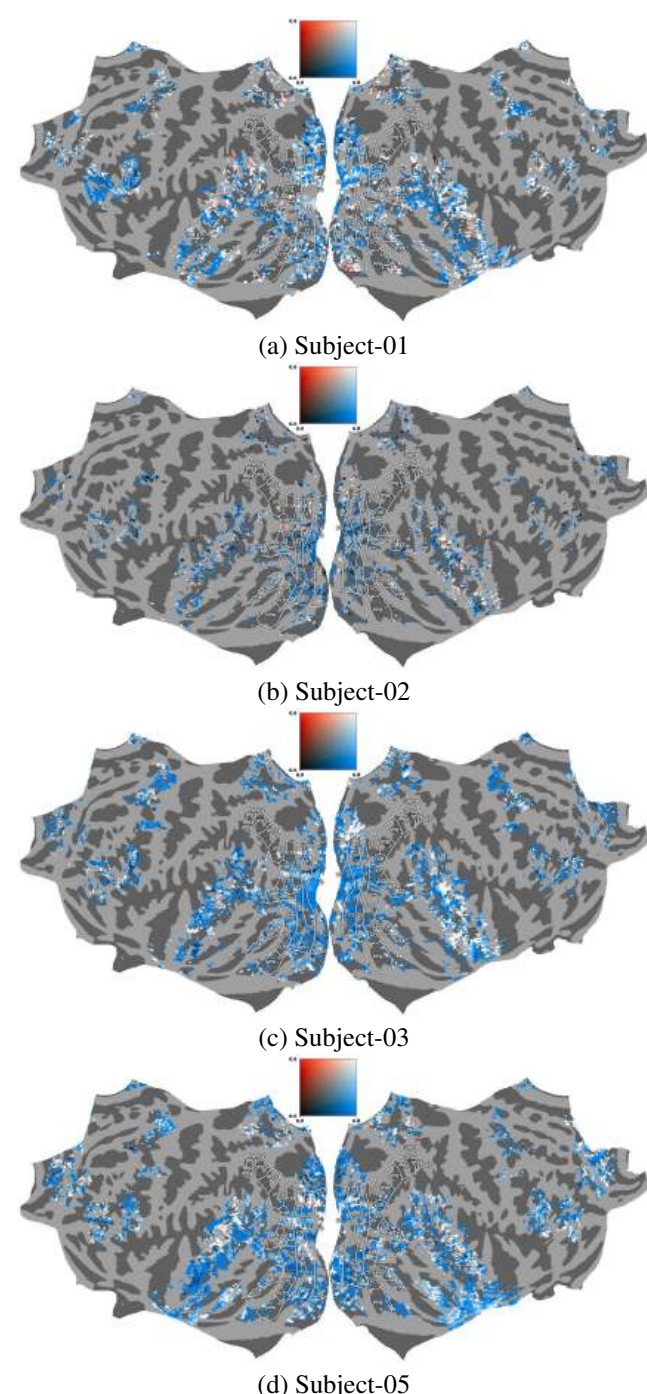

(a) Subject-01

(b) Subject-02

(c) Subject-03

(d) Subject-05

Figure 14: InstructBLIPVideo vs. TVLT: Contrast of estimated cross-subject prediction accuracy for all participants for the naturalistic movie watching. Pearson correlation scores for each voxel in each subject are projected onto the subject's flattened cortical surface. Blue and Red voxels depict higher prediction accuracy estimates during instruction-tuned video MLLM and multimodal TVLT, respectively. Voxels that have similar cross-subject prediction accuracy appear white.

derstanding. This categorization allows us to disentangle the functional specificity of brain regions engaged by different task types. The visualizations in Fig. 5 in Section 4.3 in the main paper and Fig. 23 illustrate that this grouping captures meaningful distinctions.

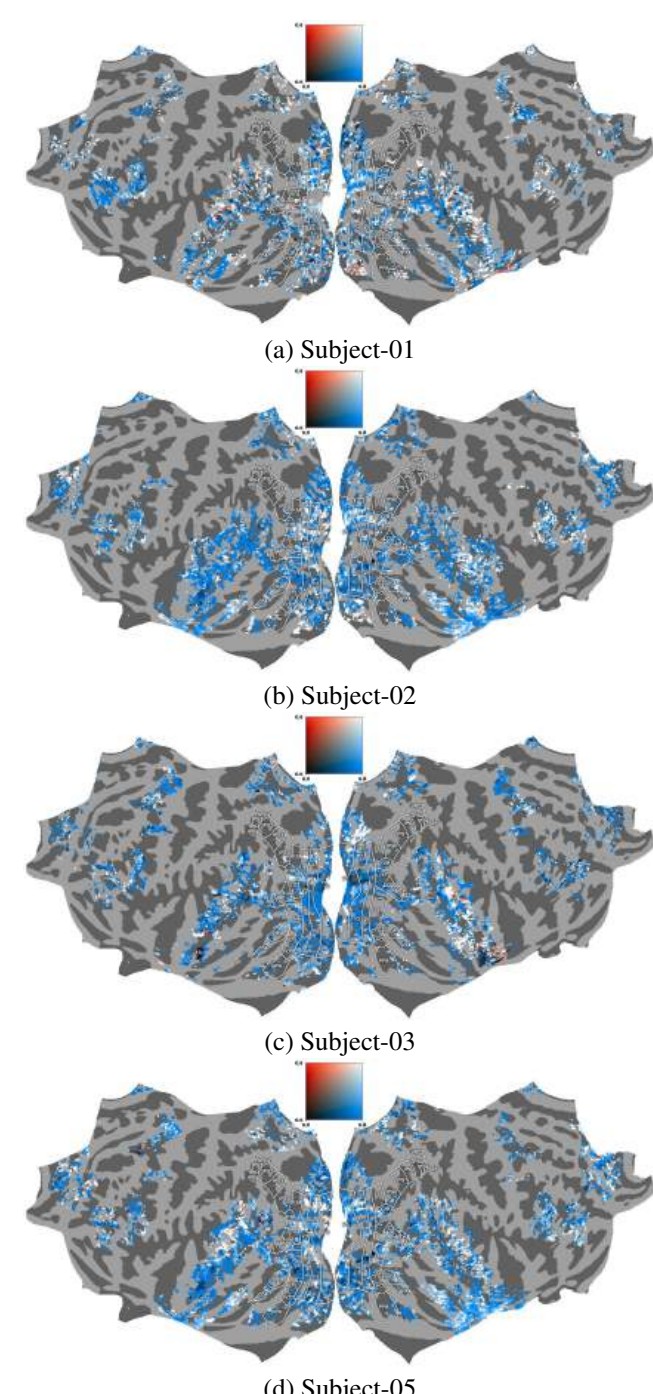

(a) Subject-01

(b) Subject-02

(c) Subject-03

(d) Subject-05

Figure 15: Video-LLaVA vs. TVLT: Contrast of estimated cross-subject prediction accuracy for all participants for the naturalistic movie watching. Pearson correlation scores for each voxel in each subject are projected onto the subject's flattened cortical surface. Blue and Red voxels depict higher prediction accuracy estimates during instruction-tuned video MLLM and multimodal TVLT, respectively. Voxels that have similar cross-subject prediction accuracy appear white.

## N    DETAILS OF EXPLAINED VARIANCE PARTITIONING

**Variance partitioning.** To disentangle task-specific instruction representations from multimodal instruction-tuned models, we used a variance partitioning approach (de Heer et al., 2017; LeBel et al., 2021). This method measures the overlap in brain variance explained by different task-specific

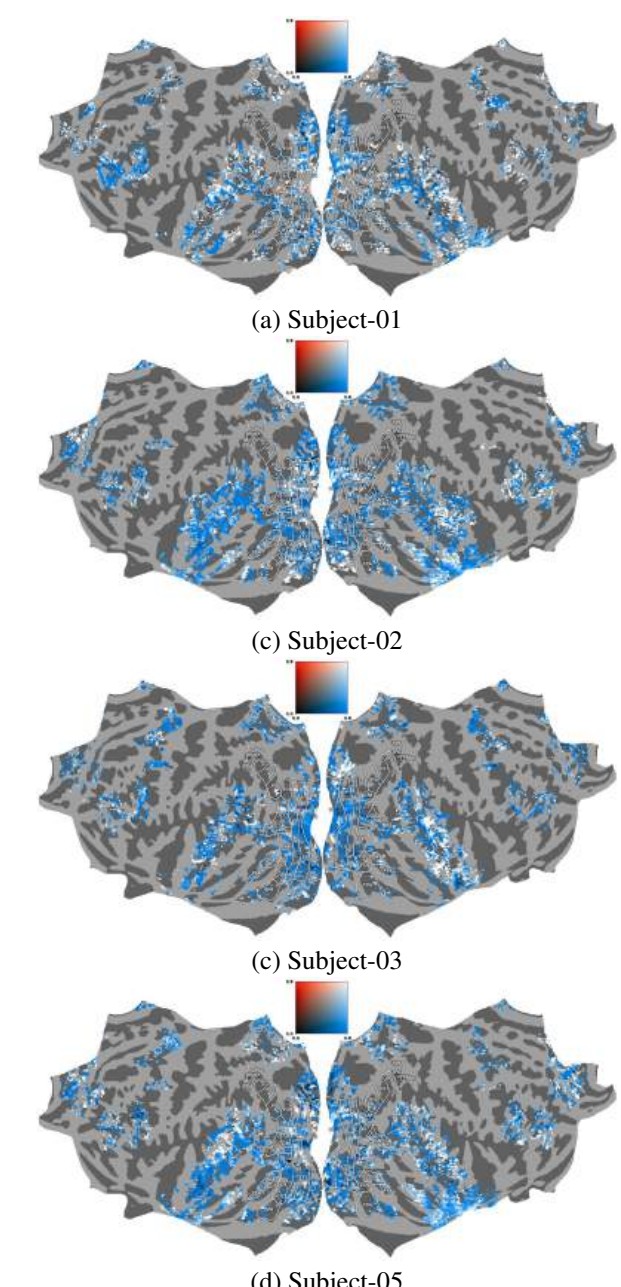

(a) Subject-01

(c) Subject-02

(c) Subject-03

(d) Subject-05

Figure 16: LLaVA-NeXT-Video vs. TVLT: Contrast of estimated cross-subject prediction accuracy for all participants for the naturalistic movie watching. Pearson correlation scores for each voxel in each subject are projected onto the subject's flattened cortical surface. Blue and Red voxels depict higher prediction accuracy estimates during instruction-tuned video MLLM and multimodal TVLT, respectively. Voxels that have similar cross-subject prediction accuracy appear white.

instruction representations. Specifically, variance partitioning separates the brain response variance that can be attributed to two models based on their unique and overlapping contributions (Vaidya et al., 2022; Deniz et al., 2019). To perform this, for every pair of instruction representations, we fit separate encoding models for each space as well as a joint encoding model, obtained by concatenating the features. Using set arithmetic, we can then derive the size of the intersection $(NBA)_v^{1 \cap 2}=(NBA)_v^1+(NBA)_v^2-(NBA)_v^{1 \cup 2}$, where NBA refers to normalized brain alignment, $v$ refers to a specific voxel, $(NBA)_v^1$ denotes alignment of model 1, $(NBA)_v^2$ denotes alignment of model 2 and $(NBA)_v^{1 \cup 2}$ denotes alignment of the joint model. Similarly, the unique contribution of model 1's feature space is computed as $(NBA)_v^{1 \setminus 2}=(NBA)_v^1-(NBA)_v^{1 \cap 2}$.

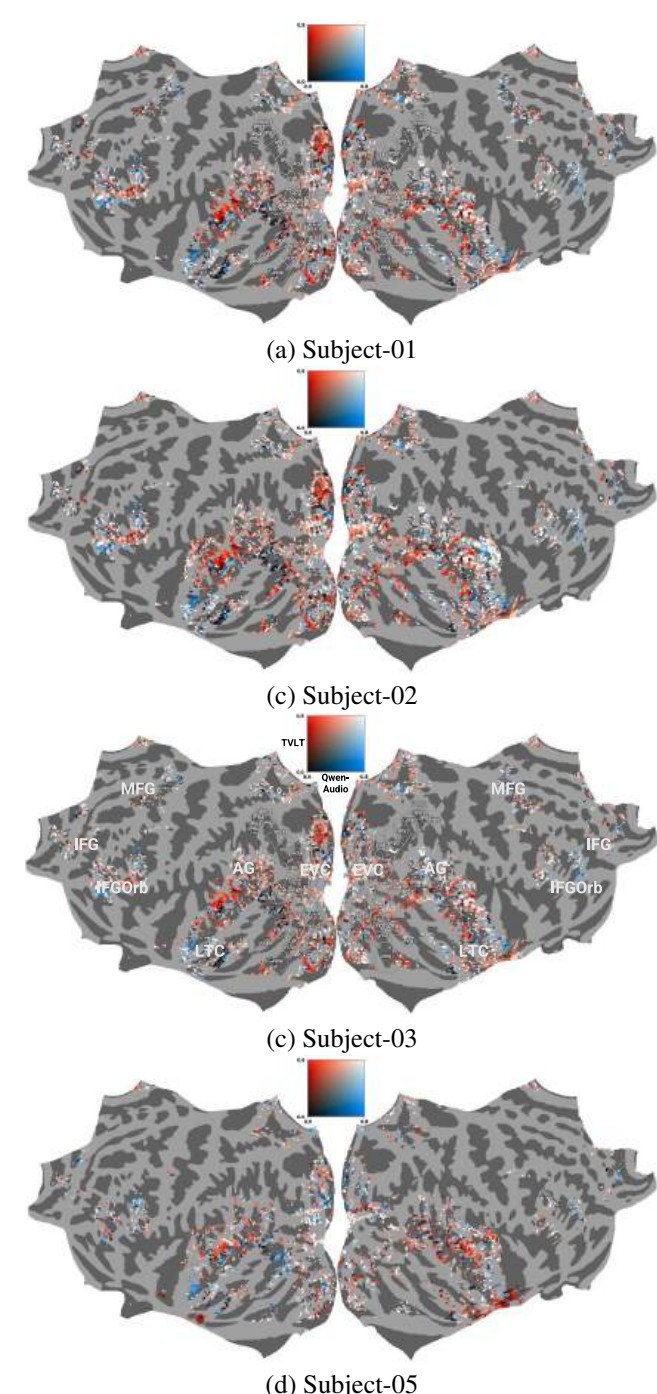

(a) Subject-01

(c) Subject-02

(c) Subject-03

(d) Subject-05

Figure 17: Qwen-Audio vs. TVLT: Contrast of estimated cross-subject prediction accuracy for all participants for the naturalistic movie watching. Pearson correlation scores for each voxel in each subject are projected onto the subject's flattened cortical surface. Blue and Red voxels depict higher prediction accuracy estimates during instruction-tuned audio MLLM and multimodal TVLT, respectively. Voxels that have similar cross-subject prediction accuracy appear white. Here, middle frontal gyrus (MFG), inferior frontal gyrus (IFG), inferior frontal gyrus orbital (IFGOrb), angular gyrus (AG), and lateral temporal cortex (LTC) are late language regions, EVC denotes early visual cortex and AC denotes auditory cortex.

**Shared and Unique Variance between Narrative Understanding and Remaining Task Instructions**

Fig. 24 shows the shared variance of the Narrative Understanding task with other video tasks for Qwen-2.5-VL.

Table 13 presents shared and unique variance explained by pairs of video tasks using brain-informed models across three neural regions: whole brain, visual cortex, and language network. The results are averaged across subjects and show how well representations from each task pair align with brain activity in specific regions.

Key Observations are as follows.

- **Whole Brain Shows Dominant Shared Variance:** Across nearly all task pairs, the whole brain region consistently exhibits the highest shared variance (often >80% in early task pairs). For example, the pair Action Recognition and Video Understanding (1–2) shows 90.69% shared variance, with very little unique variance from either task. This suggests high redundancy and common processing across tasks when considering global brain activity.

- **Visual and Language Regions Yield More Balanced Partitioning:** In contrast, visual and language-selective voxels exhibit lower shared variance and comparatively higher unique contributions from individual tasks. For the same task pair (1–2), shared variance in visual is 72.05%, and in language it is 77.46%, with higher unique components (∼10-14%). This suggests that fine-grained processing differences are more pronounced in modality-specific regions.

- **Task Similarity Reflects in Shared Variance:** Tasks that are conceptually or functionally related (e.g., Narrative Understanding-Linking Events (10-13) or Emotion and Sentiment Analysis-Linking Events (11-13)) exhibit high shared variance in all regions, indicating similar cognitive processing demands. Conversely, task pairs with less conceptual overlap (e.g., Object Recognition-Commonsense Reasoning (5-6) or Visual QA-Object Recognition (3-5)) show lower shared variance and higher unique variance, especially in language and visual regions.

- **Language Regions Show Selectivity for High-Level Tasks:** Higher-level semantic and reasoning tasks (e.g., Narrative Understanding, Commonsense Reasoning, Temporal Ordering) show increased unique variance in the language network, indicating language-specific processing distinct from visual features. For instance, pair 6-13 (Commonsense Reasoning-Linking Events) yields 16.75% unique variance for Linking Events in the language network.

- **Visual Cortex Captures Scene and Action Differentiation:** Tasks with high visual load (e.g., Action Recognition, Object and Scene Recognition, Global Appearance) contribute more uniquely in the visual cortex, especially when paired with non-visual tasks.

## O    LIMITATIONS

One possible limitation of our study lies in interpreting the differences in brain alignment between instruction-tuned video and audio MLLMs. The models we evaluate differ in several aspects, including the amount of training data and the specific objective functions used during training. To address this concern, we evaluated multiple models of each type, spanning a range of training objectives and dataset sizes, and found that our key results generalize within both video and audio MLLM categories. Still, it is possible that some of the differences in brain alignment may still be influenced by confounding factors related to model architecture, training objectives, or data scale. Future work should explore these questions using models that are more tightly controlled across these dimensions.

## P    LLM USAGE

We used OpenAI ChatGPT for grammar correction and language polishing.

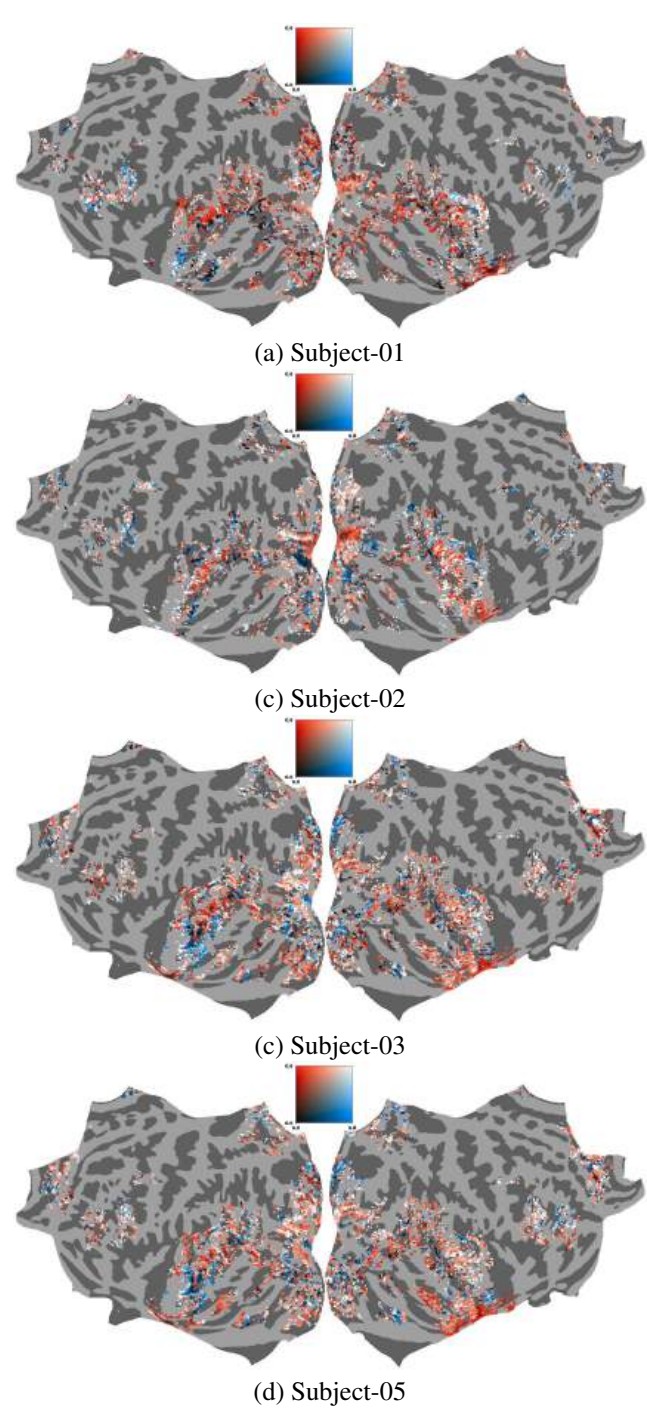

(a) Subject-01

(c) Subject-02

(c) Subject-03

(d) Subject-05

Figure 18: Kimi-Audio vs. TVLT: Contrast of estimated cross-subject prediction accuracy for all participants for the naturalistic movie watching. Pearson correlation scores for each voxel in each subject are projected onto the subject's flattened cortical surface. Blue and Red voxels depict higher prediction accuracy estimates during instruction-tuned audio MLLM and multimodal TVLT, respectively. Voxels that have similar cross-subject prediction accuracy appear white. Here, middle frontal gyrus (MFG), inferior frontal gyrus (IFG), inferior frontal gyrus orbital (IFGOrb), angular gyrus (AG), and lateral temporal cortex (LTC) are late language regions, EVC denotes early visual cortex and AC denotes auditory cortex.

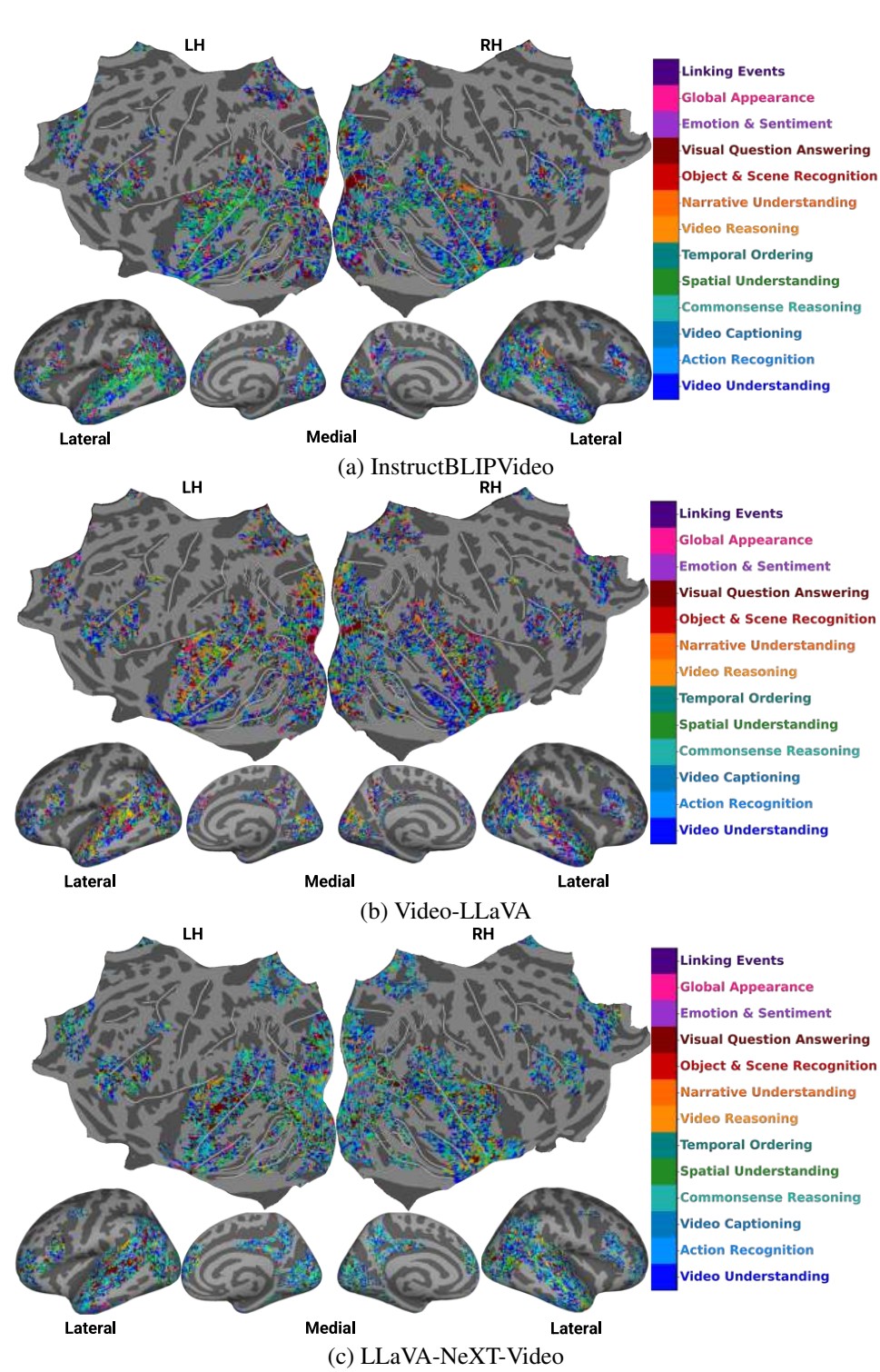

Figure 19: Each voxel is color coded with the instruction (out of 13) that led to the highest normalized brain alignment. The color bar highlights color codes for each instruction. The voxels are projected onto the flattened cortical surface averaged across all 4 subjects for 3 video MLLM (InstructBLIPVideo, Video-LLaVA and LLaVA-NeXT-Video).

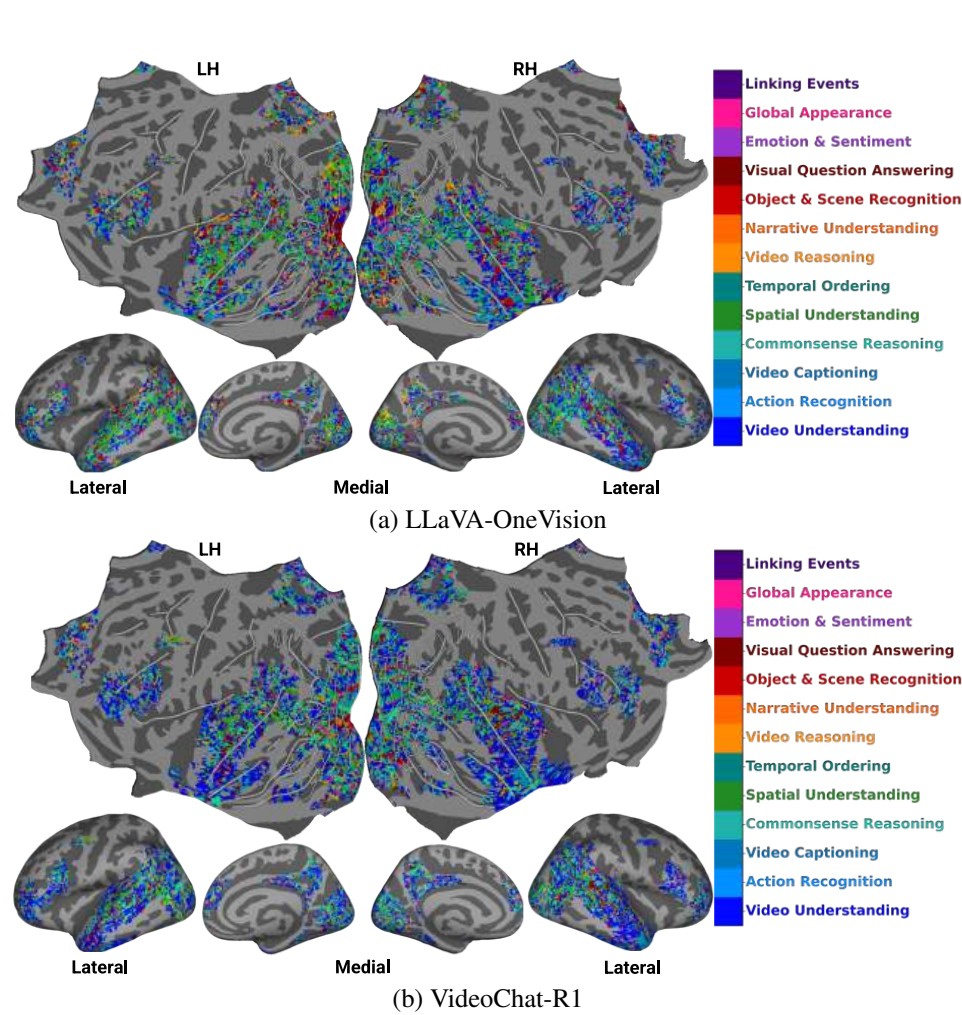

(a) LLaVA-OneVision

(b) VideoChat-R1

Figure 20: Each voxel is color coded with the instruction (out of 13) that led to the highest normalized brain alignment. The color bar highlights color codes for each instruction. The voxels are projected onto the flattened cortical surface averaged across all 4 subjects for 2 video MLLM (LLaVA-OneVision, VideoChat-R1).

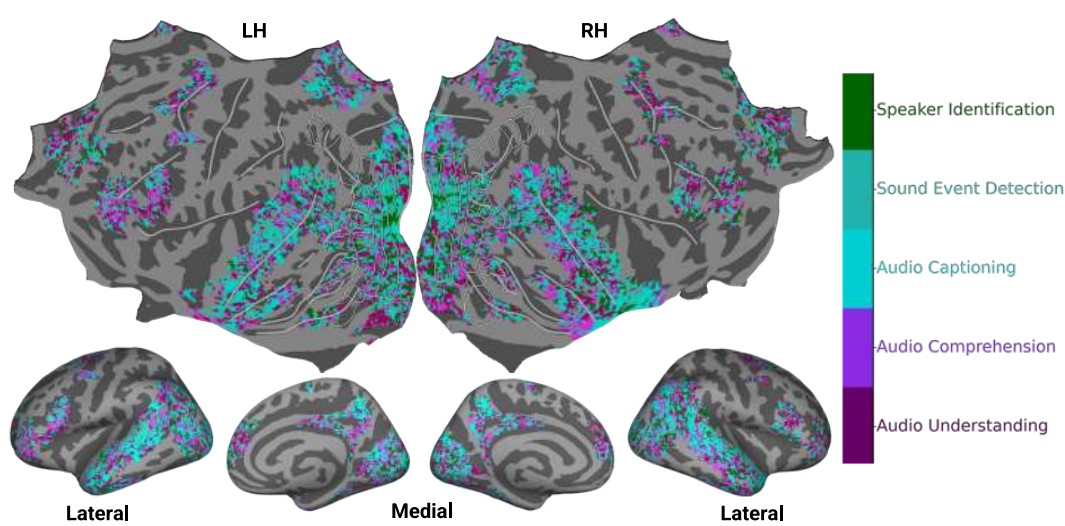

Figure 21: Kimi-Audio: Each voxel is color-coded with the instruction (out of 5) that led to the highest normalized brain alignment. The color bar highlights color codes for each instruction. The voxels are projected onto the flattened cortical surface of average across subjects on 'fsaverage' surface.

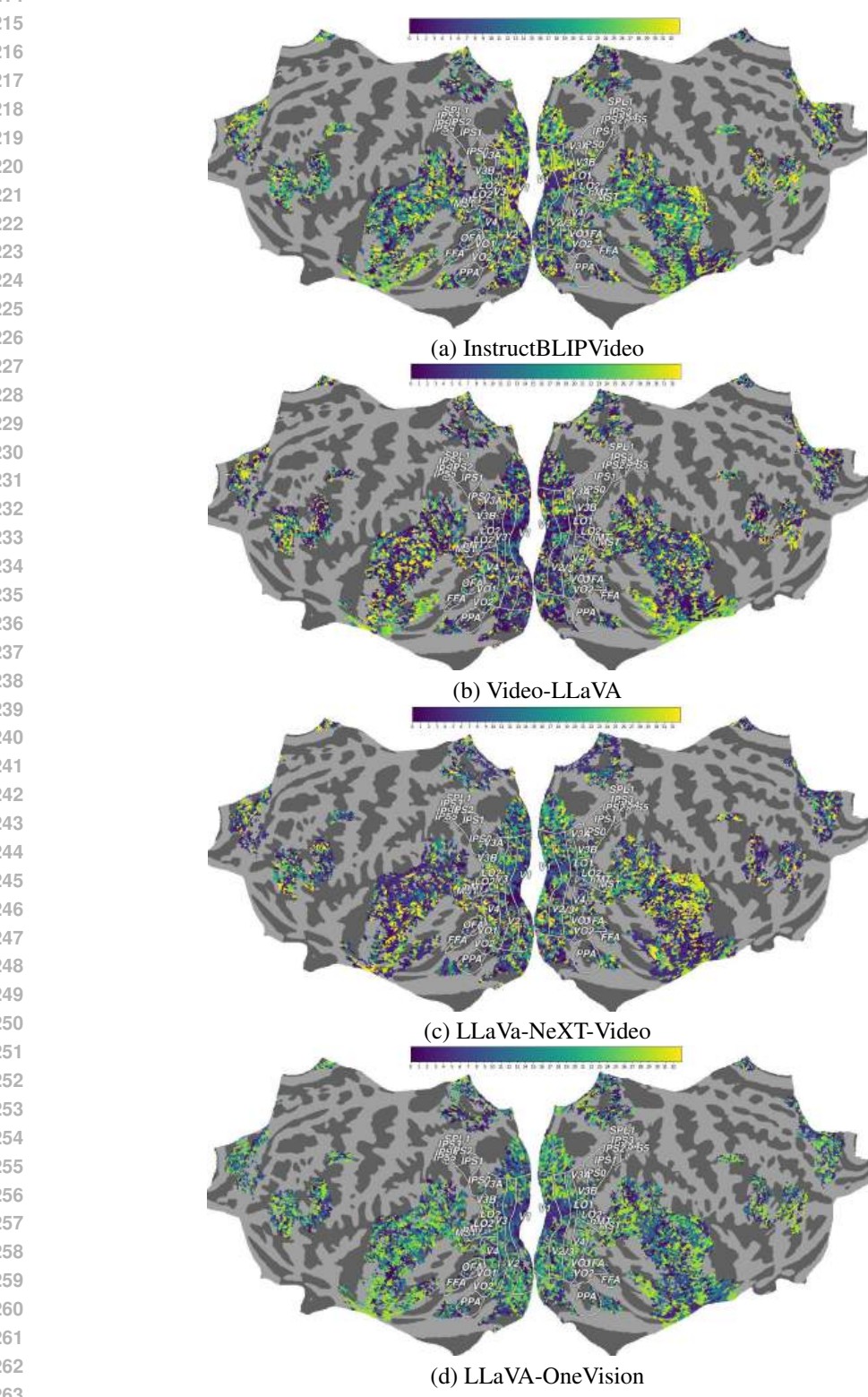

(a) InstructBLIPVideo

(b) Video-LLaVA

(c) LLaVa-NeXT-Video

(d) LLaVA-OneVision

Figure 22: Each voxel is color coded with the video MLLM layer number (out of 33) that led to the highest normalized brain alignment. The color bar highlights color codes for each layer. The voxels are projected onto the flattened cortical surface of average across all 4 subjects on 'fsaverage' surface for four MLLMs.

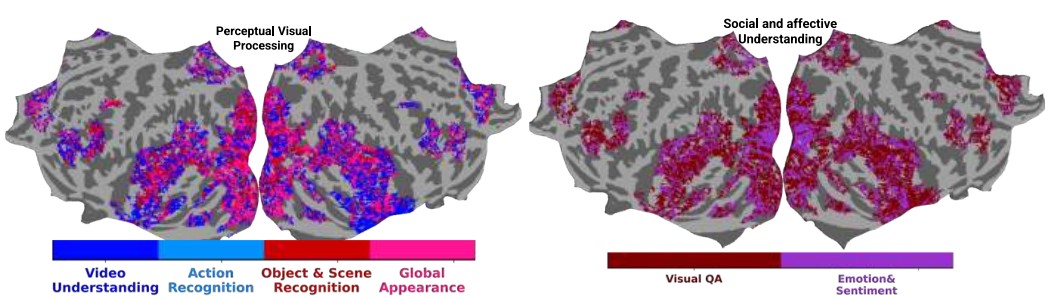

Figure 23: Semantic Task Group Analysis: Each voxel is color coded with the task instruction that led to the highest normalized brain alignment. The color bar highlights color codes for each instruction. The voxels are projected onto the flattened cortical surface averaged across all subjects for video MLLM (Qwen-2.5-VL). While this plot shows brain maps for 2 groups, brain maps for remaining 3 task groups are in Fig. 5 in Section 4.3 in the main paper.

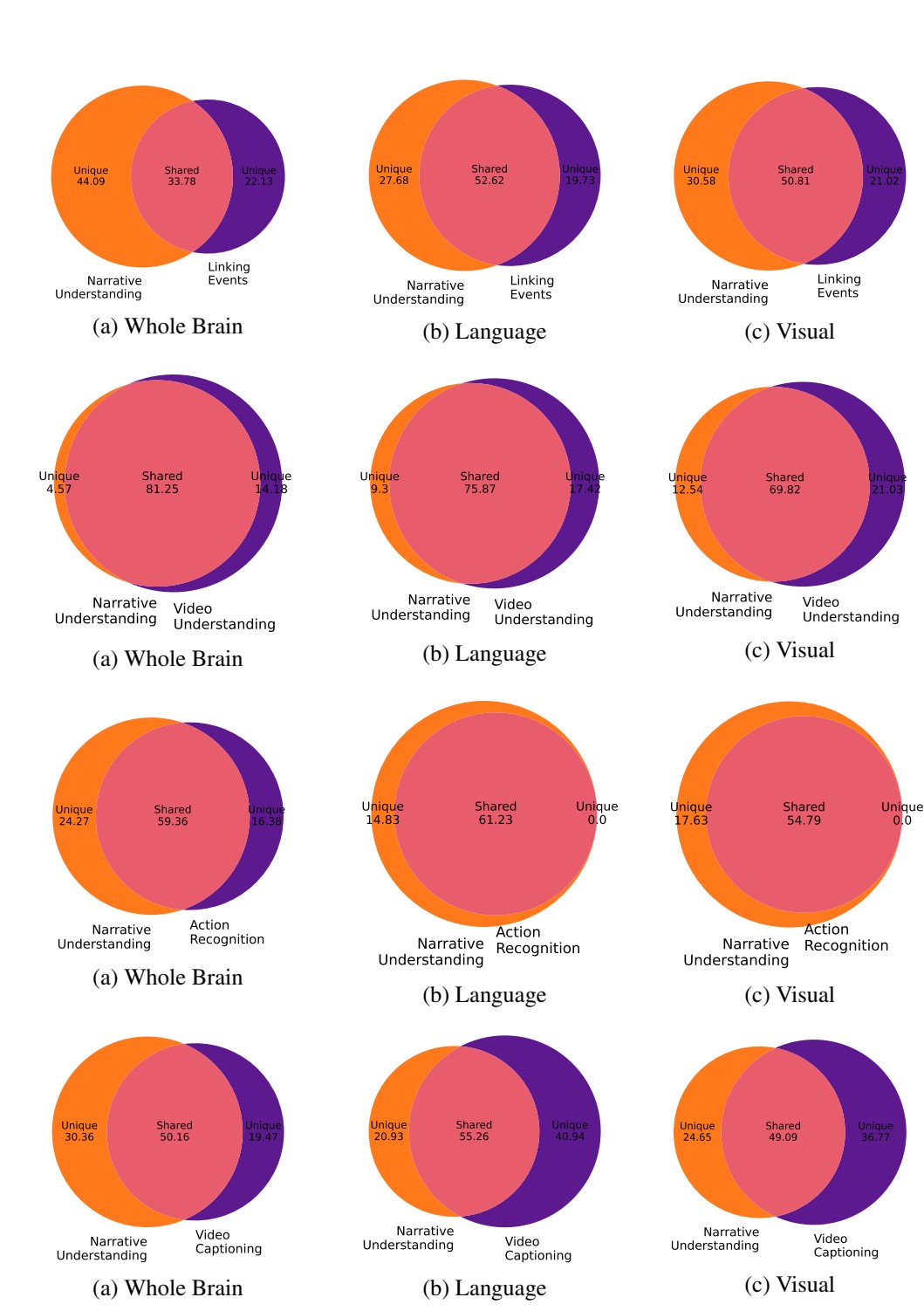

Figure 24: Shared and Unique Variance: Narrative Understanding vs. Linking Events Dark orange (left) shows variance unique to Narrative Understanding, indigo (right) shows variance unique to Linking Events, and the overlap indicates shared variance between both tasks.

| Task1 | Task2 | Whole Brain | | | Visual | | | Language | | |
|---|---|---|---|---|---|---|---|---|---|---|
| | | Shared | Uniq1 | Uniq2 | Shared | Uniq1 | Uniq2 | Shared | Uniq1 | Uniq2 |
| 1 | 2 | 90.69 | 5.26 | 4.05 | 72.05 | 13.91 | 14.04 | 77.46 | 12.07 | 10.47 |
| 1 | 3 | 83.53 | 10.05 | 6.42 | 73.67 | 10.28 | 16.05 | 77.05 | 10.72 | 12.23 |
| 1 | 4 | 84.51 | 9.65 | 5.84 | 71.87 | 13.82 | 14.31 | 75.97 | 12.27 | 11.76 |
| 1 | 5 | 79.16 | 13.51 | 7.33 | 66.82 | 14.35 | 18.83 | 73.47 | 13.07 | 13.46 |
| 1 | 6 | 81.48 | 13.34 | 5.18 | 68.44 | 17.28 | 14.28 | 73.59 | 15.37 | 11.04 |
| 1 | 7 | 83.07 | 10.44 | 6.49 | 71.99 | 11.88 | 16.13 | 75.20 | 12.30 | 12.50 |
| 1 | 8 | 81.25 | 14.18 | 4.57 | 69.82 | 17.63 | 12.54 | 75.87 | 14.83 | 9.30 |
| 1 | 9 | 86.94 | 7.57 | 5.50 | 73.42 | 10.25 | 16.34 | 78.27 | 9.05 | 12.68 |
| 1 | 10 | 84.55 | 9.06 | 6.39 | 73.46 | 10.59 | 15.95 | 76.42 | 10.32 | 13.26 |
| 1 | 11 | 85.44 | 8.51 | 6.05 | 74.92 | 11.12 | 13.96 | 76.56 | 10.96 | 12.48 |
| 1 | 12 | 82.46 | 11.66 | 5.88 | 72.88 | 12.75 | 14.37 | 76.02 | 12.50 | 11.48 |
| 1 | 13 | 91.81 | 4.20 | 3.99 | 74.92 | 11.82 | 13.26 | 80.06 | 10.00 | 9.94 |
| 2 | 3 | 83.59 | 9.72 | 6.69 | 73.14 | 11.39 | 15.47 | 74.15 | 12.80 | 13.05 |
| 2 | 4 | 86.25 | 7.40 | 6.36 | 73.32 | 13.52 | 13.16 | 74.41 | 12.14 | 13.45 |
| 2 | 5 | 77.09 | 14.33 | 8.58 | 64.55 | 17.14 | 18.31 | 70.20 | 15.08 | 14.72 |
| 2 | 6 | 79.86 | 13.99 | 6.15 | 69.43 | 17.86 | 12.71 | 73.10 | 14.96 | 11.94 |
| 2 | 7 | 83.62 | 9.46 | 6.92 | 72.53 | 12.65 | 14.82 | 71.61 | 14.43 | 13.95 |
| 2 | 8 | 81.30 | 13.10 | 5.60 | 67.98 | 18.96 | 13.05 | 72.05 | 16.07 | 11.88 |
| 2 | 9 | 86.64 | 7.42 | 5.93 | 73.55 | 12.35 | 14.11 | 75.55 | 10.62 | 13.83 |
| 2 | 10 | 85.25 | 7.97 | 6.78 | 72.98 | 12.28 | 14.73 | 73.28 | 12.51 | 14.21 |
| 2 | 11 | 84.70 | 8.31 | 7.00 | 73.27 | 12.25 | 14.48 | 72.48 | 13.27 | 14.25 |
| 2 | 12 | 82.97 | 11.16 | 5.88 | 73.06 | 14.41 | 12.54 | 72.99 | 14.99 | 12.02 |
| 2 | 13 | 91.78 | 3.66 | 4.55 | 74.89 | 12.59 | 12.52 | 78.19 | 9.77 | 12.03 |
| 3 | 4 | 68.68 | 13.67 | 17.64 | 68.53 | 18.38 | 13.09 | 71.98 | 14.19 | 13.83 |
| 3 | 5 | 50.07 | 24.61 | 25.32 | 52.60 | 24.08 | 23.32 | 60.68 | 17.79 | 21.53 |
| 3 | 6 | 61.39 | 21.67 | 16.94 | 61.59 | 22.97 | 15.44 | 65.21 | 18.68 | 16.12 |
| 3 | 7 | 65.21 | 17.99 | 16.80 | 64.73 | 20.33 | 14.94 | 66.85 | 17.80 | 15.35 |
| 3 | 8 | 66.30 | 20.20 | 13.49 | 61.04 | 23.96 | 15.00 | 62.43 | 21.86 | 15.71 |
| 3 | 9 | 70.23 | 13.71 | 16.06 | 70.07 | 16.68 | 13.25 | 72.20 | 12.52 | 15.28 |
| 3 | 10 | 66.99 | 13.00 | 20.01 | 68.60 | 15.97 | 15.42 | 64.43 | 15.79 | 19.78 |
| 3 | 11 | 68.07 | 14.39 | 17.54 | 66.84 | 17.50 | 15.66 | 66.97 | 16.85 | 16.18 |
| 3 | 12 | 61.81 | 19.24 | 18.95 | 65.81 | 19.69 | 14.50 | 67.09 | 17.92 | 14.99 |
| 3 | 13 | 83.92 | 6.44 | 9.64 | 71.83 | 16.87 | 11.31 | 76.76 | 12.86 | 10.38 |
| 4 | 5 | 55.03 | 24.36 | 20.61 | 53.05 | 20.94 | 26.00 | 59.06 | 18.82 | 22.13 |
| 4 | 6 | 61.72 | 25.66 | 12.62 | 59.66 | 24.72 | 15.62 | 63.75 | 21.99 | 14.26 |
| 4 | 7 | 69.00 | 17.62 | 13.38 | 66.08 | 17.45 | 16.47 | 67.89 | 17.50 | 14.61 |
| 4 | 8 | 63.88 | 21.85 | 14.27 | 60.24 | 23.59 | 16.17 | 65.25 | 19.95 | 14.80 |
| 4 | 9 | 71.16 | 16.55 | 12.28 | 65.51 | 18.15 | 16.34 | 68.66 | 16.14 | 15.19 |
| 4 | 10 | 66.37 | 18.11 | 15.53 | 63.85 | 17.11 | 19.04 | 57.73 | 20.94 | 21.33 |
| 4 | 11 | 72.37 | 13.56 | 14.07 | 70.00 | 13.01 | 16.99 | 70.64 | 13.35 | 16.02 |
| 4 | 12 | 66.38 | 18.76 | 14.86 | 64.80 | 17.67 | 17.53 | 67.94 | 17.21 | 14.85 |
| 4 | 13 | 86.69 | 6.09 | 7.23 | 71.23 | 16.28 | 12.49 | 76.56 | 13.87 | 9.57 |
| 5 | 6 | 50.13 | 27.24 | 22.63 | 51.63 | 27.81 | 20.56 | 58.56 | 23.05 | 18.39 |
| 5 | 7 | 49.08 | 24.63 | 26.29 | 53.55 | 25.15 | 21.30 | 55.77 | 24.66 | 19.57 |
| 5 | 8 | 47.03 | 27.55 | 25.43 | 53.22 | 28.86 | 17.93 | 53.88 | 26.92 | 19.21 |
| 5 | 9 | 55.06 | 21.61 | 23.34 | 56.84 | 24.75 | 18.42 | 62.62 | 19.24 | 18.15 |
| 5 | 10 | 47.76 | 23.54 | 28.70 | 55.84 | 22.99 | 21.17 | 54.52 | 22.48 | 23.00 |
| 5 | 11 | 52.17 | 22.58 | 25.25 | 57.44 | 22.32 | 20.24 | 57.94 | 22.48 | 19.58 |
| 5 | 12 | 47.50 | 26.51 | 25.99 | 56.38 | 25.48 | 18.15 | 58.21 | 23.50 | 18.29 |
| 5 | 13 | 79.36 | 6.98 | 13.67 | 66.31 | 16.96 | 16.74 | 71.80 | 12.91 | 15.29 |
| 6 | 7 | 60.01 | 17.04 | 22.96 | 59.05 | 17.09 | 23.86 | 61.14 | 18.01 | 20.84 |
| 6 | 8 | 54.31 | 21.48 | 24.22 | 57.44 | 21.55 | 21.01 | 62.62 | 18.13 | 19.25 |
| 6 | 9 | 64.33 | 13.06 | 22.61 | 60.10 | 16.20 | 23.69 | 64.68 | 13.72 | 21.60 |
| 6 | 10 | 57.84 | 16.91 | 25.25 | 61.41 | 14.59 | 24.00 | 61.01 | 16.15 | 22.84 |
| 6 | 11 | 62.94 | 14.26 | 22.81 | 62.17 | 15.15 | 22.68 | 63.32 | 15.40 | 21.28 |
| 6 | 12 | 55.82 | 19.64 | 24.54 | 60.18 | 17.37 | 22.45 | 60.36 | 18.93 | 20.71 |
| 6 | 13 | 81.42 | 5.21 | 13.37 | 67.46 | 13.51 | 19.02 | 71.93 | 11.31 | 16.75 |
| 7 | 8 | 58.19 | 23.15 | 18.65 | 60.58 | 23.47 | 15.95 | 61.00 | 20.86 | 18.13 |
| 7 | 9 | 70.87 | 14.02 | 15.11 | 70.43 | 15.05 | 14.51 | 71.25 | 12.70 | 16.05 |
| 7 | 10 | 68.57 | 12.51 | 18.92 | 67.67 | 13.27 | 19.06 | 63.76 | 14.39 | 21.84 |
| 7 | 11 | 60.77 | 18.94 | 20.29 | 58.79 | 21.23 | 19.98 | 55.14 | 21.77 | 23.09 |
| 7 | 12 | 66.57 | 17.86 | 15.57 | 67.97 | 17.05 | 14.98 | 67.18 | 17.38 | 15.44 |
| 7 | 13 | 85.27 | 6.01 | 8.72 | 72.66 | 15.56 | 11.78 | 74.88 | 13.08 | 12.03 |
| 8 | 9 | 62.84 | 15.99 | 21.18 | 63.11 | 15.66 | 21.22 | 68.03 | 13.67 | 18.31 |
| 8 | 10 | 60.10 | 17.38 | 22.52 | 59.39 | 16.80 | 23.81 | 60.46 | 16.80 | 22.74 |
| 8 | 11 | 60.31 | 14.63 | 25.07 | 61.67 | 13.24 | 25.09 | 61.38 | 15.64 | 22.98 |
| 8 | 12 | 60.04 | 18.69 | 21.28 | 62.31 | 17.41 | 20.28 | 65.74 | 16.70 | 17.56 |
| 8 | 13 | 81.06 | 5.66 | 13.27 | 68.01 | 14.38 | 17.61 | 74.50 | 11.65 | 13.85 |
| 9 | 10 | 69.21 | 14.34 | 16.44 | 68.83 | 12.98 | 18.19 | 67.69 | 15.88 | 16.44 |
| 9 | 11 | 70.80 | 13.15 | 16.05 | 69.96 | 14.08 | 15.96 | 70.82 | 14.04 | 15.15 |
| 9 | 12 | 69.68 | 16.60 | 13.72 | 70.09 | 14.45 | 15.46 | 70.62 | 16.10 | 13.29 |
| 9 | 13 | 87.40 | 5.23 | 7.37 | 72.02 | 15.46 | 12.53 | 77.48 | 12.70 | 9.82 |
| 10 | 11 | 68.63 | 16.35 | 15.02 | 67.96 | 16.43 | 15.61 | 64.85 | 19.12 | 16.04 |
| 10 | 12 | 65.06 | 20.66 | 14.27 | 63.79 | 21.85 | 14.36 | 61.84 | 23.65 | 14.50 |
| 10 | 13 | 85.63 | 6.39 | 7.99 | 72.34 | 16.92 | 10.73 | 75.85 | 14.09 | 10.06 |
| 11 | 12 | 61.95 | 22.51 | 15.54 | 65.60 | 19.55 | 14.85 | 63.80 | 21.51 | 14.69 |
| 11 | 13 | 86.42 | 6.00 | 7.58 | 74.60 | 14.29 | 11.11 | 76.83 | 12.89 | 10.28 |
| 12 | 13 | 83.82 | 5.77 | 10.41 | 71.56 | 15.38 | 13.06 | 75.37 | 12.20 | 12.43 |

Table 13: Variance partitioning for all the 13 video tasks averaged across all subjects for whole brain, visual and language regions with Qwen-2.5-VL model. Tasks are as follows: (1) Action Recognition (2) Video Understanding (3) Visual Question Answering (4) Video Captioning (5) Object and Scene Recognition (6) Commonsense Reasoning (7) Spatial Understanding (8) Temporal Ordering (9) Video reasoning (10) Narrative Understanding (11) Emotion and Sentiment Analysis (12) Global Appearance (13) Linking Events.

## Q   WHAT INSTRUCTION-TUNING ADDS BEYOND ZERO-SHOT ICL?

ICL models can follow zero-shot prompting i.e. in-context prompt, whereas Instruction-tuning adds a supervised signal that binds instruction tokens to stable computation paths. Below we decompose the architectural/representational differences.

To understand the difference in the working of IT and ICL models, we perform additional analysis. We compared instruction-tuned (IT) and in-context learning (ICL) models to identify fundamental differences in representational organization. For the 13 tasks, we first compute a 13×13 semantic-similarity matrix using MiniLM embeddings. We compute correlations between the upper triangles of the 13×13 semantic-similarity matrix and the corresponding representation-similarity matrix (same videos, same pipeline), per layer.

We find that for the instruction-tuned model (Qwen-2.5-VL-7B-Instruct), the correlation between instruction semantic similarity and internal representation similarity is weak across layers. (e.g., best layer L28: Pearson r=0.183, p=0.109; Spearman $\rho$=0.266, p=0.018; no layer remains significant after FDR across 29 layers. This supports a function-driven geometry rather than surface wording.

For Qwen-2.5-Omni-7B (ICL), the correlation between MiniLM-based instruction semantic similarity and model representation similarity is high (e.g., r≈0.78), indicating that prompt wording strongly drives internal states, consistent with more shallow, text-proximal matching.

Table 14: Instruction Tuning vs. In-Context Learning: Organizational Principles. *** p<0.001, FDR-significant across 29 layers; ns = not significant after FDR.

| Metric | Model | Layer 5 | Layer 14 | Layer 25 | Winner | Magnitude |
|---|---|---|---|---|---|---|
| Semantic correlation | IT (Qwen-2.5-VL) | Pearson $r = 0.077$ (ns) | 0.130 (ns) | 0.143 (ns) | — | Weak |
| | ICL (Qwen-2.5-Omni) | **0.690***** | **0.680***** | **0.777***** | ICL | ∼4.2× stronger |
| Silhouette score (cluster cohesion) | IT $\Delta$ (Func − Sem) | −0.102 | −0.171 | −0.172 | Semantic | Weak |
| | ICL $\Delta$ (Func − Sem) | −0.157 | −0.252 | −0.234 | Semantic | Moderate–Strong |
| Adjusted Rand Index (label alignment) | IT $\Delta$ (Func − Sem) | **+0.129** | **+0.142** | **+0.174** | Functional | Weak |
| | ICL $\Delta$ (Func − Sem) | −0.442 | −0.291 | −0.349 | Semantic | Very Strong |

- Silhouette score measures cluster cohesion and separation by computing the ratio of within-cluster to between-cluster distances for each sample, with values ranging from -1 (misclassified) to +1 (well-clustered) (Rousseeuw, 1987).

- Adjusted Rand Index (ARI) measures agreement between predicted clusters and ground-truth labels by counting pairwise agreements, adjusted for chance, with values ranging from -1 (worse than random) to +1 (perfect agreement). Positive ARI → clustering aligns with functional organization, Negative ARI → clustering aligns with semantic organization (Hubert & Arabie, 1985).

**Key findings: ICL vs IT.** We compared instruction-tuned (IT) and in-context learning (ICL) models to identify fundamental differences in representational organization. We find that ICL models show higher semantic organization with semantic correlation: r=0.78 vs 0.14 (IT), a 4.2x advantage (p<0.001). IT models show emerging functional organization but with weak effects: Functional ARI advantage: $\Delta$=+0.13 to +0.17 across layers, progressive strengthening: L5: +0.129 → L25: +0.174 (increasing with depth).

ICL behaves wording-sensitive (strong semantic-representation coupling), whereas instruction-tuned models behave function-driven (low coupling to wording), indicating that instruction tuning binds instructions to stable computation paths and task-specific subspaces that improve brain alignment.

## R   INCLUSION OF OTHER VIDEO MLLMS

We have added Kimi-VL (Team et al., 2025) to our evaluation and ran it through the same brain-encoding pipeline (identical preprocessing, instruction prompts, voxel-wise mapping, and normalization). The results are now reported in Fig. 25. We observe that the instruction-tuned Kimi-VL model demonstrates similar encoding performance to the other six instruction-tuned video MLLMs.

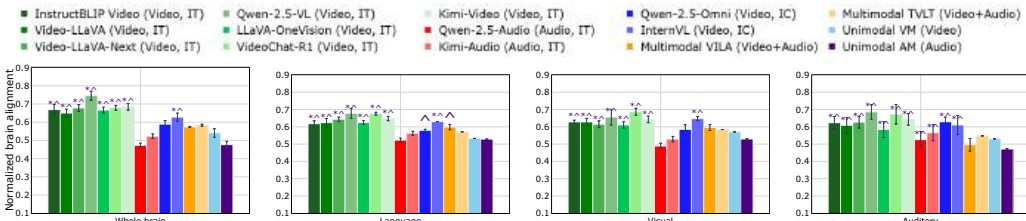

Figure 25: Average normalized brain alignment of instruction-tuned video MLLMs vs instruction-tuned audio MLLMs vs in-context learning video MLLMs vs multimodal and unimodal models across whole brain, language, visual and auditory regions. Error bars indicate the standard error of the mean across participants. ∗ implies that instruction-tuned MLLM embeddings are significantly better than multimodal models and ∧ means that instruction-tuned MLLM embeddings are significantly better unimodal models with p≤ 0.05.

# S  SELF-CONTROLLED EXPERIMENTS: COMPARING MODELS BEFORE AND AFTER INSTRUCTION TUNING

Comparing the same backbone before and after instruction tuning is the most direct way to test causality. Since the exact pre–instruction-tuning checkpoint of Qwen-2.5-VL-7B-Instruct is not publicly available, we approximate this setting by comparing two models from the same Qwen-2.5 family and the same scale (7B): Qwen-2.5-VL-7B-Instruct and Qwen-2.5-Omni-7B run in video-only mode, so that both models receive the same visual input. This keeps architecture family, parameter size, and modality comparable, and highlights the effect of instruction tuning. To clarify, we have already reported the corresponding comparison in the original draft. For clarity, we provided the same result in the paper that compares the same model category with instruction-tuned and pretrained versions.

We use the pretrained InternVL-8B model already reported in the paper and compare it to its instruction-tuned variant InternVL-8B-Instruct. For both models, we issue the same instruction ("Describe the video") and extract the resulting instruction-specific representations across layers. This analysis shows how instruction tuning on the same model category changes layer-wise representations and improves brain alignment.

**Qwen2.5-VL: before vs. after instruction tuning:** Using brain predictions across layers for Qwen-2.5-VL-7B-Instruct and Qwen-2.5-Omni-7B models, we compute per-layer $\Delta$ alignment as

$\Delta$ alignment per layer = Normalized brain alignment (Qwen-2.5-VL-7B-Instruct) - Normalized Brain alignment (Qwen-2.5-Omni-7B)

We make the following observations: (i) All layers improve (all $\Delta > 0$), (ii) Depth trend: $\Delta$ decreases with depth (early > mid > late): early 0.177, mid 0.156, late 0.139; Spearman $\rho$ = -0.77, p ≪ 0.001. (ii) The center-of-mass of the improvement is near Layer 14/29 (normalized 0.47). Thus, while the absolute alignment peak under instruction occurs in mid to late layers (Fig. 26), the incremental benefit over the pretrained baseline is strongest early and remains positive throughout depth.

**InternVL: before vs. after instruction tuning:** Using brain predictions across layers for InternVL-8B-Instruct and InternVL-8B (pretrained) models, we compute per-layer $\Delta$ alignment, as shown in Fig. 27. We make the following observations: (i) Similar to Qwen series, all layers positive: min $\Delta$ = 0.0557 (L5), max $\Delta$ = 0.1396 (L1), (ii) Depth trend: gains increase into mid/late (early 0.0934 $\pm$ 0.0250, mid 0.0961 $\pm$ 0.0128, late 0.1100 $\pm$ 0.0104); Spearman $\rho \approx$ +0.34, p $\approx$ 0.07. (iii) The center-of-mass of the improvement is near Layer 15/29 (normalized 0.535). This implies that Instruction tuning yields mid/late-layer improvements for InternVL.

Table. 15 shows quantitative analysis of $\Delta$ alignment across families, we make following observations: (i) instruction tuning preserves the same alignment hierarchy (mid→late layers peak) and shifts preferred processing toward later layers (positive shift in preferred layer), (ii) the locus of the gain is family-specific: Qwen shows larger early-layer increases, whereas InternVL shows mid/late-layer increases—consistent with representational reconfiguration rather than a uniform shift.

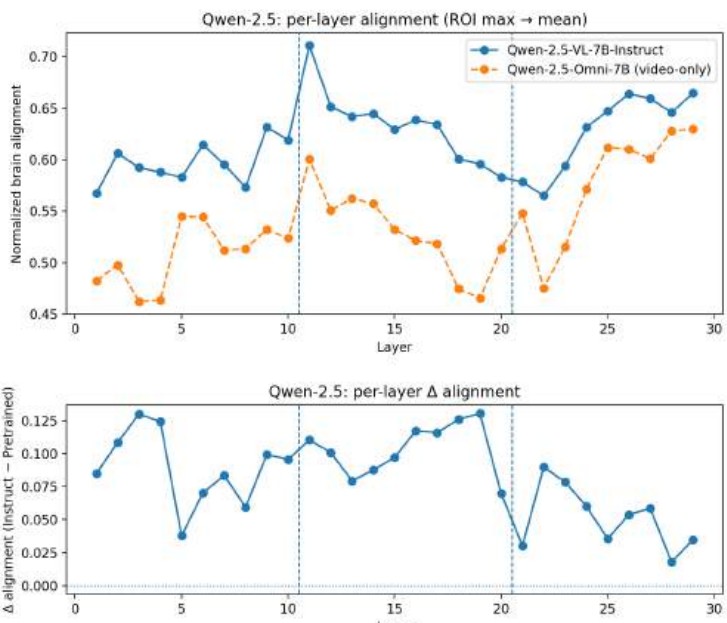

Figure 26: Qwen2.5-VL: (top) Normalized brain alignment was computed before vs. after instruction tuning: Using brain predictions across layers for Qwen-2.5-VL-7B-Instruct and Qwen-2.5-Omni-7B models. (bottom) $\Delta$ alignment per layer:= Normalized brain alignment (Qwen-2.5-VL-7B-Instruct) - Normalized Brain alignment (Qwen-2.5-Omni-7B) .

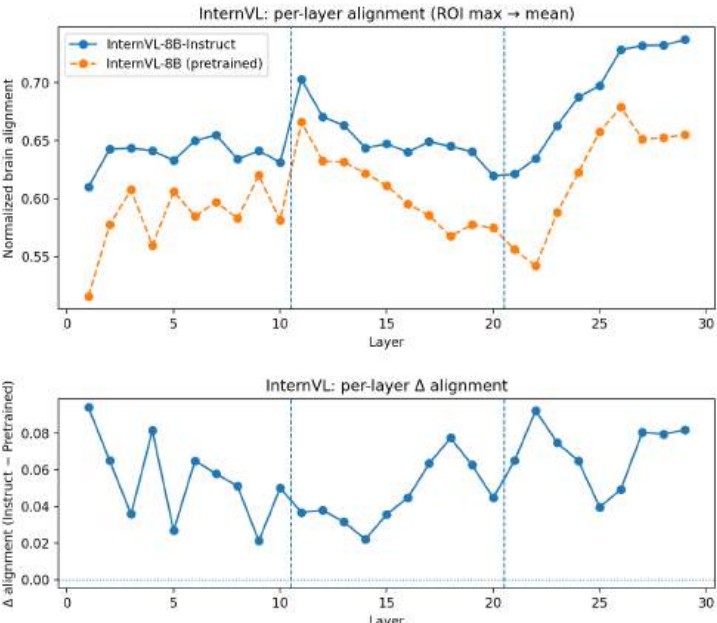

Figure 27: InternVL: (top) Normalized brain alignment was computed before vs. after instruction tuning: Using brain predictions across layers for InternVL-7B-Instruct and InternVL-7B models. (bottom) $\Delta$ alignment per layer:= Normalized brain alignment (InternVL-7B-Instruct) - Normalized Brain alignment (InternVL-7B).

Table 15: Self-controlled $\Delta$ alignment (Instruction $-$ Pretrained). Early = Layers 1–10, Mid = 11–20, Late = 21–29 (mean$\pm$SD). COM is center-of-mass of $\Delta$ across layers (normalized by depth). $\rho$ is Spearman correlation between layer index and $\Delta$.

| Model | Early $\Delta$ | Mid $\Delta$ | Late $\Delta$ | COM (norm) | $\rho$ (layer,$\Delta$) | p-value | Min $\Delta$ (layer) | Max $\Delta$ (layer) | All $\Delta>0$ |
|---|---|---|---|---|---|---|---|---|---|
| Qwen-2.5-VL-7B | $0.177 \pm 0.020$ | $0.156 \pm 0.015$ | $0.139 \pm 0.023$ | 0.474 | $-0.767$ | $< 1e{-}6$ | 0.088 (L29) | 0.207 (L1) | ✓ |
| InternVL-8B | $0.093 \pm 0.025$ | $0.096 \pm 0.013$ | $0.110 \pm 0.010$ | 0.535 | $+0.337$ | 0.074 | 0.056 (L5) | 0.140 (L1) | ✓ |

## T CORRELATION BETWEEN INSTRUCTION SEMANTICS AND MODEL REPRESENTATIONS

We conducted a comprehensive semantic similarity robustness study.

**Semantic Similarity Measurement.** To validate that our task-specific instruction set contains semantically similar pairs, we perform following:

- Using 13 video task-specific instructions, we first computed pairwise semantic similarity using two independent text embedding models (all-MiniLM-L6-v2 (Reimers & Gurevych, 2019) and MPNet (Reimers & Gurevych, 2019)). Both models produced highly consistent semantic similarity matrices (Pearson r = 0.94 between MiniLM and MPNet embeddings).
- Captures fine-grained semantic relationships between instruction texts.
- Example pairs: "Describe the video" vs. "Caption the video" (high semantic similarity).

Fig. 28 shows pairwise semantic similarity between 13 instruction prompts computed using MiniLM embeddings (left) and MPNet (right). From Fig. 28 (left), we observe that the semantic similarity ranges from 0.15 to 0.85 (mean: 0.42 ± 0.18), with multiple high-similarity pairs identified (>0.60):(Action Recognition vs. Video Understanding (0.68), Video Understanding vs. Visual Question Answering (0.65), Object & Scene Recognition vs. Action Recognition (0.68)). We also observe low semantic similarity pairs: (Commonsense Reasoning vs. Most others, Commonsense Reasoning vs. other tasks, Spatial Understanding vs. Emotional/Narrative). This confirms our instruction set contains the "semantically similar or equivalent instructions" for testing fine-grained task distinctions.

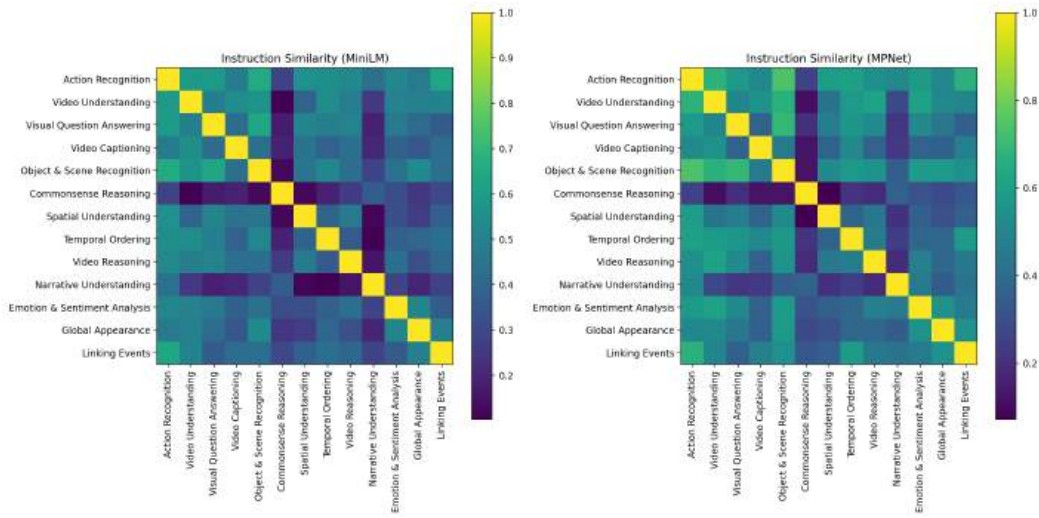

Figure 28: Instruction semantic similarity using text-embeddings.

**Instruction-tuned Model Internal Representations.** To measure how the model internally processes different task-specific instructions, we extracted language hidden states from the Qwen2.5-VL-7B-Instruct model across all processing layers. We perform the following:

- Extracted language hidden states across all 29 layers for each task-specific instruction.
- Used the same video input with varying instructions to isolate instruction effects.
- Analyzed three key layers: Early (Layer 5), Middle (Layer 14), and Late (Layer 25).
- While we present detailed results for three representative layers, we computed correlations and clustering metrics across all 29 layers to ensure findings are not artifacts of specific layer selection.

Fig. 29 shows pairwise similarity between 13 instruction prompts computed using three key layers. From Fig., we observe that layer-specific differentiation patterns emerge: (i) In early layers,

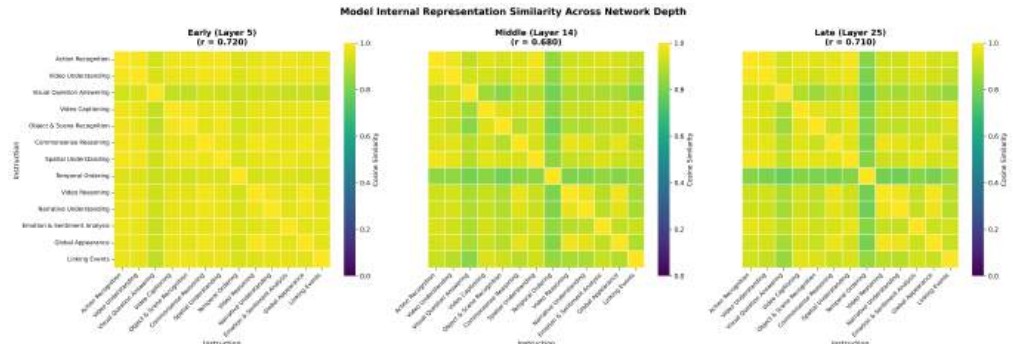

Figure 29: Qwen-2.5-VL-7B-Instruct: Model internal representational similarity across layer depth.

Table 16: Correlation across network depth. Coupling between instruction semantic similarity and instruction-conditioned representation similarity.

| Layer range | Pearson $r$ (mean $\pm$ sd) | p-value range | Spearman $\rho$ (mean $\pm$ sd) | Significance |
|---|---|---|---|---|
| Early (L1–L10) | $0.066 \pm 0.015$ | 0.497–0.717 | $0.184 \pm 0.022$ | No individual layer sig. after FDR |
| Middle (L11–L20) | $0.126 \pm 0.009$ | 0.219–0.330 | $0.199 \pm 0.011$ | Some uncorrected $p < 0.05$; none FDR-sig |
| Late (L21–L28) | $0.149 \pm 0.018$ | 0.109–0.240 | $0.247 \pm 0.013$ | Several uncorrected $p < 0.05$; none FDR-sig |
| Overall (L1–L29) | $0.113 \pm 0.041$ | 0.109–0.717 | $0.207 \pm 0.034$ | No FDR-significant effects |

relatively uniform representation similarity suggests that initial encoding stage has limited task specialization, (ii) In middle layers, increased variation in similarity patterns indicates evidence of task-specific transformations, and (iii) In later layers, pattern similar to early layer but with subtle differences implies refinement of task-specific representations. Overall, the quantitative analysis from Table 1 reveal that middle layer shows greatest differentiation, while early and late layers show convergent patterns.

While we observe clear structure in model representations, a critical question remains: Do these patterns reflect semantic similarity or functional task requirements? To answer this, we next compare model representations with semantic similarity computed from text embeddings.

**Correlation Analysis: Semantic Similarity vs. Model Representations.** For each layer, we computed correlations between: Semantic similarity (from text embeddings) and Model representation similarity (from hidden states) as follows:

For each layer L in $[1, 2, \cdots, 29]$: (i) Extract upper triangle of semantic similarity matrix (78 pairs) (ii) Extract upper triangle of model similarity matrix (78 pairs) (iii) Compute Pearson correlation between the two sets (iv) Compute Spearman correlation for robustness (v) Test significance against random baseline

We compute both (a) the semantic similarity between instructions using text embedding models, and (b) the similarity of internal model representations (e.g., cosine similarity between layer vectors). A strong positive correlation between these measures would demonstrate that instruction tuning enhances the model's sensitivity to fine-grained semantic distinctions. Contrary to the hypothesis that surface semantic similarity should drive internal similarity, as shown in Table 16, we observe near-zero to modestly positive coupling (Pearson r $\approx$ 0.04–0.18; Spearman $\rho \approx$ 0.15–0.27), with several mid/late layers show uncorrected Spearman p<0.05, but no layer survives FDR correction across 29 layers (Benjamini–Hochberg). This pattern is highly consistent throughout the network depth, indicating that instruction tuning does not enhance semantic sensitivity but instead prioritizes functional task organization.

**Statistical validation (Mantel permutation).** We assess whether instruction semantics predict representation similarity using a Mantel permutation test (10,000 label shuffles) per layer, with FDR correction across layers. Pearson r and Spearman $\rho$ are the raw matrix correlations; Mantel z and q report significance. From Table 17, we find that Mantel tests show no significant association between semantic and representation spaces across layers, indicating representations are not organized by surface semantic similarity.

**Clustering Quality Analysis.** We perform clustering analysis to determine whether the model organizes instructions by semantic similarity or functional task categories.

Table 17: **Mantel permutation test (semantic vs. representation similarity).** 10,000 label permutations; FDR applied across layers.

| Layer | Pearson $r$ | Spearman $\rho$ | Mantel $z$ | $q$-value (FDR) |
|---|---|---|---|---|
| L5 (early) | 0.077 | 0.210 | 0.65 | 0.482 |
| L14 (middle) | 0.130 | 0.201 | 1.12 | 0.376 |
| L25 (late) | 0.143 | 0.237 | 1.23 | 0.338 |

Table 18: **Functional vs. Semantic Organization.** $\Delta$ = Functional − Semantic.

| Layer | Silhouette (Func) | Silhouette (Sem) | $\Delta$ Silh | ARI (Func) | ARI (Sem) | $\Delta$ ARI |
|---|---|---|---|---|---|---|
| L5 | −0.263 | −0.161 | −0.102 | −0.003 | −0.132 | +0.129 |
| L14 | −0.290 | −0.119 | −0.171 | +0.010 | −0.132 | +0.142 |
| L25 | −0.324 | −0.152 | −0.172 | +0.042 | −0.132 | +0.174 |

Note: Silhouette $\in$ [-1, 1]; larger values indicate better separation. Both labelings yield modest/negative silhouettes (overlapping clusters), but functional clustering is consistently higher (less negative) than semantic (positive $\Delta$). ARI $> 0$ for functional and $< 0$ for semantic further indicates that functional labels align better with representation geometry.

From Table 18, we make the following key findings:

- Functional categories consistently outperform semantic clusters across all metrics and layers.

- Silhouette difference: $\Delta$ = -0.17 to +0.10 (semantic labels are less-negative)

- ARI advantage: $\Delta$ = +0.129 to +0.174 (strong functional alignment)

- Middle layer specialization: mid/later layers shows strongest functional differentiation (ARI $\Delta$ = +0.423)

Progressive functional specialization: ARI advantage increases with depth, demonstrating task-specific organization strengthens in later layers. We focus on ARI rather than Silhouette because ARI measures alignment with ground-truth task categories (what instruction tuning teaches), while Silhouette measures cluster compactness (which may reflect pre-trained semantic structure). Overall, Instruction tuning successfully imparts functional task structure (ARI evidence) while preserving semantic coherence (Silhouette evidence), resulting in dual organizational principles.

**Visualization: t-SNE Projections (illustrative only)** We performed t-SNE dimensionality reduction to visualize how the model organizes instruction task-specific representations in each layer (see Fig. 30, attached). We emphasize that t-SNE is used for visualization only; conclusions rely on ARI/silhouette and Mantel statistics. From Fig. 30, we make the following observations: (1) Instructions cluster by functional task requirements rather than semantic similarity. (2) Clear spatial separation between: (i) Perceptual prompts(object recognition, appearance) $\rightarrow$ Right region, (ii) Reasoning prompts(commonsense, video reasoning) $\rightarrow$ Center region, (iii) Temporal-causal prompts(action recognition, event linking) $\rightarrow$ Bottom region, (iv) Descriptive prompts(video understanding, captioning, narrative) $\rightarrow$ Distributed pattern. (3) Semantically similar instructions (e.g., "Describe video" vs. "Caption video") are spatially separated. (4) Functional categories show consistent grouping patterns across layers.

# U  DISCUSSION ON CONTROLLING ARCHITECTURAL AND PRETRAINING DIFFERENCES ACROSS MLLMS

MLLMs differ in pretraining corpus and design. We therefore took three complementary steps to ensure fair and scientifically valid comparisons.

- (i) Controls in the evaluation protocol: Across all models, our encoding pipeline is the same as we use the same instruction templates, same video and audio datasets as input for feature extraction, use of the same cross-subject prediction scores for estimating normalized brain alignment per model.

- (ii) Matching scale: While all the MLLMs we tested are  7B to 8B models across both instruction-tuned video and audio MLLMs and in-context learning pretrained MLLMs. Thus, model training sizes close to several million across all instruction-tuned video MLLMs, with varying modality mixes and task diversity.

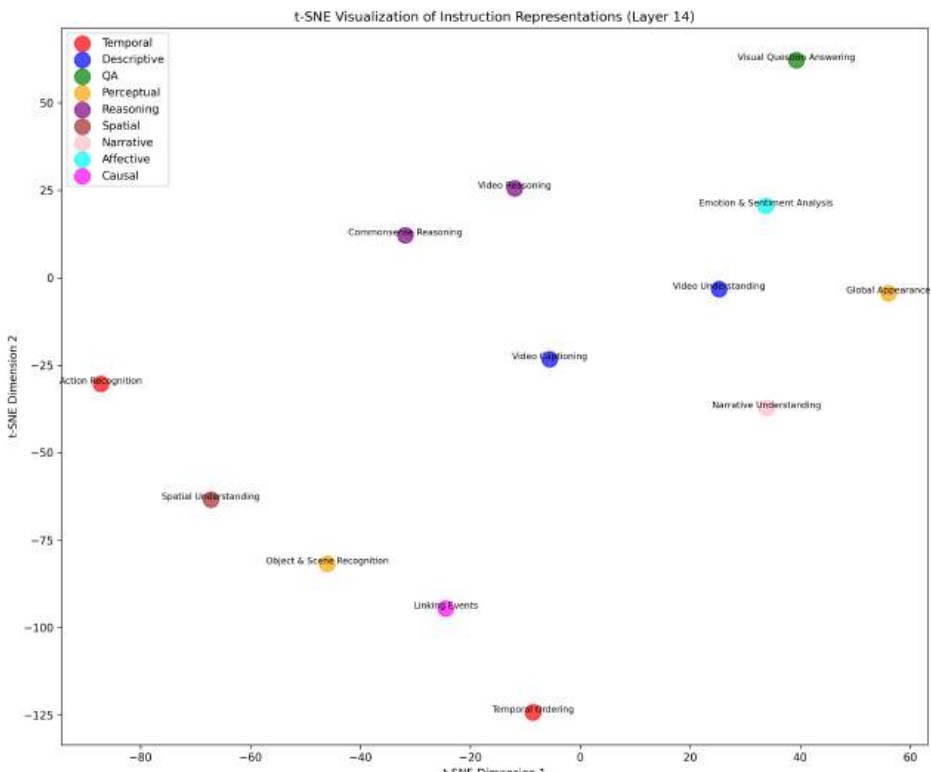

Figure 30: Clustering instruction task-specific representations.

- (iii) Acknowledging training-data heterogeneity and checking robustness: The instruction-tuned video MLLMs differ substantially in training sources: Some (like LLaVA-NeXT-Video and Video-LLaVA) include explicit video instruction datasets, while others (like InstructBLIPVideo and LLaVA-OneVision) primarily adapt image-based instructions to video via frame sampling. However, across all instruction-tuned videos MLLMs show similar normalized brain alignment to our results suggesting that depth alone does not account for the observed performance differences.

We also acknowledge that testing a larger number of models within a given class can help determine if the effect of modality is robust across various model configurations.

**Literature precedent:** Our approach follows established practice in neuro-AI, where models with differing training corpora/architectures are compared under a shared brain-encoding protocol and ceiling normalization to study representational alignment (Schrimpf et al., 2021; Toneva & Wehbe, 2019; Antonello et al., 2021; Aw & Toneva, 2023; Aw et al., 2023). Specifically, the extensive precedent in the literature, from studies comparing 43 models (Schrimpf et al., 2021) to those examining 101 models (Antonello et al., 2021) in language models. Similarly Oota et al. (2024a) and Antonello et al. (2024) compared several text and speech models during language comprehension. These studies demonstrate that this approach is both valid and valuable for understanding the relationship between artificial and biological language processing. It is important to observe that all the above studies utilize a number of models that are different in training architecture and training datasets, however the primary goal of all these studies is to investigate how close the semantic representations captured by each model aligns with brain-relevant semantics. We have added pretraining corpus details across MLLMs in Table 19.

Table 19: **Training data and scale of compared multimodal large language models (MLLMs).**

| Model | Training Data | Size |
|---|---|---|
| InstructBLIP-Video | 26 public datasets: MSCOCO, TextCaps, NoCaps, VQA v2, iVQA, MSRVTT-QA, MSVD-QA | Not specified |
| Video-LLaVA | LAION-CC-SBU (558K), Valley (702K), LLaVA-Instruct (665K image-text + 100K video-text) | ∼1.3M total |
| LLaVA-NeXT-Video | LLaVA-Video-178K synthetic + real video QA/caption data | ∼1.4M total |
| Qwen2.5-VL | Web-scale 4T tokens (text + multimodal), post-trained on 2M mixed samples | 4T tokens + 2M samples |
| VideoChat-R1 | Qwen2.5-VL backbone, 18K video instruction samples (temporal grounding, tracking, QA) | 18K samples |
| LLaVA-One-Vision | OneVision dataset (3.2M single-image + 1.6M multi-image/video), Evo-Instruct (143K) | ∼5M total |
| TVLT | HowTo100M, YTTemporal180M (video-audio pairs) | Large-scale (not stated) |
| VideoMAE | Kinetics-400, Something-Something v2, Epic Kitchens 100 | ∼550K videos |
| AST | AudioSet, ESC-50, Speech Commands | ∼2+M audio clips |

# V    VALIDATION OF "HIERARCHICAL CORRESPONDENCE" BETWEEN MODEL LAYERS AND BRAIN REGIONS

We compare hierarchical correspondence between model layers and brain regions in two settings: (i) Scaling the instruction-tuned model with correct video prompt: Qwen-2.5-vl-3B Instruct vs. Qwen-2.5-vl-7B Instruct, (ii) an instruction-tuned model with a non-natural language prompt, e.g.: ##### #### #### ##### this does not provide any meaning while passing video as input. Similar to Qwen-2.5-VL-7B Instruct model, we perform qualitative analysis by considering each voxel is color coded with the MLLM layer number (out of 29) that led to the highest normalized brain alignment. Fig 31 shows the resulting brain maps for the 3B-Instruct model (a) and for the 7B-Instruct model with the non-language prompt (b). We make the following observations: (i) Non-language control (random prompt): There is no hierarchy of information processing observed across layers, where the brain prediction is dominated by early layers across cortex (ii) Instruction condition (3B model): The hierarchical pattern still remains the same i.e., a systematic early to mid/late gradient across cortex, i.e., the same hierarchical pattern we observed with the larger model.

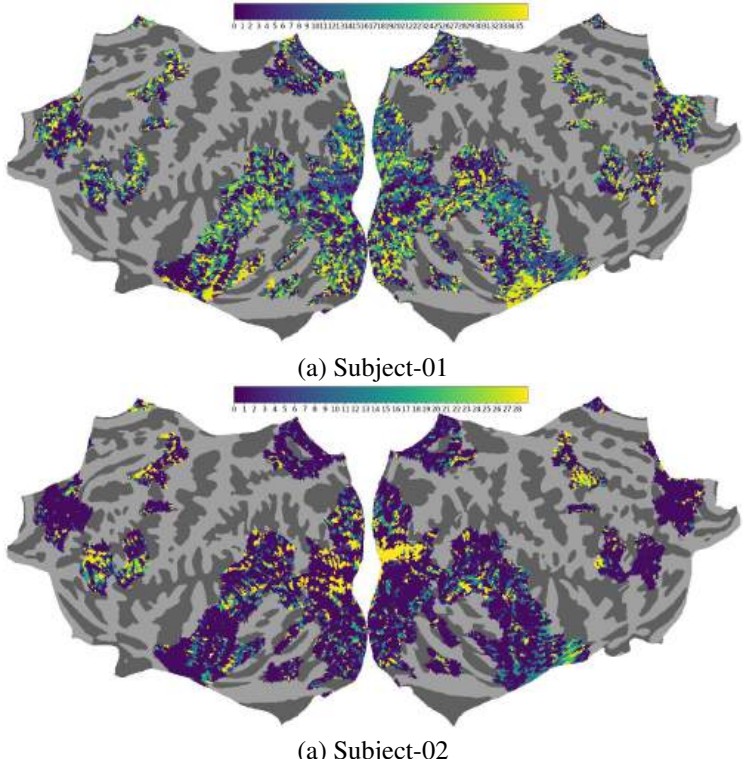

(a) Subject-01

(a) Subject-02

Figure 31: (a) Qwen-2.5-VL-3B Instruct and (b) Qwen-2.5-VL-7B Instruct non-natural language prompt (layer-wise alignment): Each voxel is color coded with the MLLM layer number (out of 29) that led to the highest normalized brain alignment. The color bar highlights color codes for each layer. The voxels are projected onto the flattened cortical surface of average across subjects on 'fsaverage' surface.

We also perform quantitative analysis to show a layerwise normalized brain alignment across three models, as shown in Fig. 32. Instruction prompts (Qwen-2.5-VL-7B; green solid) show a clear hi-

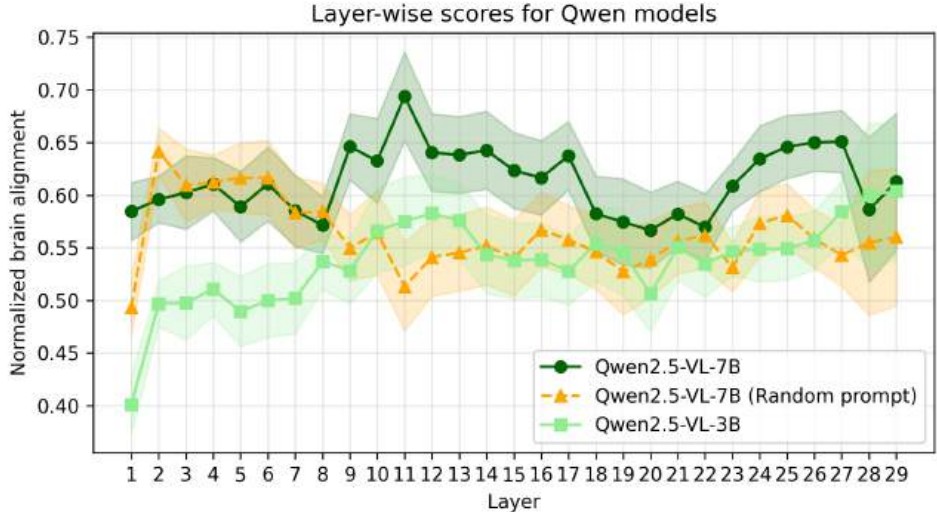

Figure 32: Layerwise normalized brain alignment for Qwen models.

erarchy: alignment rises from early layers and is strongest in mid to late layers. A non-language control (same model, orange dashed) flattens toward early layers, indicating that the hierarchy depends on instruction semantics rather than generic prompting. The smaller 3B model (light-green squares) exhibits the same shape with lower amplitude, demonstrating scale robustness of the hierarchy. Overall, we observe a significant shift in preferred layer under instruction vs. non-language, where a positive layer-trend is only observed for valid natural language instructions.

## W  NORMALIZED BRAIN ALIGNMENT: CROSS-SUBJECT VS. REPEAT-BASED EV CEILING

We also compute the explainable variance (EV) using repeated test movies and perform thresholding on EV voxels. The EV is computed now based on (Schoppe et al., 2016). Using EV with threshold of 0.05, the normalized brain alignment on this mask. For fair comparison, we used the same reliability threshold ($\geq 0.05$) as in the cross-subject ceiling analysis.

From the Table 20, we find that the normalized brain alignment computed with the repeat-based EV ceiling is very similar to that obtained with the cross-subject ceiling; the model ranking is unchanged. We also include cortical flatmaps of repeat-based explainable variance (EV) for each participant in the Fig. 33, showing the spatial distribution of reliable voxels (EV $\geq 0.05$).

Table 20: Normalized brain alignment: cross-subject vs. repeat-based EV ceiling

| Model | Cross-subject | Repeat EV | $\Delta$ (Repeat − Cross) | %$\Delta$ |
|---|---|---|---|---|
| InstructBLIP Video | 0.669 | 0.645 | −0.024 | −3.64% |
| Video-LLaVA | 0.650 | 0.652 | +0.002 | +0.35% |
| LLaVA-NeXT-Video | 0.678 | 0.642 | −0.037 | −5.39% |
| Qwen-2.5-VL-7B | 0.746 | 0.689 | −0.056 | −7.54% |
| LLaVA-OneVision | 0.666 | 0.639 | −0.028 | −4.13% |

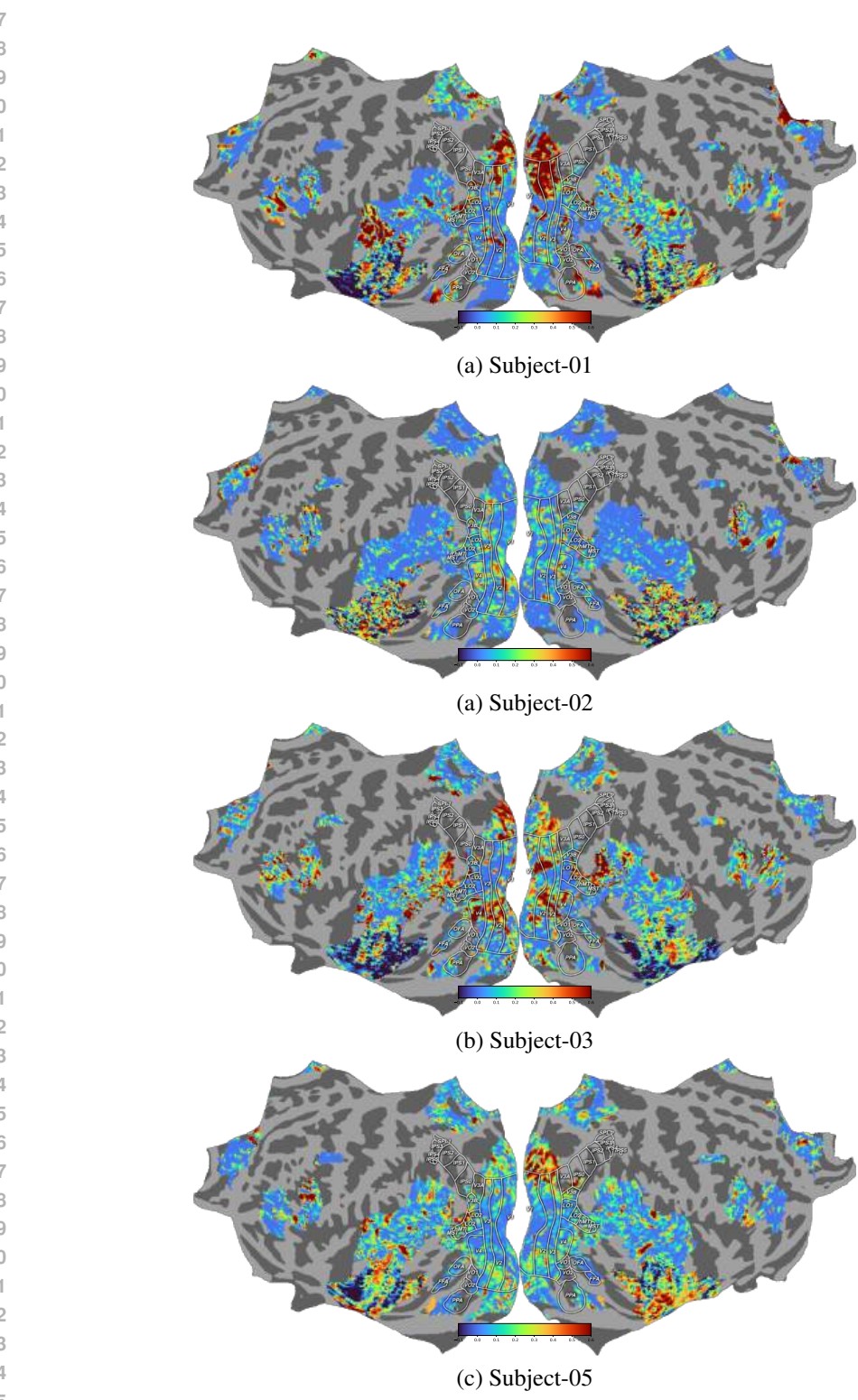

(a) Subject-01

(a) Subject-02

(b) Subject-03

(c) Subject-05

Figure 33: Estimated explainable variance for all four participants for the naturalistic movie watching. Explainable variance scores for each voxel in each subject are projected onto the subject's flattened cortical surface.

