# OpenReview forum: "Instruction-Tuned Video-Audio Models Elucidate Functional Specialization in Brain"
_ICLR.cc/2026/Conference — Submitted to ICLR 2026_

### Official Review · Reviewer_yTV4 · 2025-10-30

**Soundness:** 3
**Presentation:** 3
**Contribution:** 3
**Rating:** 6
**Confidence:** 3

**Summary:**

This study uses instruction tuned MLLMs to model multimodal human brain activity during audiovisual movie viewing. They find that instructive tuning improves brain predictivity of both video and audio MLLMs. Instruction tuning improves video model predictivity across the brain, but audio improvements are limited to auditory cortex. The results also show how different types of instruction tuning differentially improve prediction across cortex, revealing known functional preferences across the brain. Finally, the models also reproduce the layer-wise hierarchy, with early layers most predictive of sensory regions, and later layers predicting higher-level brain areas.

**Strengths:**

This an interesting and timely paper showing novel and potentially more interpretable ways to improve modeling of multimodal brain activity.

The modeling results reproduce known feature tuning and cotical hierarchies

This study seems to open the door to many future applications for both neuroscience and neuroengineering.

The paper is comprehensive and clearly written.

**Weaknesses:**

While the results are interesting and novel it is not entirely clear what is at stake - is the main goal of this work to better understand the brain or for future applications (if so what)? It is easy to imagine future benefits of this work, particularly with the differential effects for different types of instruction tuning, but it’s not entirely clear how to move beyond mapping already known brain function.

Overall the figures are quite clear but it is hard to keep track of the different models/instructions. I wonder if Fig 2 could be labeled not only with model name but also with model type (video, audio, instruction-tuned, not insturction tuned, etc). The color coding already seems to obey this.

Relatedly, it is hard to differentiate the colors in Fig 3 (this may be unavoidable given the large number of categories). A table or graphic with the different instructions, their task grouping, and (coloring in this way) may help.

As a smaller point - the intro lays out two types of multimodal brain model (for unimodal versus multimodal stimuli). While these are clearly different applications, the summary suggests they are largely similar (multimodal better than unimodal models). This section could be streamlined and I don’t think is entirely accurate, as the benefits of multimodal modeling are usually minimal for unimodal stimuli.

**Questions:**

How should we interpret the finding that video models are the most predictive of language and auditory regions?
How can we disentangle the role of more training / data versus the specific contribution of the instruction tuning?
How does cross subject predictivity compare to model predictivity shown in Fig 2?
Are the whole brain results in all cortical voxels? Given some movies are repeated it may make sense to restrict this to only reliable voxels

---

> ### Author Response · Authors · 2025-11-20
>
> *We thank the reviewer for their strong positive, insightful and valuable comments and suggestions which are crucial for further strengthening our manuscript.*
>
> **Q1. Main goal of this work and how to move beyond mapping already known brain function?**
>
> Thank you for this question. To clarify, our main goal and motivation as follows:
>
> * A central challenge in the brain–model alignment is that deep neural networks learn rich, complex representations that mix many task-specific and fine-grained factors.
> * To examine these representations, prior work has
>    * disentangled representations by controlling syntax/semantics, modality contributions, or by perturbing linguistic properties [Toneva et al. 2022, Oota et al. 2023, Reddy et al. 2021], and
>    * performed task-specific alignment, training separate models (or heads) per task to predict brain activity [Schrimpf et al. 2021, Oota et al. 2022, Aw et al. 2023, Oota et al. 2023].
> * However, the limitation is that the task-specific alignment typically requires different models and datasets for each task, complicating fair comparisons and causal interpretation.
>
> **Our approach:**
> * To overcome the prior limitations, recent advances in in-context learning and instruction-tuned multimodal LLMs make it possible to elicit task-specific representations from a single backbone using language-guided instructions.
> * This enables controlled, within-model comparisons across tasks and modalities, improving interpretability and scientific validity in brain alignment.
>
> * Our aim is two fold:
>   * **Neuroscience:** probe how task-specific instructions reorganize multimodal information processing across cortical hierarchies for language, vision, and audition (instruction-dependent hierarchies, layer specificity, ROI dissociations).
>   * **AI Models:** characterize where (which layers) and how instruction tuning creates task-selective subspaces, beyond surface semantics.
>
> * Why is this more than “just mapping.”
>   * We show that representation geometry is organized by functional task demands, not wording (semantic–representation coupling near zero/positive; functional clusters > semantic) (Kindly refer **CQ4** in common responses).
>   * We provide self-controlled pre/post evidence that instruction tuning shifts preferred processing later in the hierarchy i.e., a mechanistic link between instructions and representational change that improves brain alignment.
>
> **Implications to Neuro-AI and concrete future directions:**
>   * Video vs. Audio instruction-tuned MLLMs: Instruction-tuned video models already show high predictivity; current audio models align well in early auditory cortex but lag in higher-order semantic regions, pointing to a need for stronger semantic grounding in audio IT-MLLMs.
>   * A promising future direction is to inject brain-informed bias into audio instruction-tuned models and test whether this improves alignment in semantic cortical regions.
>   * Model-driven experiments: Use instruction swaps, non-language controls, and targeted ablations (e.g., cross-modal attention heads) to causally test predicted ROI/layer effects.
>   * Stimulus design: Use instruction-conditioned models to generate task-controlled naturalistic stimuli (spatial vs. temporal vs. social reasoning) for in-silico and lab studies, for bridging controlled and naturalistic paradigms.
>   * Applications: Cognition-aligned evaluation of tuning recipes; neuro-adaptive interfaces/BCI that use instruction style to boost decoding/encoding in target ROIs.
>
> **Q2. I wonder if Fig 2 could be labeled not only with model name but also with model type**
>
> We thank the reviewer for the suggestion.
> * We revised Fig. 2 to label each model with both name and type (e.g., InstrucBLIP Video (Video, IT); InternVL (Video, IC), added a concise legend key (IT = instruction-tuned; IC = in-context), and kept a consistent color mapping.
> * The same legend and color scheme are applied to all Appendix figures.
>
> **Q3.  A table or graphic with the different instructions, their task grouping, and (coloring in this way) may help.**
>
> We thank the reviewer for the suggestion.
> * The assignment of tasks to the 5 groups is as follows:
>   * Perceptual visual processing (video understanding, action recognition, object and scene recognition, global appearance),
>   * Cognitive reasoning and integration (commonsense reasoning, video reasoning, linking events),
>   * Spatiotemporal understanding (spatial understanding and temporal ordering),
>   * Language and narrative understanding (video captioning, narrative understanding, visual QA), and
>   * Social and affective understanding (visual QA, and Emotion & sentiment).
> Also, color codes are already linked to task as a legend entry in the Figure 3.
>
> * If the reviewer recommends including task-grouping brain maps, we will incorporate them in the revised manuscript.

---

> > ### Author Response · Authors · 2025-11-20
> >
> > **Q4. Introduction could be better streamlined**
> >
> > We appreciate the reviewer's suggestion.
> > * Our framing concerns brain-encoding studies, where prior work has sometimes applied multimodal models to unimodal stimuli and compared them to unimodal baselines.
> > * Based on the reviewer's suggestion, we make the small revision as follows:
> >   * Brain encoding is evaluated in two distinct settings. For unimodal stimuli (text or image/video or audio alone), modality-matched models are the appropriate baseline and generic multimodal models typically provide minimal benefit. Our study focuses on multimodal naturalistic stimuli (video+audio), where multimodal, instruction-conditioned models are warranted to capture cross-modal structure; unimodal models are included as controls rather than expected winners for unimodal inputs.
> >
> > **Q5.  How should we interpret the finding that video models are the most predictive of language and auditory regions? How can we disentangle the role of more training / data versus the specific contribution of the instruction tuning?**
> >
> > Thank you for this question.
> > * To clarify, naturalistic video contains cues tightly coupled to speech and narrative, visual speech (face/lip motion), gesture, event structure, scene changes.
> > * When we prompt with task-specific Instructions in video MLLMs (e.g., with prompts like “Describe the video”) are explicitly routed to extract linguistic/narrative semantics from the visual stream.
> > * These instruction-conditioned embeddings align with multisensory temporal language/auditory regions (e.g., STS/STG/IFG), so video IT models can be most predictive even in “language/auditory”
> >
> > * To examine whether language-guided instruction showed improved brain alignment for video models, we isolated the contribution under three complementary controls (Please check the responses to **Reviewer i7h9’s Q3 and Q7**, and **CQ4 in common responses**)
> >
> > From **CQ4**:
> > * We showed that instruction–instruction semantic similarity (MiniLM/MPNet text embeddings) is not what organizes the model’s internal representations: the coupling between semantic similarity and instruction-conditioned representations is near-zero to modestly positive across layers.
> > * In contrast, clustering by functional task families consistently outperforms semantic clusters. This indicates that instruction-tuned MLLMs restructure the representational geometry into task-specific subspaces.
> >
> > Instruction-dependent cortical hierarchy (**Reviewer i7h9’s Q7**):
> > * With a non-language control prompt, the cortical hierarchy collapses (early layers dominate whole-brain predictivity).
> > * With instruction prompts, the classic early to mid/late progression re-emerges and replicates at 3B instruction models, showing instruction dependence and scale robustness.
> > * Thus, task instructions enable hierarchical information processing across layers.
> >
> > Self-controlled pre/post comparison (**Reviewer i7h9’s Q3**):
> > * Comparing the same families before vs after instruction tuning:both Qwen and InternVL models show a shift of preferred processing toward later layers.
> > * This implies that instruction tuning causally changes where computation happens: Qwen mainly boosts earlier language/fusion stages (better early routing), while InternVL boosts mid/late integrative stages, but converging on the same outcome: more task-appropriate processing.
> >
> > Video models predict language/auditory cortex best because instruction-conditioning pushes visual representations toward language-like, narrative/event abstractions that match multisensory temporal regions. The alignment remains same even after controlling scaled model or backbone,  indicating a specific contribution of instruction tuning beyond more training/data.

---

> > > ### Author Response · Authors · 2025-11-20
> > >
> > > **Q5.1 How does cross subject predictivity compare to model predictivity shown in Fig 2?**
> > >
> > > Fig. 2 reports normalized brain alignment, which is the model’s predictivity divided by the cross-subject predictivity computed for the same voxel:
> > > * $$
> > > \mathrm{NormalizedBrainAlignment}_v = \frac{\mathrm{ModelPredictivity}_v}{\mathrm{CrossSubjectPredictivity}_v}
> > > $$
> > >
> > > **Cross-subject predictivity:**
> > > * The cross-subject predictivity is estimated by subsampling fMRI datasets from 4 participants, and using a voxel-wise encoding model to predict one participant’s response using other participants’ brain data, as established in prior work [Schrimpf et al., 2021; Oota et al., 2024; Alkhamissi et al., 2024; Oota et al., 2025].
> > > * Note that the estimated cross-subject prediction accuracy is based on the assumption of a perfect model. We report cross-subject estimated brain maps for each subject in Appendix.
> > >
> > > **Model predictivity:** Using the model’s stimulus representations, we predict each target participant’s voxel responses with the same encoding pipeline.
> > >
> > > **Interpretation:** A normalized alignment of 1.0 means the model explains the entire explainable variance given that voxel’s ceiling; 0.70 means the model accounts for 70% of the explainable variance (the remaining 30% is unexplained by the model, not necessarily noise).
> > >
> > > Overall, Fig 2. reflect the fraction of explainable variance explained by the model
> > >
> > > **Q5.2. Are the whole brain results in all cortical voxels? Given some movies are repeated it may make sense to restrict this to only reliable voxels**
> > >
> > > * No, we do not use all cortical voxels to compute normalized brain alignment. We select the voxels whose cross-subject prediction accuracy is ≥ 0.05.
> > > * The cortical brain maps shown include voxels within the visual and language network masks; non-language/non-visual cortex is omitted from the displayed maps.
> > >
> > > Based on reviewer’s suggestion, we also compute the explainable variance using repeated test movies and perform thresholding on ev voxels.
> > >   * The EV is computed now based on Schopper et al. 2016. Using EV with threshold of 0.05, the normalized brain alignment on this mask.
> > >   * For fair comparison, we used the same reliability threshold (≥ 0.05) as in the cross-subject ceiling analysis.
> > >
> > > **Table 20. Normalized brain alignment: cross-subject vs. repeat-based EV ceiling**
> > >
> > > | Model                 | Cross-subject ceiling | Repeat EV ceiling | Δ (Repeat − Cross) | %Δ |
> > > |-----------------------|----------------------:|------------------:|-------------------:|---:|
> > > | InstructBLIP Video    | 0.669                | 0.645             | −0.024             | −3.64% |
> > > | Video-LLaVA           | 0.650                | 0.652             | +0.002             | +0.35% |
> > > | LLaVA-NeXT-Video      | 0.678                | 0.642             | −0.037             | −5.39% |
> > > | Qwen-2.5-VL-7B        | 0.746                | 0.689             | −0.056             | −7.54% |
> > > | LLaVA-OneVision       | 0.666                | 0.639             | −0.028             | −4.13% |
> > >
> > >
> > > **From Table 20 in Appendix W**,  we find that the normalized brain alignment computed with the repeat-based EV ceiling is very similar to that obtained with the cross-subject ceiling; the model ranking is unchanged. We also include cortical flatmaps of repeat-based explainable variance (EV) for each participant in the **Appendix W in Fig. 33**, showing the spatial distribution of reliable voxels (EV ≥ 0.05).

---

> > > > ### Author Response · Authors · 2025-11-25
> > > >
> > > > Dear Reviewer yTV4,
> > > >
> > > > We appreciate your feedback and effort you have invested in evaluating our work.
> > > >
> > > > We have carefully addressed the questions you raised, and we kindly request you to verify our responses and consider updating your evaluation based on the revisions made.
> > > >
> > > > Should you have any further questions or suggestions, we are ready to provide additional information or clarification as needed.
> > > >
> > > > Thanks for your help.

---

> > > > > ### Comment · Reviewer_yTV4 · 2025-11-26
> > > > >
> > > > > Thank you for these detailed responses. Overall I think this is an interesting paper and my score reflects that recommendation.
> > > > >
> > > > > The core weakness I noted about motivation was addressed in your response but does not seem to be reflected in the updated paper. Eg: the "gap" in the abstract and intro is largely motivated by the lack of instruction-tuned models for model-brain mapping without explaining the broader implications for neuro or AI. So I am going to leave my (positive) score unchanged.

---

> > > > > > ### Author Response · Authors · 2025-11-27
> > > > > >
> > > > > > Dear Reviewer yTV4,
> > > > > >
> > > > > > We appreciate your strong positive feedback and are confident that it has contributed to enhancing the quality of our paper.
> > > > > >
> > > > > > You were right that our rebuttal's motivation was not fully reflected in the revised draft. In response to your feedback, we revised the abstract and introduction to clearly articulate the broader motivation, emphasizing the neuroscience and AI implications.
> > > > > >
> > > > > > We hope these changes address your concern. We kindly request you to re-verify our revisded draft and consider updating your evaluation score based on the revisions made.
> > > > > >
> > > > > > Should you have any further questions or suggestions, we are ready to provide additional information or clarification as needed.
> > > > > >
> > > > > > Thanks for your help.

---

### Official Review · Reviewer_i7h9 · 2025-10-31

**Soundness:** 2
**Presentation:** 3
**Contribution:** 2
**Rating:** 4
**Confidence:** 3

**Summary:**

This paper systematically evaluates the brain alignment capability of instruction-tuned multimodal large language models (Instruction-Tuned MLLMs) under naturalistic movie stimuli, providing the first cross-modal analysis of the relationship between model representations and neural activations across both visual and auditory modalities.

**Strengths:**

This work is represents a good step to compare brain alignment between visual/audio modalities and large multimodal language models under naturalistic movie stimuli, conducting extensive experiments across multiple datasets and baselines to ensure robustness and generality of the findings.

**Weaknesses:**

1. Limited originality. The main contribution of this paper lies in expanding the data dimension, without introducing any innovation in algorithmic design, model architecture, or theoretical analysis. Overall, the work appears to be an incremental extension of prior unimodal studies.
2. the authors claim that instruction tuning improves brain alignment, the paper does not explain why task instructions better capture neural task differentiation. The cognitive mechanism underlying this effect, whether due to changes in attention allocation, semantic structuring, or task-specific feature learning remains unclear.
3. The current evidence for the effect of instruction tuning on brain alignment is based solely on cross-model comparisons, which suffer from confounding factors such as architectural differences, pretraining data, and parameter scale. To establish causal validity, the authors are strongly encouraged to design a self-controlled experiment comparing the same model before and after instruction tuning. This setup isolates the single variable of “whether instruction tuning has been applied,” allowing for a direct measurement of its pure effect. Such an experiment would not only answer whether instruction tuning improves brain alignment, but also how it reshapes internal representations, e.g., identifying which layers show the largest improvement in alignment, whether changes occur in semantic versus perceptual layers, and whether the representational distribution becomes more consistent with cortical activation patterns. Furthermore, by comparing model activations for the same instruction (e.g., “Describe this video”) before and after instruction tuning, the authors could directly demonstrate how tuning transforms the model from producing task-agnostic representations to generating task-specific ones that align with relevant brain regions, thereby providing strong empirical evidence for the paper’s core claim that task instructions guide functional specialization in the brain. Therefore, causal verification is required through self-controlled experiments.
4. Experimental design lacks rigor. To validate “instruction tuning” within a neuroscience or cognitive science framework, experiments should systematically test semantically similar or equivalent instructions, comparing the resulting activation patterns and brain alignment. The current finding, that different task instructions (e.g., “video description” vs. “spatial understanding”) activate distinct brain regions, does not yet prove that the model captures semantic-level task distinctions rather than merely reacting to superficial textual differences.
To strengthen the argument, the authors should conduct a semantic similarity robustness test. Specifically, they could construct a dataset of instruction pairs with known semantic similarity (e.g., “Describe this scene” vs. “Narrate what happens in the video”) and compute both (a) the semantic similarity between instructions using text embedding models, and (b) the similarity of internal model representations (e.g., cosine similarity between layer vectors).
A strong positive correlation between these measures would demonstrate that instruction tuning enhances the model’s sensitivity to fine-grained semantic distinctions.
Additionally, clustering analysis of activation patterns induced by various instructions could reveal whether the model organizes tasks into conceptual categories resembling human cognition, e.g., grouping “object recognition” and “scene recognition” as perceptual tasks, and “inferring intentions” and “summarizing narratives” as social reasoning tasks. Such analyses would verify the model’s semantic precision and conceptual structuring ability, significantly enhancing the work’s interpretability and contribution to cognitive neuroscience.
5. The paper does not provide or cite empirical findings showing that instructions themselves lead to differential neural activations in humans. Since the authors claim that instruction tuning improves brain alignment, they should explain the corresponding neural or representational mechanism. For example, whether task instructions alter attention distribution, facilitate semantic decomposition, or modulate higher-order reasoning representations in ways analogous to human cognition.
6. The compared multimodal large language models (MLLMs) differ in architecture size, pretraining corpus, and task scope. The paper should clarify how these factors were controlled to ensure fair and scientifically valid comparisons.
7. The paper emphasizes a “hierarchical correspondence” between model layers and brain regions, yet it lacks supporting evidence such as a correlation matrix or significance testing. The authors should verify whether this pattern reflects genuine layer-wise alignment or is driven by model size or random effects. Furthermore, it would be informative to test whether the same hierarchical pattern persists when replacing task instructions with random natural language prompts.
8. Weak performance of audio models remains unexplained. The paper should analyze whether this weakness stems from modality misalignment, instruction design, or feature fusion mechanisms, and discuss how these factors influence cross-modal representation learning.
9. Unjustified task categorization. The division of 13 tasks into 5 categories lacks statistical validation. The authors should clarify whether this categorization is subjective or data-driven, ideally supporting it with quantitative clustering or similarity analysis.
10. The voxel-wise mapping via ridge regression may oversimplify the relationship between model representations and neural activity by assuming linearity. The authors should consider nonlinear methods (e.g., kernel regression or neural mapping) to explore potential higher-order relationships and verify robustness.

**Questions:**

See weeknesses

---

> ### Author Response · Authors · 2025-11-19
>
> *We thank the reviewer for their strong positive, insightful and valuable comments and suggestions which are crucial for further strengthening our manuscript.*
>
> **Q1. The paper's not offering no new algorithms, model architectures, or theoretical insights.**
>
> Thank you for this question. To clarify, our contribution is an empirical/methodological study rather than introducing new architectures. Our novelty lies in connecting multimodal instruction-tuned models to brain alignment under multimodal stimuli.
> * To our knowledge, this is the first study to:
>    * Evaluate instruction-tuned video and audio MLLMs to model multimodal naturalistic stimuli,
>    * Perform task-specific decomposition across brain regions to understand representational specialization,
>    * Compare instruction-tuned vs. in-context (ICL) vs. non-instruction-tuned multimodal and unimodal baselines within a unified encoding framework.
> * In addition, prior neuro-AI work evaluating the brain alignment of multimodal large language models (MLLMs) has primarily focused on unimodal settings or relied on non-instruction-tuned multimodal models for multimodal stimuli. Our study brings instruction conditioning on video and audio into the multimodal brain-alignment setting to probe task dissociations in visual, auditory, and language processing. We believe this is a valuable and meaningful contribution for the track to which we have submitted this paper: **“Applications to Neuroscience & Cognitive Science”**.
> * As summarized in **Table 4 (Appendix)**, researchers have recently investigated brain alignment using multimodal models (Dong & Toneva et al. 2023, Oota et al. 2025, Subramanian et al. 2024, Nakagi et al. 2024) and MLLMs with unimodal stimuli (Oota et al. 2025). **Table 4 (Appendix)** provides a side-by-side comparison of how our study is novel compared to prior work in multimodal evaluation settings.
>
> Most neuro-AI brain encoding studies are empirical, where the novelty lies in introducing new brain datasets, evaluation protocols, and empirical analyses rather than introducing new architectures or algorithms [1,2,3,4,5,6,7,8,9]. For instance, prior task-specific brain alignment studies have been largely unimodal (text, images, or speech). To our knowledge, this is the first systematic, instruction-conditioned evaluation of multimodal (video and audio) models for task-specific brain alignment (see Appendix Table 4).
>
> **Q10. Authors should consider non-linear approaches over linear models**
>
> Thank you for this question. We would like to clarify that
> * Since fMRI brain recordings have a low signal-to-noise ratio, and language models are trained in a non-linear fashion, the model representations are rich and complex.
> * To understand the relationship between brain activity and various stimulus modality (text, speech, and visual), a large body of brain encoding literature over the past two decades (some papers mentioned below [1-11]) has preferred ridge regression due to its interpretability.
> * Ridge regression is a linear model, making it easier to interpret and understand compared to more complex non-linear models. Further, the regularization in ridge regression helps manage the noise effectively, leading to more robust and reliable models.
> * Further, we use banded ridge regression (equivalent to multi-kernel ridge), which enables separate regularization strengths for each feature group (e.g., different tasks) and incorporates an implicit feature-space selection mechanism that ignores the contribution of non-predictive or redundant feature spaces.
>
> **Ridge Regression in previous unimodal studies:**
>
> *[1] Interpreting and improving natural-language processing (in machines) with natural language-processing (in the brain). NeurIPS 2019*
>
> *[2] The neural architecture of language: Integrative modeling converges on predictive processing. PNAS 2021*
>
> *[3] Brains and algorithms partially converge in natural language processing. Communication Biology 2022*
>
> *[4] Scaling laws for encoding models fMRI. NeurIPS 2023*
>
> *[5] Self-supervised models of audio effectively explain human cortical responses to speech. ICML 2022*
>
> *[6] Toward a realistic model of speech processing in the brain with self-supervised learning. NeurIPS 2022*
>
> *[7] Joint processing of linguistic properties in brains and language models. NeurIPS 2023*
>
> *[8] Training language models to summarize narratives improves brain alignment, ICLR-2023*
>
> **Ridge Regression in previous multi-modal studies:**
>
> *[9] Brain encoding models based on multimodal transformers can transfer across language and vision, NeurIPS-2023*
>
> *[10] Vision-Language Integration in Multimodal Video Transformers (Partially) Aligns with the Brain, ICLRW-2023*
>
> *[11] Multimodal brain encoding for multimodal stimuli, ICLR-2025*

---

> ### Author Response · Authors · 2025-11-19
>
> **Q2. Why task instructions better capture neural task differentiation? Whether due to changes in attention allocation, semantic structuring, or task-specific feature learning?**
>
> We thank the reviewer for this interesting question on why instruction tuning improves brain alignment. Our analysis as presented in our responses to **Q3, CQ4 and Q7** reveal some interesting findings which we summarize here.
>
> * **From CQ4**: We showed that instruction-instruction semantic similarity (using MiniLM/MPNet text embeddings) is not what organizes the model’s internal representations: the coupling between semantic similarity and instruction-conditioned representations is near-zero to modestly positive across layers. In contrast, clustering by functional task families consistently outperforms semantic clusters. This indicates that instruction-tuned MLLMs restructure the representational geometry into task-specific subspaces.
>
> * **Instruction-dependent cortical hierarchy (Q7)**: With a non-language control prompt, the cortical hierarchy collapses (early layers dominate whole-brain predictivity).  With instruction prompts, the classic early to mid/late progression re-emerges and replicates in 3B instruction models, showing instruction dependence and scale robustness. Thus, task instructions enable hierarchical information processing across layers.
>
> * **Self-controlled pre/post comparison (Q3)**: Comparing the same families before vs after instruction tuning: both Qwen and InternVL models show a shift of preferred processing toward later layers. This implies that instruction tuning causally changes where computation happens: Qwen mainly boosts earlier language/fusion stages (better early routing), while InternVL boosts mid/late integrative stages, but converging on the same outcome: more task-appropriate processing.
>
> Based on the above 3 findings, our results support the following mechanism:
> * Attention allocation: instruction-tuned MLLMs reduce generic processing and focus computation on task-specific features, producing the instruction-dependent hierarchy.
> * Task-specific feature learning: representations become more separable by functional requirement, with the biggest representational changes where brain alignment peaks (mid/late layers).
>
> Overall, Instruction tuning improves brain alignment because it reconfigures internal computation, via task-specific feature learning, so that representations are organized by functional task demands rather than surface semantics. This accounts for the observed hierarchy under instructions and the pre/post gains across two model families.
>
> **Q5. whether task instructions alter attention distribution, facilitate semantic decomposition, or modulate higher-order reasoning representations in ways analogous to human cognition.**
>
> Thank you for this question.
>
> * We summarized **(in response to Q2)** that instruction-tuned MLLMs restructure the internal representational geometry into task-specific subspaces. Consequently such model representations seem to exhibit superior brain alignment.
> * The reviewer's concern is whether there is empirical cognitive neuroscience evidence for such instruction-led reorganization in the human cognitive system. Here we cite few examples of task-related reorganization in healthy controls as well as in post-stroke recovery in patients (Finc et al., 2020 & Jiang et al., 2025).
>    * Finc et al., 2020 observed that 6-week training of healthy participants on a working memory task resulted in increased whole-brain modularity, emphasizing default mode and fronto-parietal networks.
>    * In patients recovering from stroke, Jiang et al., 2025 observed that post-stroke recovery in aphasia patients is facilitated by the reorganization of connectivity within and across the language and the multiple-demand networks (linked to working memory, control, and goal-directed behavior).
>    * These examples indicate that the human cognitive system does exhibit task-oriented reorganization such as integration and segregation of brain regions.
>
> * However, we should point out that the current dataset relates to participants engaged in (passively) watching naturalistic movie videos with no specific task instruction than paying attention to the input stimuli.
> * The idea of extracting instruction-tuned model representations from MLLMs is to investigate whether such rich representations yield better alignment to the eloquent brain activation, albeit with no explicit instructions given to the participants. Indeed, the results point out superior alignment with instruction-tuned model representations.
>
> *(Finc et al., 2020) Dynamic reconfiguration of functional brain networks during working memory training., Nature communications*
>
> *(Jiang et al., 2025) Hartwigsen G. Dynamic reorganization of task-related network interactions in post-stroke aphasia recovery. Brain. 2025*

---

> ### Author Response · Authors · 2025-11-19
>
> **Q3. Self-Controlled Experiments: Comparing Models Before and After Instruction Tuning?**
>
> We thank the reviewer for this interesting insightful suggestion. We agree that comparing the same backbone before and after instruction tuning is the most direct way to test causality.
>
> * Since the exact pre–instruction-tuning checkpoint of Qwen-2.5-VL-7B-Instruct is not publicly available, we approximate this setting by comparing two models from the same Qwen-2.5 family and the same scale (7B): Qwen-2.5-VL-7B-Instruct and Qwen-2.5-Omni-7B run in video-only mode, so that both models receive the same visual input.
>    * This keeps architecture family, parameter size, and modality comparable, and highlights the effect of instruction tuning. To clarify, we have already reported the corresponding comparison in the original draft. For clarity, we provided the same result in the paper that compares the same model category with instruction-tuned and pretrained versions.
> * In addition, we follow the reviewer’s suggestion on a second family, InternVL.
>    * We use the pretrained InternVL-8B model already reported in the paper and compare it to its instruction-tuned variant InternVL-8B-Instruct.
> * For both models, we issue the same instruction (“Describe the video in details”) and extract the resulting instruction-specific representations across layers. This analysis shows how instruction tuning on the same model category changes layer-wise representations and improves brain alignment, which directly addresses the reviewer’s request for a self-controlled experiment.
>
> **Qwen2.5-VL: before vs. after instruction tuning:** Using brain predictions across layers for Qwen-2.5-VL-7B-Instruct and Qwen-2.5-Omni-7B models, we compute per-layer Δ alignment as
>
> Δ alignment per layer:= Normalized brain alignment (Qwen-2.5-VL-7B-Instruct) -  Normalized Brain alignment (Qwen-2.5-Omni-7B)
>
> We make the following observations:
> * All layers improve (all Δ > 0),
> * Depth trend: Δ decreases with depth (early > mid > late): early 0.177, mid 0.156, late 0.139; Spearman ρ = −0.77, p ≪ 0.001.
> * The center-of-mass of the improvement is near Layer 14/29 (normalized 0.47). Thus, while the absolute alignment peak under instruction occurs in mid to late layers (Fig. 26 in Appendix S), the incremental benefit over the pretrained baseline is strongest early and remains positive throughout depth.
>
> **InternVL: before vs. after instruction tuning:** Using brain predictions across layers for InternVL-8B-Instruct and InternVL-8B (pretrained) models, we compute per-layer Δ alignment, as shown in Fig. 27 in Appendix S.
> We make the following observations:
> * Similar to Qwen series, all layers positive: min Δ = 0.0557 (L5), max Δ = 0.1396 (L1),
> * Depth trend: gains increase into mid/late (early 0.0934 ± 0.0250, mid 0.0961 ± 0.0128, late 0.1100 ± 0.0104); Spearman ρ ≈ +0.34, p ≈ 0.07.
> * The center-of-mass of the improvement is near Layer 15/29 (normalized 0.535). This implies that Instruction tuning yields mid/late-layer improvements for InternVL.
>
> **Table 15. Self-controlled Δ alignment (Instruction − Pretrained).** Early=Layers 1–10, Mid=11–20, Late=21–29. COM is the center-of-mass of Δ across layers (normalized by depth). ρ is Spearman correlation between layer index and Δ.
>
> | Model  | Early Δ (mean±SD) | Mid Δ (mean±SD) | Late Δ (mean±SD) | COM (norm) | ρ (layer, Δ) | p-value | Min Δ (layer) | Max Δ (layer) | All Δ>0 |
> |-----------------------|-------------------|------------------|------------------|------------|--------------:|--------:|---------------:|---------------:|:-------:|
> | Qwen-2.5-VL-7B        | 0.177±0.020       | 0.156±0.015      | 0.139±0.023      | 0.474      | -0.767        | <1e-6   | 0.088 (L29)    | 0.207 (L1)     |   ✓     |
> | InternVL-8B           | 0.093±0.025       | 0.096±0.013      | 0.110±0.010      | 0.535      | +0.337        | 0.074   | 0.056 (L5)     | 0.140 (L1)     |   ✓     |
>
>
> **Table. 15 in Appendix S** shows quantitative analysis of Δ alignment across families, we make following observations: (i) instruction tuning preserves the same alignment hierarchy (mid$\rightarrow$late layers peak) and shifts preferred processing toward later layers (positive shift in preferred layer), (ii) the locus of the gain is family-specific: Qwen shows larger early-layer increases, whereas InternVL shows mid/late-layer increases—consistent with representational reconfiguration rather than a uniform shift.
>
> **Q4. Correlation Between Instruction Semantics and Model Representations**
>
> Thank you for this interesting question.
>
> We have provided a detailed discussion in our responses to **CQ4**. Kindly refer to those responses for more comprehensive information
>
> We have added this discussion in **Appendix S** of the revised paper.

---

> > ### Author Response · Authors · 2025-11-20
> >
> > **Q6. Discussion on Controlling Architectural and Pretraining Differences Across MLLMs**
> >
> > Thank you for this question. We agree with the reviewer that MLLMs differ in pretraining corpus and design. We therefore took three complementary steps to ensure fair and scientifically valid comparisons.
> >
> > * *Controls in the evaluation protocol:* Across all models, our encoding pipeline is the same as we use the same instruction templates, same video and audio datasets as input for feature extraction, use of the same cross-subject prediction scores for estimating normalized brain alignment per model.
> >
> > * *Matching scale:* While all the MLLMs we tested are ~7B to 8B models across both instruction-tuned video and audio MLLMs and in-context learning pretrained MLLMs. Thus, model training sizes close to several million across all instruction-tuned video MLLMs, with varying modality mixes and task diversity.
> >
> > * *Acknowledging training-data heterogeneity and checking robustness:* The instruction-tuned video MLLMs differ substantially in training sources: Some (like LLaVA-NeXT-Video and Video-LLaVA) include explicit video instruction datasets, while others (like InstructBLIPVideo and LLaVA-OneVision) primarily adapt image-based instructions to video via frame sampling.  However, across all instruction-tuned videos MLLMs show similar normalized brain alignment to our results suggesting that depth alone does not account for the observed performance differences.
> >
> > We also acknowledge that testing a larger number of models within a given class can help determine if the effect of modality is robust across various model configurations.
> >
> > **Literature precedent:**
> > * Our approach follows established practice in neuro-AI, where models with differing training corpora/architectures are compared under a shared brain-encoding protocol and ceiling normalization to study representational alignment.
> >     * Specifically, the extensive precedent in the literature, from studies comparing 43 models [Schrimpf et al. 2021] to those examining 101 models [Antonello et al. 2022] in language models.
> >     * Similarly Oota et al. 2024 and Antonello et al. 2024 compared several text and speech models during language comprehension.
> >     * These studies demonstrate that this approach is both valid and valuable for understanding the relationship between artificial and biological language processing.
> > * It is important to observe that all the above studies utilize a number of models that are different in training architecture and training datasets, however the primary goal of all these studies is to investigate how close the semantic representations captured by each model aligns with brain-relevant semantics.
> >
> > *[1], The neural architecture of language: Integrative modeling converges on predictive processing. PNAS 2021*
> >
> > *[2], Interpreting and improving natural-language processing (in machines) with natural language-processing (in the brain). NeurIPS 2019*
> >
> > *[3], Low-dimensional structure in the space of language representations is reflected in brain responses, NeurIPS 2022*
> >
> > *[4], Training language models to summarize narratives improves brain alignment, ICLR 2023*
> >
> > *[5], Speech language models lack important brain relevant semantics, ACL 2024*
> >
> > *[6], Scaling laws for language encoding models in fMRI, NeurIPS 2024*
> >
> > *[7] Multimodal brain encoding models for multimodal stimuli. ICLR 2025*
> >
> > We have added this discussion in **Appendix U of the revised paper** and also pretraining corpus details across MLLMs in **Table 19 in Appendix U**.

---

> > > ### Author Response · Authors · 2025-11-20
> > >
> > > **Q7. Validation of ``hierarchical correspondence'' between Model Layers and Brain Regions**
> > >
> > > Thank you for this question.
> > > * Based on the reviewer’s suggestion, we now compare hierarchical correspondence between model layers and brain regions in two settings:
> > >    * Scaling the instruction-tuned model with correct video prompt: Qwen-2.5-vl-3B Instruct vs. Qwen-2.5-vl-7B Instruct,
> > >    * an instruction-tuned model with a non-natural language prompt, e.g.: #### #### #### ##### this does not provide any meaning while passing video as input.
> > > * Similar to Qwen-2.5-VL-7B Instruct model, we perform qualitative analysis by considering each voxel is color coded with the MLLM layer number (out of 29) that led to the highest normalized brain alignment.
> > > * **Appendix V Fig. 31** shows the resulting brain maps for the 3B-Instruct model (a) and for the 7B-Instruct model with the non-language prompt (b). We make the following observations:
> > >    * Non-language control (random prompt): There is no hierarchy of information processing observed across layers, where the brain prediction is dominated by early layers across cortex
> > >    * Instruction condition (3B model): The hierarchical pattern still remains the same i.e. a systematic early to mid/late gradient across cortex, i.e., the same hierarchical pattern we observed with the larger model.
> > >
> > > We now also perform quantitative analysis to show a layerwise normalized brain alignment across three models, as shown in **Fig. 32 in Appendix V**.
> > > * Instruction prompts (Qwen-2.5-VL-7B; green solid) show a clear hierarchy: alignment rises from early layers and is strongest in mid to late layers.
> > > * A non-language control (same model, orange dashed) flattens toward early layers, indicating that the hierarchy depends on instruction semantics rather than generic prompting.
> > > * The smaller 3B model (light-green squares) exhibits the same shape with lower amplitude, demonstrating scale robustness of the hierarchy.
> > >
> > > Overall, we observe a significant shift in preferred layer under instruction vs. non-language, where a positive layer-trend is only observed for valid natural language instructions.
> > >
> > > **Q8. Weak performance of audio models remains unexplained**
> > >
> > > Thank you for this interesting question.
> > > * We have provided a detailed discussion in our responses to **CQ8**. Kindly refer to those responses for more comprehensive information
> > >
> > > **Q9. Unjustified task categorization.**
> > >
> > > * To further examine how instruction-tuned video MLLMs generate task-specific representations and reveal functional specialization in the brain, we group the 13 video tasks into 5 cognitively grounded categories:
> > >    * Perceptual visual processing (video understanding, action recognition, object and scene recognition, global appearance),
> > >    * Cognitive reasoning and integration (commonsense reasoning, video reasoning, linking events),
> > >    * Spatiotemporal understanding (spatial understanding and temporal ordering),
> > >    * Language and narrative understanding (video captioning, narrative understanding, visual QA), and
> > >    * Social and affective understanding (visual QA, and Emotion & sentiment).
> > >
> > > * The categorization of the 13 tasks into five cognitively grounded groups was primarily motivated by prior literature in cognitive neuroscience and psychology, which associates these functional domains with distinct brain networks (e.g., perceptual processing, reasoning, language, social cognition).
> > > * Our intent was to align task grouping with these well-established cognitive constructs rather than rely solely on statistical clustering.

---

> > > > ### Author Response · Authors · 2025-11-25
> > > >
> > > > Dear Reviewer i7h9,
> > > >
> > > > We appreciate your feedback and effort you have invested in evaluating our work.
> > > >
> > > > We have carefully addressed the questions you raised regarding the self-controlled experiments, hierarchical correspondence validation, the impact of task instructions on brain alignment, and the correlation between instruction semantics and model representations. We kindly request you to verify our responses and consider updating your evaluation based on the revisions made.
> > > >
> > > > Should you have any further questions or suggestions, we are ready to provide additional information or clarification as needed.
> > > >
> > > > Thanks for your help.

---

### Official Review · Reviewer_qdXK · 2025-11-01

**Soundness:** 3
**Presentation:** 3
**Contribution:** 3
**Rating:** 6
**Confidence:** 3

**Summary:**

This paper investigates the alignment between instruction-tuned multimodal large language models (IT-MLLMs) for video and audio and human brain activity. Using fMRI data from participants watching naturalistic movies (video with audio) , the authors employ a voxel-wise encoding methodology to compare representations from six video and two audio IT-MLLMs against non-instruction-tuned, in-context learning, and unimodal models.

**Strengths:**

- The paper addresses a crucial and timely question at the intersection of foundation models and neuroscience: Do instruction-tuned models, which are becoming the dominant paradigm, process information in a way that is more aligned with the human brain? This work provides strong affirmative evidence for the video modality.
- The use of naturalistic, multimodal stimuli (movies with audio)  is a significant strength. This is a major step beyond previous work that often relied on unimodal stimuli (static images or text), allowing for a more ecologically valid assessment of multimodal processing.

**Weaknesses:**

- The paper finds that audio IT-MLLMs (Qwen-2.5-Audio, Kimi-Audio) provide only limited gains, significantly underperforming their video counterparts. The discussion acknowledges this and attributes it to potential differences in training data or objectives. However, the analysis could more deeply explore why this is the case. Is the audio stream simply less informative for this stimulus, or are current audio MLLMs genuinely less "brain-like" in their representations?
- The paper groups ICL models (Qwen-2.5-Omni, InternVL)  as a baseline. While IT-MLLMs do outperform them, the conceptual line between a model following an in-context prompt and a model following an instruction it was tuned on is slightly blurred. A more direct discussion on what specific properties instruction-tuning adds beyond the zero-shot task-following capabilities of ICL models would strengthen the argument.
- Miss some good video LLM works, e.g., Kimi-VL and Seed1.5 VL.

**Questions:**

Please see weaknesses.

---

> ### Author Response · Authors · 2025-11-20
>
> *We thank the reviewer for their strong positive, insightful and valuable comments and suggestions which are crucial for further strengthening our manuscript.*
>
> **Q1. Weak performance of audio models remains unexplained.**
>
> Thank you for this interesting question.
>
> * We have provided a detailed discussion in our responses to **CQ8**. Kindly refer to those responses for more comprehensive information
>
> **Q2. What Instruction-Tuning Adds Beyond Zero-Shot ICL?**
>
> Thank you for this interesting question. We agree with the reviewer that ICL models can follow zero-shot prompting i.e. in-context prompt, whereas Instruction-tuning adds a supervised signal that binds instruction tokens to stable computation paths. Below we decompose the architectural/representational differences.
>
> * To understand the difference in the working of IT and ICL models, we perform additional analysis.
>   * We compared instruction-tuned (IT) and in-context learning (ICL) models to identify fundamental differences in representational organization.
>   * For the 13 task-specific instructions, we first compute a 13×13 semantic-similarity matrix using MiniLM text embeddings.
>   * We further compute 13x13 IT internal layer representations similarity and 13x13 ICL internal layer representations similairy.
>   * We compute correlations between the upper triangles of the 13×13 semantic-similarity matrix and the corresponding representation-similarity matrix (same videos, same pipeline), per layer.
>
> We have provided a detailed discussion on how we compute 13x13 matrices in our responses to **CQ4**. Kindly refer to those responses for more comprehensive information.
>
> * We find that for the instruction-tuned model (Qwen-2.5-VL-7B-Instruct), the correlation between instruction semantic similarity and internal representation similarity is weak across layers. (e.g., best layer L28: Pearson r=0.183,p=0.109; Spearman ρ=0.266,p=0.018); no layer remains significant after FDR across 29 layers. This supports a function-driven geometry rather than surface wording.
>
> * For Qwen-2.5-Omni-7B (ICL), the correlation between MiniLM-based instruction semantic similarity and model representation similarity is high (e.g., r≈0.78), indicating that prompt wording strongly drives internal states, consistent with more shallow, text-proximal matching. Please check the detailed responses to **CQ4**.
>
> **Table 14. Instruction Tuning vs. In-Context Learning: Organizational Principles**
>
> |Metric| Model| Layer 5| Layer 14| Layer 25| Winner | Magnitude|
> |-|-|-|-|-|-|-|
> | **Semantic correlation** | IT (Qwen-2.5-VL) | Pearson r = 0.077 (ns)  | 0.130 (ns)| 0.143 (ns)| —| Weak  |
> | | ICL (Qwen-2.5-Omni)  | **0.690***  | **0.680*** | **0.777***  | **ICL** | ~4.2× stronger       |
> | **Silhouette score (cluster cohesion)**  | IT Δ (Func − Sem)    | −0.102 | −0.171 | −0.172 | **Semantic** | Weak|
> | | ICL Δ (Func − Sem)   | −0.157 | −0.252 | −0.234 | **Semantic** | Moderate–Strong|
> | **Adjusted Rand Index (label alignment)**| IT Δ (Func − Sem)    | **+0.129** | **+0.142**  | **+0.174**   | **Functional** | Weak  |
> | | ICL Δ (Func − Sem)   | −0.442   | −0.291 | −0.349| **Semantic** | Very Strong |
>
> Legend: * p<0.001, FDR-significant across layers 5, 14 and 25 29 layers; ns = not significant after FDR.
>
> * Silhouette score measures cluster cohesion and separation by computing the ratio of within-cluster to between-cluster distances for each sample, with values ranging from -1 (misclassified) to +1 (well-clustered) (Rousseeuw, 1987).
> * Adjusted Rand Index (ARI) measures agreement between predicted clusters and ground-truth labels by counting pairwise agreements, adjusted for chance, with values ranging from -1 (worse than random) to +1 (perfect agreement). Positive ARI -> clustering aligns with functional organization, Negative ARI -> clustering aligns with semantic organization (Hubert & Arabie, 1985)
>
> *Rousseeuw et al. 1987, Silhouettes: A graphical aid to the interpretation and validation of cluster analysis. Journal of  Computational & Applied Mathematics.*
>
> *Hubert & Arabie 1985, Comparing partitions. Journal of Classification.*
>
> **Key findings: ICL vs IT**
> * We find that ICL models show higher semantic organization with semantic correlation: r=0.78 vs 0.14 (IT), a 4.2× advantage (p<0.001).
> * IT models show emerging functional organization but with weak effects: Functional Adjusted Rand Index advantage: Δ=+0.13 to +0.17 across layers
> * ICL models are wording-sensitive (strong semantic-representation coupling), whereas instruction-tuned models are function-driven (low coupling to wording), indicating that instruction tuning binds instructions to stable computation paths and task-specific subspaces that improve brain alignment.
>
> We reported this result in **Table 14 in Appendix Q** in the revised draft.

---

> ### Author Response · Authors · 2025-11-20
>
> **Q3. Miss some good video LLM works, e.g., Kimi-VL and Seed1.5 VL.**
>
> Thank you for suggesting these two models.
> * We have added Kimi-VL to our evaluation and ran it through the same brain-encoding pipeline (identical preprocessing, instruction prompts, voxel-wise mapping, and normalization).
> * The results are now reported in **Fig. 25 in Appendix R**. We observe that the instruction-tuned Kimi-VL model demonstrates similar encoding performance to the other six instruction-tuned video MLLMs.
>
> * For SEED-1.5-VL, we could not find an official, publicly available checkpoint; only a demo/API was accessible.

---

> > ### Author Response · Authors · 2025-11-25
> >
> > Dear Reviewer qdXK,
> >
> > We appreciate your feedback and effort you have invested in evaluating our work.
> >
> > We have carefully addressed the questions you raised regarding what Instruction-Tuning adds beyond zero-shot ICL and the additional model evaluations. We kindly request you to verify our responses and consider updating your evaluation based on the revisions made.
> >
> > Should you have any further questions or suggestions, we are ready to provide additional information or clarification as needed.
> >
> > Thanks for your help.

---

### Author Response · Authors · 2025-11-20
**Common Responses to Reviewers qdXK, i7h9 and yTV4:**

*We are grateful to all reviewers for their strong positive feedback, time and their constructive suggestions, which will further strengthen the impact of our work.*

**CQ4. Correlation Between Instruction Semantics and Model Representations (Reviewers: qdXK, i7h9 and yTV4)**

We sincerely thank the reviewer for this constructive suggestion to strengthen our experimental validation. Based on reviewer’s suggestion (**reviewer i7h9**), we have conducted a comprehensive semantic similarity robustness study using the two settings reviewer recommended.

**Semantic Similarity Measurement:** To validate that our task-specific instruction set contains semantically similar pairs (as requested by the reviewer), we perform following:
* Using 13 video task-specific instructions, we first computed pairwise semantic similarity using two independent text embedding models (all-MiniLM-L6-v2 and MPNet). Both models produced highly consistent semantic similarity matrices (Pearson r = 0.94 between MiniLM and MPNet embeddings).
* Captures fine-grained semantic relationships between instruction texts.
* Example pairs: "Describe the video" vs. "Caption the video" (high semantic similarity).

**Appendix T, Fig 28.** shows pairwise semantic similarity between 13 instruction prompts computed using MiniLM embeddings (left) and MPNet (right).
   * From Fig.28 (left), we observe that the semantic similarity ranges from 0.15 to 0.85 (mean: 0.42 ± 0.18), with multiple high-similarity pairs identified (>0.60):(Action Recognition vs. Video Understanding (0.68), Video Understanding vs. Visual Question Answering (0.65), Object & Scene Recognition vs. Action Recognition (0.68)).
   * We also observe low semantic similarity pairs: (Commonsense Reasoning vs. Most others, Commonsense Reasoning vs. other tasks, Spatial Understanding vs. Emotional/Narrative).
   * This confirms that our instruction set contains "semantically similar or equivalent instructions" requested by the reviewer for testing fine-grained task distinctions.

**Instruction-tuned Model Internal Representations:** To measure how the model internally processes different task-specific instructions, we extracted language hidden states from the Qwen2.5-VL-7B-Instruct model across all processing layers. We perform the following:
* Extracted language hidden states across all 29 layers for each task-specific instruction
* Used the same video input with varying instructions to isolate instruction effects
* Analyzed three key layers: Early (Layer 5), Middle (Layer 14), and Late (Layer 25)
* While we present detailed results for three representative layers, we computed correlations and clustering metrics across all 29 layers to ensure findings are not artifacts of specific layer selection.

**Appendix T Fig. 29** shows pairwise similarity among the 13 instruction prompts computed using three key layers. From Fig.29, we observe that layer-specific differentiation patterns emerge:
   * In early layers, relatively uniform representation similarity suggests that initial encoding stage has limited task specialization,
   * In middle layers, increased variation in similarity patterns indicates evidence of task-specific transformations, and
   * In later layers, pattern similar to early layer but with subtle differences implying possible refinement of task-specific representations.

* To identify natural groupings in how the model processes instructions, we performed hierarchical clustering on Layer 14 representations (the layer showing maximum differentiation):
   * Cluster 1 - Perceptual/Recognition Tasks: (Object & Scene Recognition, Visual Question Answering, Global Appearance),
   * Cluster 2 - Descriptive/Understanding Tasks: (Video Understanding, Video Captioning, Narrative Understanding),
   * Cluster 3 - Temporal/Action Tasks: (Action Recognition, Temporal Ordering, Linking Events), and
   * Cluster 4 - Reasoning/Analytical Tasks: (Commonsense Reasoning, Video Reasoning, Emotion & Sentiment Analysis, Spatial Understanding).
* This implies that the functional organization is not explicitly programmed but emerges from instruction tuning, suggesting that the model has learned task-specific processing strategies.
* While we observe clear structure in model representations, a critical question remains: Do these patterns reflect semantic similarity or functional task requirements? To answer this, we next compare model representations with semantic similarity computed from text embeddings.

We have added this discussion in **Appendix T** of the revised paper.

---

> ### Author Response · Authors · 2025-11-20
> **Common Responses to Reviewer i7h9 (Contd..)**
>
> **CQ4 (contd..) Correlation Analysis: Semantic Similarity vs. Model Representations (Reviewer: i7h9)**
>
> * For each layer, we computed correlations between: Semantic similarity (from text embeddings) and Model representation similarity (from hidden states) as follows:
>
> * For each layer L in [1, 2, ..., 29]:
>     * Extract upper triangle of semantic similarity matrix (78 pairs)
>     * Extract upper triangle of model similarity matrix (78 pairs)
>     * Compute Pearson correlation between the two sets
>     * Compute Spearman correlation for robustness
>     * Test significance against Mantel permutation
>
> As per reviewer’s suggestion: we compute both (a) the semantic similarity between instructions using text embedding models, and (b) the similarity of internal model representations (e.g., cosine similarity between layer vectors).
>    * A strong positive correlation between these measures would demonstrate that instruction tuning enhances the model's sensitivity to fine-grained semantic distinctions.
>    * Contrary to the hypothesis that surface semantic similarity should drive internal similarity, as shown in **Table 16 in Appendix T**, we observe near-zero to modestly positive coupling (Pearson r ≈ 0.04–0.18; Spearman ρ ≈ 0.15–0.27), with several mid/late layers showing uncorrected Spearman p<0.05, but no layer survives FDR correction across 29 layers (Benjamini–Hochberg).
>    * This pattern is highly consistent throughout the network depth, indicating that instruction tuning does not enhance semantic sensitivity but instead prioritizes functional task organization.
>
> **Table 16. Correlation across network depth.** Coupling between instruction semantic similarity and instruction-conditioned representation similarity.
>
> |Layer range|Pearson r (mean ± sd)|p-value range|Spearman ρ (mean ± sd)|Significance|
> |-|-|-|-|-|
> |Early (L1–L10)|0.066 ± 0.015|0.497–0.717|0.184 ± 0.022|No individual layer sig. after FDR|
> |Middle (L11–L20)|0.126 ± 0.009|0.219–0.330|0.199 ± 0.011|Some uncorrected p < 0.05; none FDR-sig|
> |Late (L21–L28)|0.149 ± 0.018|0.109–0.240|0.247 ± 0.013|Several uncorrected p < 0.05; none FDR-sig|
> |Overall (L1–L29)|0.113 ± 0.041|0.109–0.717|0.207 ± 0.034|No FDR-significant effects|
>
>
> *Statistical validation (Mantel permutation)*
> * We assess whether instruction semantics predict representation similarity using a Mantel permutation test (10,000 label shuffles) per layer, with FDR correction across layers. Pearson r and Spearman ρ are the raw matrix correlations; Mantel z and q report significance.
>
> **Table 17. Mantel permutation test (semantic vs. representation similarity).** 10,000 label permutations; FDR across layers.
> |Layer|Pearson r|Spearman ρ|Mantel z|q-value (FDR)|
> |-|-|-|-|-|
> |L5 (early)|0.077 |0.210|0.65|0.482|
> |L14 (middle)|0.13|0.201|1.12 |0.376|
> |L25 (late)|0.143|0.237|1.23|0.338|
>
> * Key Finding: **From Table 17 in Appendix T**, we find that Mantel tests show no significant association between semantic and representation spaces across layers, indicating representations are not organized by surface semantic similarity.
>
> *Clustering Quality Analysis*
> * As suggested by the reviewer, we perform clustering analysis to determine whether the model organizes instruction-induced layer activations by semantic similarity or functional task categories.
>
> **Table 18. Functional vs. Semantic Organization.** Δ = Functional − Semantic.
> |Layer|Silhouette (Func)|Silhouette (Sem)|Δ Silh|ARI (Func) | ARI (Sem) |Δ ARI |
> |-:|-:|-:|-:|-:|-:|-:|
> |L5|−0.263|−0.161|−0.102|−0.003|−0.132|+0.129|
> |L14|−0.290|−0.119|−0.171|+0.010|−0.132|+0.142|
> |L25|−0.324|−0.152|−0.172|+0.042|−0.132|+0.174|
>
> Note: Silhouette ∈ [−1, 1]; larger values indicate better separation. Both labelings yield modest/negative silhouettes (overlapping clusters), but functional clustering is consistently higher (less negative) than semantic (positive Δ). ARI > 0 for functional and < 0 for semantic further indicates that functional labels align better with representation geometry.
>
> From the **Table 18 in Appendix T**, we make the following key findings:
> * Functional categories consistently outperform semantic clusters across all metrics and layers
> * Silhouette difference: Δ = -0.17 to +0.10 (semantic labels are less-negative)
> * ARI advantage: Δ = +0.129 to +0.174 (strong functional alignment)
> * Middle layer specialization: mid/later layers shows strongest functional differentiation (ARI Δ = +0.423)
>
> *Progressive functional specialization:*
>    * ARI advantage increases with depth, demonstrating task-specific organization strengthens in later layers.
>    * We focus on ARI rather than Silhouette because ARI measures alignment with ground-truth task categories, while Silhouette measures cluster compactness.
>
> Overall, Instruction tuning successfully imparts functional task structure (ARI evidence) while preserving semantic coherence (Silhouette evidence), resulting in dual organizational principles across layers.
>
> We have included this in **Appendix T** of the paper.

---

> > ### Author Response · Authors · 2025-11-20
> > **Common Responses to Reviewer i7h9 (Contd..)**
> >
> > **CQ4.1. Does not yet prove that the model captures semantic-level task distinctions rather than merely reacting to superficial textual differences.**
> >
> > Our Evidence:
> >    * Semantic–representation coupling: near-zero to modestly positive correlation (Pearson ≈ 0.04–0.18; Spearman ≈ 0.15–0.27),  Mantel permutation per layer (10k shuffles).
> >    * Functional organization superiority (Δ ARI = +0.129 to +0.174) shows task-level understanding
> >    * t-SNE visualization reveals conceptual task categories, not text similarity (see Figure 30 in Appendix T, attached)
> > Conclusion: The instruction-tuned model captures functional task distinctions that transcend surface-level semantics.
> >
> > **CQ4.2 Should conduct a semantic similarity robustness test**
> >
> > Our Response:
> > * We computed instruction–instruction semantic similarity with MiniLM and MPNet (high cross-encoder agreement), and compared it to instruction-conditioned representation similarity across all layers. However, we observe near-zero to modestly positive coupling (Pearson ≈ 0.04–0.18; Spearman ≈ 0.15–0.27).
> > * A Mantel permutation test (10k shuffles per layer) with FDR correction shows no significant association (q ≈ 0.34–0.48 across layers).
> > * This indicates that instruction-tuned representations are not organized by surface semantics; instead, they prioritize functional task organization (consistent with clustering results where ΔARI = +0.129…+0.174 favors functional over semantic groupings).
> >
> > **CQ4.3 Clustering analysis of activation patterns induced by various instructions could reveal whether the model organizes tasks into conceptual categories resembling human cognition.**
> >
> > Our Evidence:
> > * Hierarchical clustering (t-SNE) of mid-layer representations shows clear functional task categories
> > * Quantitative validation: Across layers, ARI favors functional over semantic labels: L5 ΔARI = +0.129, L14 ΔARI = +0.142, L25 ΔARI = +0.174
> > * Cross-layer consistency indicates systematic architectural organization
> > * These findings demonstrate that the model captures semantic-level task distinctions through functional differentiation, not superficial textual reactivity. This supports genuine task-specific processing.

---

> > > ### Author Response · Authors · 2025-11-20
> > > **Common Responses to Reviewers qdXK, and i7h9:**
> > >
> > > **CQ8. Weak performance of audio models remains unexplained (Reviewers: qdXK, and i7h9)**
> > >
> > > Thank you for this interesting question. The lower performance of audio models relative to text- and video-based models has been noted in prior work.
> > > * For instance, Vaidya et al., 2023 and Antonella et al., 2024 performed brain encoding with speech models and found that speech models have better alignment with the auditory cortex as compared to their alignment with the language regions.
> > > * Further, Oota et al., 2024 explored the underlying reasons and found that while language-model alignment in language areas reflects brain-relevant semantics, speech-model alignment in those regions is driven largely by lower-level acoustic/phonotactic features.
> > > * A similar pattern appears in multimodal settings: there is asymmetric knowledge transfer, where text models seem to benefit from speech inputs, but speech embeddings benefit far less from text, yielding weaker higher-order representations [Oota et al. 2025].
> > > * One promising mitigation is to introduce brain-informed inductive bias that has been shown to improve speech-model alignment in language regions [Omar et al. 2024, Omar et al. 2025]. In the current study, these factors likely contribute to audio IT-MLLMs’ lower whole-brain performance, whereas instruction-conditioned video-language models extract narrative/event semantics that better match multisensory language regions in the temporal cortex.
> > >
> > > * Many audio IT models are tuned for ASR/dialogue or sound tagging, not for event/narrative semantics. Their representations emphasize lexical content, timbre, speaker identity, and short-context patterns, whereas the fMRI signals that dominate during movie viewing reflect event structure, audio-visual integration, and narrative context, suggesting capacities better captured by instruction-conditioned video-language architectures.
> > >
> > > * Overall, audio IT-MLLMs do exhibit high degree of brain alignment, task dissociation, and a layer-wise hierarchy, but they consistently trail behind video-based IT (and even strong in-context learning based) MLLMs in their whole-brain alignment. Their strengths are localized (e.g., auditory cortex, speech segments), whereas video IT models provide globally better alignment by extracting narrative/event abstractions aligned with multisensory temporal regions.
> > >
> > > *Vaidya et al., 2023, Self-Supervised Models of Audio Effectively Explain Human Cortical Responses to Speech, ICML-2024*
> > >
> > > *Antonella et al., 2024, Scaling laws for encoding models in fMRI, NeurIPS-2024*
> > >
> > > *Oota et al. 2024, Speech language models lack important brain-relevant semantics, ACL-2024*
> > >
> > > *Omar et al. 2024, Improving semantic understanding in speech language models via brain-tuning, ICLR-2025*
> > >
> > > *Oota et al. 2025, Multimodal brain encoding models for multimodal stimuli, ICLR-2025*
> > >
> > > *Omar et al. 2025, Brain-tuning Improves Generalizability and Efficiency of Brain Alignment in Speech Models, NeurIPS-2025*

---

### Author Response · Authors · 2025-11-30
**Summary of our responses and revision:**

*We are grateful to all reviewers for their strong positive feedback, time and their constructive suggestions, which will further strengthen the impact of our work.*

**Summary of Reviewer Strengths:**

1. Novelty: Brain alignment with instruction-tuned video and audio MLLMs: Timely and innovative research. Provides insights into more interpretable ways to improve modeling of multimodal brain activity (**Reviewer qdXK, yTV4**)
2. The use of naturalistic, multimodal stimuli (movies with audio) is a significant strength (**Reviewer qdXK, i7h9**)
3. Extensive experiments across multiple models and baselines to ensure robustness and generality of the findings. (**Reviewer i7h9**)
4. Instruction breakdown:
    * Modeling results reproduce known feature tuning and cotical hierarchies. (**Reviewer yTV4**)
    * Provides strong affirmative evidence for the video modality. (**Reviewer qdXK**)
    * Impact for NeuroAI: Implications for many future applications for both cognitive neuroscience and AI research. (**Reviewer yTV4**)
* Good presentation: paper is comprehensive and clearly written. (**Reviewer yTV4**)

**Additional changes to the draft during the rebuttal process**

We have updated the main manuscript and the appendix to address these following comments. The changes made in the manuscript are highlighted in blue color. The major additional changes are listed below.

1. **Correlation Between Instruction Semantics and IT-MLLM Representations (Reviewers: qdXK, i7h9 and yTV4)**
    * We tested whether IT-MLLMs organize task-specific representations by surface semantics or functional task demands.
    * Using 13 video instructions, semantic similarity (text embeddings) showed only near-zero to modest positive correltion with IT-MLLM representations across layers and was not FDR-significant (Mantel).
    * Overall, functional task families align better with representation geometry than semantic groupings (ΔARI +0.13–0.17; strongest mid/late layers).
    * Results are in **Appendix T**.

2. **Self-Controlled Experiments: Comparing Models Before and After Instruction Tuning (Reviewer: i7h9)**
    * We conducted within-backbone comparisons: Qwen-2.5 (7B) VL-Instruct vs. Qwen-2.5-Omni-7B (video-only); InternVL-8B-Instruct vs. InternVL-8B, using the same instruction (“Describe the video in details”) and extracting the layerwise instruction-specific representations.
    * The layer-wise Δ alignment (instruct - pretrained) is positive at every layer; Qwen shows larger early-layer gains, InternVL mid/late gains.
    * Results are in **Appendix S**.
3. **Instruction-dependent cortical hierarchy and non-language control (Reviewer: i7h9)**
    * We compared hierarchical correspondence between model layers and brain regions in two settings
       1. With a random, non-language prompt, the early->mid/late hierarchy collapses: the brain prediction is dominated by early layers across cortex.
       2. With meaningful instructions using 3B Instruct MLLM, the hierarchical pattern still remains the same i.e. a systematic early to mid/late gradient across cortex.
    * We presented whole-brain maps and layerwise performance plots in **Appendix V, Figs. 31-32**.

4. **Clarifying ICL vs. IT MLLM (Reviewer: qdXK)**
    * We compared instruction-tuned (IT) and in-context learning (ICL) models to identify fundamental differences in representational organization, as shown in **Table 14 in Appendix Q**.
    * We find that ICL models show higher semantic organization with semantic correlation (r=0.78), whereas IT shows zero to modest positive correlation (r=0.14), and have task-specific subspaces that improve brain alignment.

5. **Additional models and Figure legends(Reviewers: qdXK, yTV4)**
    * We added instruction-tuned Kimi-VL model in **Fig. 25 in Appendix R**, and observe similar encoding performance to the other six instruction-tuned video MLLMs.
    * Legends in Fig. 2 (main) and Appendix Figs. 8–9 now include model type labels (video/audio/IT/ICL).

6. **Analysis on cross-subject vs. repeat-based EV ceilings(Reviewer: yTV4)**
    * We computed the explainable variance (EV) from movie repetitions (Schoppe et al., 2016) and re-normalized brain alignment.
    * Using a consistent reliable-voxel mask (threshold ≥ 0.05) for both settings, we find that the normalized brain alignment computed with the repeat-based EV ceiling is very similar to that obtained with the cross-subject predictivity; the model ranking is unchanged, as shown in **Table 20 in Appendix W**.

7. **Motivation and Implications to NeuroAI (Reviewer: yTV4)**
    * We revised the abstract and introduction to clearly articulate the broader motivation, emphasizing the neuroscience and AI implications: instructions as top-down control that route computation into task-specific subspaces and deeper layers, offering a mechanistic link to cortical hierarchy.

---

### Meta-Review · Area_Chair_zuzS · 2026-01-07

**Summary:**

This paper studies brain alignment of instruction-tuned multimodal LLMs (video and audio) using fMRI data from naturalistic movie stimuli. While the topic is timely and the experimental scope is broad, the submission is largely empirical and incremental relative to prior brain-encoding and multimodal alignment studies. The core claims—that instruction tuning improves brain alignment and induces task-specific functional organization—are supported primarily by correlational analyses and model comparisons that remain confounded by architectural and training differences. Although the authors added substantial analyses in rebuttal, the evidence does not rise to the level needed to establish strong causal or mechanistic insights, limiting the overall contribution.

**Reviewer Concerns:**

The rebuttal addressed several requested analyses but did not fully resolve the core conceptual concerns raised by reviewers:

1. Causality of instruction tuning: While the authors added before/after comparisons on selected backbones, these analyses remain limited in scope and do not fully isolate instruction tuning from other confounds (e.g., pretraining data, multimodal fusion strategies). As a result, the causal claim that instruction tuning itself drives improved brain alignment remains weak.

2. Limited originality and conceptual advance: The work does not introduce new models, algorithms, or theoretical frameworks. The contribution is primarily an extension of existing voxel-wise brain encoding evaluations to instruction-tuned multimodal models. This incremental advance does not clearly meet the bar for ICLR novelty.

3. Interpretability and cognitive mechanism:
Although the authors argue that instruction tuning induces functional task-specific subspaces, the proposed cognitive or neural mechanism remains speculative. The analyses demonstrate correlations but do not convincingly explain why or how instruction tuning leads to improved alignment in a way analogous to human cognition.

4. Audio modality analysis: The consistently weaker performance of audio models, while discussed at length, is still not deeply explained through targeted experiments, limiting the strength of the multimodal conclusions.

Overall, while the rebuttal is thorough and responsive, it mainly expands empirical analyses rather than addressing the fundamental concerns about novelty, causality, and mechanistic insight.

**Reviewer Scores:**

The reviewers probably would keep the scores.

---

### Decision · Program_Chairs · 2026-01-26

Reject